# Medicane Zorbas: Origin of an uncertain potential vorticity streamer position and impact on cyclone formation

Raphael Portmann[1], Juan Jesús González-Alemán[2], Michael Sprenger[1], and Heini Wernli[1]

[1]ETH Zurich, Institute for Atmospheric and Climate Science, Zurich, Switzerland
[2]University of Castilla-La Mancha, Environmental Sciences Institute, Toledo, Spain

**Correspondence:** Raphael Portmann (raphael.portmann@env.ethz.ch)

**Abstract.** Mediterranean tropical-like cyclones (medicanes) can have high societal impact and their accurate forecast remains a challenge for numerical weather prediction models. They are often triggered by upper-level potential vorticity (PV) anomalies, such as PV streamers and cutoffs. But knowledge is incomplete about their detailed formation processes and factors limiting their predictability. This study exploits a European Centre for Medium-Range Weather Forecast (ECMWF) operational ensemble forecast with an uncertain PV streamer position over the Mediterranean, that, three days after initialization, resulted in an uncertain development of Medicane Zorbas in September 2018. An ad-hoc clustering of the ensemble members according to the PV streamer position is used and cluster differences show a region of enhanced uncertainty in the initial conditions in the upper troposphere on the stratospheric side of the jet stream over the Gulf of Saint Lawrence. This feature is advected over the North Atlantic where it amplifies rapidly on the stratospheric side of an emerging upper-level jet streak and further propagates into the Mediterranean. Uncertainties in the ageostrophic circulation associated to the jet streak contribute substantially to the initial amplification over the North Atlantic. The further amplification and downstream propagation of the tropopause-level PV uncertainty leads to a large spread in the position of the PV streamer over the Mediterranean after three days, directly limiting the predictability of the position, thermal structure and evolution of the Mediterranean cyclone. In particular, the eastward displacement of the PV streamer in more than a third of the ensemble members results in a very different cyclone scenario, where low values of low-level equivalent potential temperature and the missing erosion of the high upper-level PV prevent the development of a deep warm core that is typical for medicanes. Overall, this study illustrates that uncertainties in large-scale initial conditions and uncertainty amplification over the North Atlantic can determine the practical predictability limits of a high-impact weather event in the Mediterranean.

## 1 Introduction

### 1.1 Medicanes: high impact, limited understanding, and uncertain forecasts

In the last 15 years, research on cyclones at the interface between the classical concepts of tropical and extratropical cyclones has been increasing. Sometimes, these cyclones are called hybrid cyclones (e.g. Quinting et al., 2019). They are usually characterized by a lower-tropospheric warm core and an upper-tropospheric cold core. Frequently occurring hybrid cyclones are so-called subtropical cyclones, which mainly occur over ocean basins (e.g. Caruso and Businger, 2006; Moore et al., 2008;

Guishard et al., 2009; González-Alemán et al., 2015). Often, they initially form from elongated equatorward intrusions of stratospheric high potential vorticity (PV), so-called PV streamers, that subsequently detach from the main stratospheric high PV reservoir, forming a PV cutoff. The large-scale ascent and destabilization by the high upper-level PV favors convection, which starts to form a lower-tropospheric warm core. Fronts dissolve and the environment becomes more barotropic. Subtropical cyclones have gained attention because they can have substantial impact on society (Guishard et al., 2007), but also because

they can undergo tropical transition (Davis and Bosart, 2004), i.e. they can amplify through the so-called wind induced surface heat exchange mechanism (WISHE; Emanuel, 1986) and convert into tropical cyclones.

A similar phenomenon occurs over the Mediterranean Sea, and the resulting cyclones are sometimes called medicanes (Mediterranean hurricanes; Emanuel, 2005; Tous and Romero, 2013; Cavicchia et al., 2014) or Mediterranean tropical-like cyclones (Miglietta et al., 2013). There is not yet a clear definition of medicanes. A common property seems to be the development of

a (symmetric) warm-core structure throughout the whole troposphere, often called deep warm core. The deep warm core of medicanes is not necessarily produced directly by convection but can, especially at lower levels, be promoted by horizontal advection and the seclusion of warm air in the cyclone center, which is why it has been argued that some medicanes are rather subtropical than tropical-like cyclones (see e.g. Fita and Flaounas, 2018). Medicanes can undergo tropical transition and acquire the typical appearance of a hurricane (Emanuel, 2005) with convective cloud bands wrapped around a cloud-free central

eye and a typical size of their associated cloud clusters on the order of 300 km in diameter. In this case, convection forms and maintains a robust deep warm core and the medicanes are associated with strong horizontal pressure gradients, wind, and rainfall. This may result in high damage, although medicanes rarely attain hurricane intensity. The relative role of positive upper-level PV anomalies and air-sea interaction for the intensification of medicanes is currently debated. In most cases the positive upper-level PV anomaly seems to be important for the initial intensification (Miglietta et al., 2017).

In situations with the formation of a medicane, ensemble prediction systems often show large uncertainties in cyclone occurrence, cyclone intensity, and intensity of the upper-level warm core even at few days lead time (e.g. Di Muzio et al., 2019). For a medicane case in 2006, it was shown that beyond 36 h lead time ensemble members that were missing a PV streamer over the Mediterranean had a lower probability of forecasting a medicane (Chaboureau et al., 2012; Pantillon et al., 2013). This was linked to uncertainties in an extratropical transition event over the North Atlantic, which rapidly propagated downstream

into the Mediterranean. However, even from the ensemble members that formed a PV streamer over the Mediterranean, only about 9% actually forecasted a medicane. This indicates that even if the PV streamer is forecasted, other synoptic-scale aspects related to the PV streamer, for example uncertainty in its position, shape, or intensity may be limiting the predictability of the medicane. Also in this study about Medicane Zorbas, the position of a PV streamer in the Mediterranean varies between ensemble members and it is decisive for the evolution of the medicane. PV streamers result from the non-linear development

of Rossby waves, which is known to be susceptible to the growth of forecast uncertainties. Therefore we now briefly review the relevant literature on the origin and amplification of forecast uncertainties in Rossby waves.

## 1.2 Origin and amplification of forecast uncertainties in Rossby waves

Understanding forecast error and uncertainty is crucial to reveal the limits of and potential for improving weather forecasting systems. In recent years, research increasingly focused on the origin and amplification of forecast errors along the upper-level wave guide, i.e. the near-tropopause band with a high isentropic PV gradient (Davies and Rossa, 1998; Schwierz et al., 2004), because there they tend to be largest (e.g. Dirren et al., 2003; Hakim, 2005; Davies and Didone, 2013). Many studies used the PV framework to investigate forecast errors along the Rossby wave guide (for example Fehlmann and Davies, 1997; Dirren et al., 2003; Davies and Didone, 2013; Gray et al., 2014; Baumgart et al., 2018).

Forecast errors in Rossby waves can *originate* from errors in initial conditions or model errors (e.g. the misrepresentation of diabatic processes). In certain situations they can result in so called "forecast busts" over Europe, i.e. periods of anomalously low predictability (Rodwell et al., 2013; Magnusson, 2017; Grams et al., 2018). The misrepresentation of diabatic processes within warm conveyor belts can be an important error source for Rossby waves (Gray et al., 2014; Martinez-Alvarado et al., 2016). Warm conveyor belts affect the Rossby wave guide via their low-PV outflow in the upper troposphere (Grams et al., 2011; Madonna et al., 2014). PV errors near the tropopause represent errors in the structure and amplitude of Rossby waves that then propagate downstream.

To become relevant, forecast errors and uncertainties must *amplify*. It is well known that even slight uncertainties in initial conditions can grow with increasing forecast lead time and potentially result in very different large-scale weather patterns a few days after initialisation (e.g. Lorenz, 1969; Leutbecher and Palmer, 2008). However, the growth of forecast uncertainty is not homogeneous but depends on the flow itself. To quantify this flow-dependent growth of forecast uncertainty, operational ensemble predictions are conducted since the early 1990s and have continuously improved since then (Bauer et al., 2015; Palmer, 2019). The uncertainty can be quantified with the ensemble spread that serves as a proxy for the expected forecast error of the ensemble mean if the forecast is reliable (which is not true in all flow situations, see Rodwell et al., 2018). Different case studies have shown than forecast error in Rossby waves can amplify due to a range of processes. Baumgart et al. (2018) pointed out the importance of non-linear (barotropic) dynamics near the tropopause and to a lesser extent baroclinic interaction and effects of upper-tropospheric divergent winds. The contribution of non-linear tropopause dynamics to amplification (and downstream propagation) of forecast errors can be understood by the mutual interaction of negative and positive PV errors near the tropopause (Dirren et al., 2003; Davies and Didone, 2013; Baumgart et al., 2018). Grams et al. (2018) identified warm conveyor belts in a forecast bust case as key for amplifying forecast errors in the tropopause region. Together, these studies indicate that forecast errors in near-tropopause Rossby waves can grow just due to their internal non-linear dynamics, but baroclinic growth and warm conveyor belts and their associated divergent low-PV outflow can enhance this amplification. The relevance of these processes to the amplification of forecast errors and uncertainty is case and location dependent (Baumgart and Riemer, 2019) and their interplay potentially complex. Therefore, more case studies are useful to better understand the involved mechanisms.. In the context of high impact weather, a better understanding of the synoptic-scale elements controlling the practical predictability (i.e. the predictive skill of state-of-the-art forecasting systems), can help to improve situational awareness.

### 1.3 Zorbas: An uncertain medicane

This study investigates the ECMWF ensemble prediction initialised 84 h before cyclogenesis of Medicane Zorbas in September 2018, which is characterized by an uncertain position of the precursor PV streamer over the Mediterranean after 72 h lead time. The main objectives are: (i) to identify the origin of the forecast uncertainty and analyze the synoptic environment in which the ensemble spread amplifies and propagates; and (ii) to investigate the different large-scale scenarios over the Mediterranean offered by the ensemble and understand how they are related to the formation of a medicane. Other studies have shown that a detailed analysis of ensemble forecasts, especially of different scenarios in a particular ensemble prediction, can be highly rewarding for better understanding the involved dynamics and the practical predictability limits. For example, such scenarios have been used to identify key dynamical elements limiting the predictability of tropical cyclones (Torn et al., 2015; Pantillon et al., 2016; Gonzalez-Aleman et al., 2018; Maier-Gerber et al., 2019), medicanes (Pantillon et al., 2013), and atmospheric blocking (Quandt et al., 2017).

The remainder of this article is structured as follows. A description of the data and methods used (section 2) is followed by an overview of the synoptic evolution of Zorbas and the large-scale situation previous to cyclogenesis (section 3). Then, a pragmatic clustering is introduced that uses the uncertainty in the PV streamer position to separate the ensemble forecast into three distinct PV streamer scenarios (section 4). Section 5 discusses the origin, amplification, and propagation of the forecast uncertainty before the formation of the PV streamer. sectionion 6 presents the synoptic evolution of the three scenarios in the Mediterranean and investigates how the scenarios influence the resulting Mediterranean cyclone. Finally, in section 7, we draw the main conclusions and discuss implications for further research on uncertainty in Rossby wave forecasts and on medicane dynamics.

## 2 Data and methods

### 2.1 Data

The basic data for this study are from the ECMWF Integrated Forecasting System (IFS, Cycle 45r1; ECMWF, 2018). Operational ensemble forecasts with 50 perturbed members initialized at 0000 UTC 24 Sep 2018 and 0000 UTC 27 Sep 2018, the operational analysis, and operational short-term (12 -hourly accumulated, initialized at 0000 and 1200 UTC) forecast of 6-hourly accumulated precipitation, are used. The ECMWF operational ensemble forecast is based on perturbed initial conditions as well as stochastic perturbations of model physics (for details see ECMWF, 2018). The spectral resolution of the operational ensemble is TCO639 (about 18 km) on 91 model levels, and the resolution of the operational analysis TCO1279 (about 9 km) on 137 model levels. The data are available every 6 h and have been interpolated to a regular grid with a horizontal resolution of 1°. From the standard variables we additionally compute PV on isentropic surfaces (every 5 K), and equivalent potential temperature ($\theta_e$) on pressure levels (every 25 hPa). As a measure for forecast skill, anomaly correlation coefficients (ACC) are calculated for geopotential height at 500 hPa for each ensemble member of the forecasts initialized at 0000 UTC 24 Sep 2018 and 0000 UTC 27 Sep 2018 using the daily mean ERA-Interim climatology from 1979-2014 as a reference (for details

see supplementary material). Anomalies of 900 hPa $\theta_e$ were computed with respect to the September/October ERA-Interim climatology from 1979-2017.

Additionally, satellite data (infrared channel 9 ($10.8\,\mu$m) of MSG SEVIRI) provided by the European Organisation for the Exploitation of Meteorological Satellites (EUMETSAT) are used.

## 2.2    Cyclone phase space and cyclone tracking

The cyclone phase space (CPS; Hart, 2003), is a useful tool to diagnose the thermal structure of cyclones throughout their life-cycle. The CPS is a descriptor of the three-dimensional thermal structure of cyclones at a given timestep in terms of three

parameters: lower-tropospheric horizontal thermal asymmetry ($B$), which measures the across-track 900-600 hPa thickness gradient, i.e. frontal nature, and thermal winds in the lower ($-V_T^L$; 900-600 hPa) and upper troposphere ($-V_T^U$; 600-300 hPa), which measure the vertical thermal structure. In this three-dimensional parameter space, cyclones can be classified as frontal ($B > 0$) or non-frontal ($B \leq 0$), cold-core ($-V_T^L < 0$ and $-V_T^U < 0$), hybrid ($-V_T^L > 0$ and $-V_T^U < 0$), or deep warm-core ($-V_T^L > 0$ and $-V_T^U > 0$). In this study, cyclones that at least once in their life-cycle fulfill the deep warm-core criterion are

classified as medicanes. For simplicity, the symmetry parameter $B$ is not considered.

Cyclone tracks at 6-h temporal resolution are obtained for each of the 50 ECMWF ensemble members and the operational analysis using the cyclone detection and tracking method described by Picornell et al. (2001). This method was specifically designed to study meso-scale cyclones in the Mediterranean Sea, including medicanes (Gaertner et al., 2018). More specifically, 6-hourly SLP fields are used to identify pressure minima after applying a Cressman filter (radius of 200 km; Sinclair, 1997)

to smooth out noisy features and small cyclonic structures. Weak cyclones are then filtered with a SLP gradient threshold of 0.5 hPa per 100 km. Cyclone tracks are identified with the aid of the horizontal wind field at 700 hPa, which is considered the steering level for cyclone movement. For one member (member 32) the cyclone track showed an unrealistically large jump from the first to the second timestep and therefore the first timestep was removed from the track. Note that mean sea level pressure data of the ensemble forecast is only available in 6-hourly resolution until 144 h lead time. Therefore, all cyclone

tracks in the ensemble initialized at 0000 UTC 24 Sep 2018 end at the latest at 0000 UTC 30 Sep 2018. The CPS is calculated every 6 h based on the track positions and the CPS values at each time step are smoothed using a running mean filter with a 24-h window. Due to the small size of medicanes, a radius of 150 km is used to calculate the CPS values, consistent with previous studies (e.g. Gaertner et al., 2018) .

## 2.3    Normalized PV differences

To compare PV of two ensemble clusters at different lead times, it is useful to compute normalized cluster-mean differences (see e.g. Torn et al., 2015):

$$\Delta \mathrm{PV}_{\mathrm{AB}} = \frac{\overline{\mathrm{PV}}_A - \overline{\mathrm{PV}}_B}{\sigma_{\mathrm{PV}}} \tag{1}$$

where $\sigma_{\Delta \mathrm{PV}_{AB}}$ is the standard deviation of all ensemble members and the subscripts A and B denote any two clusters of an ensemble forecast. Hence, $\Delta \mathrm{PV}_{\mathrm{AB}}$ becomes large when the cluster-mean difference of PV between A and B at a given location

is much larger than the ensemble standard deviation at the same location, i.e. when the two clusters contain the members of the ensemble that are most different from each other. Large absolute differences in regions of strong gradients, particularly at the tropopause, are given less weight. Additionally, it allows different lead times to be easily compared. For example, if $\Delta\mathrm{PV}_{AB}$ increases with lead time, the cluster differences grow faster than the ensemble standard deviation, which means that the clusters become increasingly distinct from each other relative to the full ensemble.

## 2.4 Statistical significance

In order to be confident that the differences between two ensemble clusters are robust, a two-sided Wilcoxon rank-sum test (Wilks, 2011) for each cluster pair and considered field was applied. With such a test, the null hypothesis is investigated, that it is equally likely that at a certain grid point the value of a randomly picked ensemble member in one cluster is larger or smaller than in a randomly picked ensemble member of the other cluster. When applying such a statistical test to a field, the false discovery rate should be controlled in order to avoid over-interpretation of the results (Wilks, 2016). This can be done by correcting the $p$-values of the statistical test taking into account the number of tests. For this study we are only interested in a domain covering the North Atlantic and the Mediterranean and consider a box from 30 to 70°N and 80°E to 30°W. Therefore, a number of 4400 tests are used for the correction. As suggested by Wilks (2016), a Benjamini-Hochberg correction is used in this study and the false discovery rate is set to a rather conservative value of $\alpha_{\mathrm{fdr}} = 0.1$ for all analyses. Regions where the null hypothesis is rejected on this level of $\alpha_{\mathrm{fdr}}$ can then be used to identify where and when robust differences in the clusters emerge in the ensemble forecast.

## 2.5 Trajectory calculations

Computing trajectories provides insight into the Lagrangian history of air parcels. In this study, the Langrangian analysis tool LAGRANTO (Wernli and Davies, 1997; Sprenger and Wernli, 2015) is used to identify warm conveyor belt trajectories (ascent rate larger than 600 hPa in 48 h, see e.g. Madonna et al., 2014) and to compute backward trajectories from the cyclogenesis region (see supplementary material).

## 3 Synoptic overview

This section provides an overview of the life-cycle of Medicane Zorbas, the large-scale synoptic situation over the Euro-Atlantic region prior to its genesis, and of the synoptic elements that accompanied the first stage of Zorbas' life-cycle during which it acquired a deep warm core.

Medicane Zorbas formed at 1200 UTC 27 Sep 2018 close to Benghazi, moved into the Central Mediterranean Sea and then sharply turned eastward and moved over Greece into the Aegean Sea, where it finally decayed four days after its formation (Fig. 1a). Zorbas led to considerable damage through severe winds, torrential rainfall, major flooding and even tornadoes. The main affected region was Southern Greece, especially Crete, Peloponnese, Evia, and the region around Athens. According to the CPS (Fig. 1b), Zorbas formed as a cold-core cyclone and within 18 h acquired a deep warm core that was sustained for more

than three days. Zorbas reached its maximum intensity (992 hPa) already 12 h after cyclogenesis. Satellite images indicate the formation of an eye-like feature shortly before Zorbas reached Greece on 29 Sep, suggesting that Zorbas likely underwent tropical transition (not shown). However, this aspect of the life-cycle is not in the focus of this study. Instead, we focus on the synoptic aspects influencing cyclogenesis, the initial intensification, and the formation of a deep warm core.

The period before cyclogenesis of Zorbas was characterized by the transition from a relatively zonal jet in the Euro-Atlantic region at 0000 UTC 24 Sep 2018 (Fig. 2a) with a jet streak (J1) over the Gulf of Saint Lawrence and a moderate ridge (R) and trough (T) pattern over the eastern North Atlantic/Scandinavia to a very wavy jet at 0000 UTC 26 Sep 2018 (Fig. 2b) with an elongated PV streamer (S) over the central North Atlantic, a large ridge (R) and a north-eastward directed jet streak (J2) over the western North Atlantic, and a narrow, elongated trough (T) over eastern Europe . During this transition, the ridge and the

associated surface high pressure system (H) amplified and the latter moved from the eastern North Atlantic to eastern Europe. The PV streamer over the North Atlantic then interacted with Hurricane Leslie (L) resulting in substantial warm conveyor belt activity. Note that the anticyclonic wave breaking (i.e. the amplification of the ridge and the non-linear elongation of the trough over eastern Europe) preceded the North Atlantic warm conveyor belt activity, indicating that in this case the warm conveyor belt was less relevant for the ridge amplification than for typical events with the formation of strong Mediterranean cyclones

(Raveh-Rubin and Flaounas, 2017). Subsequently, the narrow trough over eastern Europe further elongated into a PV streamer reaching the Central Mediterranean Sea at 0000 UTC 27 Sep 2018 (see Fig. 3a).

The formation of the PV streamer over the Mediterranean was followed by cyclogenesis of Zorbas at the PV streamers' south-eastern flank and its rapid intensification (see Fig. 3a-c). At the same time, the PV streamer broke up resulting in the formation of a PV cutoff, which was rapidly eroded during the intensification of the surface cyclone. Shortly before cyclogenesis, the

205 large mean sea level pressure gradients between the surface high pressure system over eastern Europe and a surface low pressure area over the Levantine Sea resulted in strong low-level advection of air with low $\theta_e$ across the Aegean Sea (Fig. 3a,d). Air with anomalously high $\theta_e$ was present over Lybia and at cyclogenesis in immediate proximity of the cyclone center (hatched regions in Fig. 3d,e). At a first glance, it seems that the air with high $\theta_e$ was advected cyclonically into the cyclogenesis area. However, a detailed trajectory analysis (see supplementary material) shows that the low-level air in the 850-950 hPa layer and

210 within a radius of 250 km around the cyclone centre originated from the Aegean and Black Sea and was substantially moistened by sea surface fluxes as it traveled across the Aegean Sea. This is consistent with the direction of low-level winds (Fig. 3d,e). The relevance of a similar process for medicane formation in the western Mediterranean has already been pointed out by Miglietta and Rotunno (2019). A day after cyclogenesis, the cyclone developed into a barotropic system with a deep warm core structure. The surface cyclone, the low-level warm core, the minimum of geopotential height at 500 hPa, and the upper-level

PV maximum were vertically aligned (Fig. 3c,f). The upper-level PV cutoff decayed and the remaining, small-scale PV maximum with values around 2 PVU above the cyclone centre at 1200 UTC 28 Sep 2018 was most likely diabatically produced, as stratospheric upper-level positive PV anomaly would imply an upper-level cold core. The low-level maximum of $\theta_e$ was lower at this stage than at cyclogenesis and not anomalous with respect to climatological values.

The 12 hours after cyclogenesis were characterized by very intense precipitation north-west of the cyclone centre according

to the IFS short-term forecasts (red contours in Figure 4a, maximum value of 101 mm in 12 h). The dense cloud patch and

lightning activity (see supplementary material) in this area are indicative of strong latent heating and the presence of deep convection. According to the IFS model output, 38% of the area-averaged precipitation was produced by the convection scheme. This indicates that additionally strong large-scale ascent occurred. Such a situation with strong ascent and latent heating is expected to result in vorticity generation at lower levels, which is consistent with the cyclone path in this period. It also helps to explain the rapid diabatic erosion of the PV cutoff (see e.g. Portmann et al., 2018). In contrast, at the time when the deep warm core was well established, the precipitation was substantially weaker (Fig. 4b) and mostly produced by the convection scheme (67%). The cloud structure indicates that well-defined fronts were absent, i.e. Zorbas acquired a more tropical-like appearance. The evolution of Zorbas in the first day of its life-cycle agrees well with the climatological evolution found for the strongest Mediterranean cyclones around their time of maximum intensity (Flaounas et al., 2015). However, remarkable in this case was the rapid erosion of the upper-level PV cutoff, the anomalously high values of 900 hPa $\theta_e$ at cyclogenesis and the very intense precipitation in the early stage of its life-cycle. Values of 900 hPa $\theta_e$ at cyclogenesis reached above 330 K in this case. This value provides the approximate maximum isentropic level to which these air masses can ascend by the release of latent heat. In this case, the PV cutoff was located at 325 K, and therefore diabatic ascent of the low-level air present at cyclogenesis had the potential to erode the stratospheric upper-level PV anomaly.

## 4   Ensemble clustering according to position of PV streamer

In the following, a procedure is presented that allows separating the ensemble members of the 0000 UTC 24 Sep 2018 initialization into meaningful clusters based on position of the Mediterranean PV streamer at day 3 of the forecast. This is motivated by the previous section, which showed that the PV streamer is crucial for the cyclogenesis of Zorbas. At 0000 UTC 27 Sep 2018, the position of the PV streamer varies strongly in the ensemble, with about equal shares of ensemble members where the PV streamer is roughly correct, too far west and too far east, respectively. This offers the opportunity to use this ensemble forecast to study the dynamical processes that lead to this significant uncertainty in the position of this upper tropospheric PV structure. The ensemble members are clustered based on the PV streamer position at 0000 UTC 27 Sep 2018 (shown in Fig. 3a), which is immediately before the PV streamer breaks up cyclonically and cyclogenesis occurs in the operational analysis. A pragmatic clustering method was designed specifically for this situation to separate the strongly differing PV streamer positions in the ensemble members. The three identified PV streamer scenarios, i.e. clusters, are the basis for all remaining analyses. For the clustering, a box is defined around the PV streamer as identified at 0000 UTC 27 Sep 2018 in the operational analysis (Mediterranean box, 5-30°E, 30-45°N, see black box in Fig. 5). The clustering uses PV vertically averaged between 320 and 330 K, hereafter called $PV_{av}$. Before the averaging, all PV values with PV<2 PVU are set to zero to remove the contribution of the variability of tropospheric PV values. Hence, $PV_{av}$ is high in areas where the PV streamer is strong and deep, and low where it is weak and shallow. The clustering is based on two different steps: First, from all 50 ensemble members the ones are identified for which the region with $PV_{av}$ ≥2 PVU in the box has more than 75% overlap with the corresponding area in the analysis. In these 19 members (cluster C), the streamer has a similar location as in the analysis (see blue shading in Fig. 5). The

remaining members are separated into two clusters depending on whether the maximum $PV_{av}$ is shifted to the west (cluster W, 12 members, green shading in Fig. 5) or east (cluster E, 18 members, red shading in Fig. 5) relative to the analysis. There is one ensemble member that cannot be attributed to one of the three clusters because its overlap is less than 75% but the maximum of $PV_{av}$ is located at the same longitude as in the analysis. The histogram of the longitude where the maximum of $PV_{av}$ occurs between 36-73°N (inset in Fig. 5) shows three clearly distinct peaks, one for each cluster, supporting the simple clustering approach. There are a few borderline members but they don't affect the main results of this study.

The meaningfulness of this clustering for studying the predictability of this case is further supported by the fact that it helps explaining the temporal development of the anomaly correlation coefficient (ACC, for details see supplementary material) in the Mediterranean box. To this aim, Fig. 6a summarizes the synoptic sequence of the case, i.e. the formation of the PV streamer over the Mediterranean, the break-up resulting in a PV cutoff (grey boxes), the time evolution of the minimum sea-level pressure of Zorbas (solid line), and its thermal structure as diagnosed from the CPS (colors). As shown in Fig. 6b, the ACC of geopotential height at 500 hPa in the Mediterranean starts to decrease in the majority of the ensemble members, when the PV streamer reaches the Mediterranean on 26 Sep 2018 and cyclogenesis occurs, while it remains high (close to 1) until 29 Sep 2018 for most members of cluster C (blue lines in Fig. 6b). After the decrease from 1 to around 0.8 the median ACC remains fairly constant until 29 Sep 2018. In comparison, for the ensemble forecast initialized at 0000 UTC 27 Sep 2018, i.e. at the time when the PV streamer has developed, the ACC remains high in all members during the intensification and deepest phase of Zorbas, decreasing only after 29 Sep 2018 (Fig. 6c), likely due to errors associated with a second PV streamer reaching the Mediterranean in the northern part of the box (not shown). It can be concluded that errors in the position of the PV streamer limited the large-scale predictability as measured by the ACC of geopotential height on 500 hPa in the Mediterranean, and that cluster C contains the members with the most accurate forecasts.

## 5  PV streamer scenarios emerge from uncertainty amplification in a North Atlantic jet streak

We now investigate how the diverging PV streamer scenarios identified in section 4 evolved from differences in the initial conditions and a jet streak over the North Atlantic. To this end, the normalized PV differences (see section 2.3) between clusters E and W ($\Delta PV_{EW}$ are analyzed, as these are the clusters that deviate the most in terms of the PV streamer position. At forecast initialization time, a relatively large area of positive normalized PV differences is discernible on the stratospheric side of the jet streak over the Gulf of Saint Lawrence (Fig. 7, jet streak is marked as J1 in Fig. 2a). The normalized PV differences in this area are between 0.5 and 1.5 standard deviations and not statistically significant. However, it seems that with increasing lead time this PV difference in the initial conditions moves eastward along the 2-PVU contour, amplifies over the North Atlantic and results in a dipole of significant negative and positive PV differences in the trough over eastern Europe (T) that ultimately result in the shifted PV streamer formation over the Mediterranean (Fig. 8a-d). In the following, this development is discussed in more detail.

The evolution of the normalized PV differences shows similarities with the evolution of the PV error in Baumgart et al. (2018), but here it occurs on a much shorter time scale. The first significant PV difference appears after 18 h (Fig. 8b) over the central

North Atlantic, reaching values above 1.5 standard deviations. A clear cyclonic difference wind field (arrows) is present. Subsequently, during the non-linear development of the Atlantic ridge and the trough over eastern Europe, the PV difference propagates downstream in a wave-like manner giving rise to alternating positive and negative PV differences resulting in a more progressed anticyclonic Rossby wave breaking in cluster W compared to cluster E (Fig. 8a-d). Ultimately, this results in a zonally shifted tip of the narrow trough, and later, the PV streamer (consistent with Fig. 5). This downstream development of the original PV difference can be explained by non-linear (barotropic) Rossby wave dynamics (see Baumgart et al., 2018). However, the initial rapid amplification of the positive normalized PV difference at 1800 UTC 24 Sep is not straightforward to explain. A contribution of warm conveyor belts can be excluded, as no intersection points of warm conveyor belt trajectories with the 325 K isentropic level (light green crosses in Fig. 8a-d) occur in the vicinity of the investigated PV difference.

The synoptic environment of the amplification of this positive normalized PV difference is characterized by a jet streak (Fig. 8e-h). At 0600 UTC 24 Sep 2018 the jet streak is further upstream and much weaker, but at 1800 UTC 24 Sep 2018, when the rapid amplification takes place, the positive PV difference is located on the stratospheric side of the jet streak. Subsequently, the positive PV difference propagates faster downstream than the jet streak maximum. The positive PV differences are associated with a slight shift in the jet streak between clusters E and W at 1800 UTC 24 Sep 2018 and thereafter (solid and dashed white contours). There is some precipitation in the region of the positive PV difference at 1800 UTC 24 Sep 2018, but the values are low and the precipitation area small compared to the later time steps. Furthermore, the positive PV difference is located in a region that is clearly stratospheric. Hence, a contribution of direct diabatic PV modification to the amplification is very unlikely and it seems to be the upper-tropospheric dynamics associated to the jet streak that drives the amplification at 1800 UTC 24 Sep 2018.

Jet streaks are regions with strong ageostrophic winds. In order to understand the effects of the geostrophic and ageostrophic wind components to the amplification of the positive PV difference we qualitatively apply the equation for the local tendency of the PV error following Davies and Didone (2013) and Baumgart et al. (2018):

$$\frac{\partial \mathrm{PV}^*}{\partial t} = -\mathbf{v}^* \cdot \nabla_\theta \mathrm{PV} - \mathbf{v}^* \cdot \nabla_\theta \mathrm{PV}^* - \mathbf{v} \cdot \nabla_\theta \mathrm{PV}^* + \mathrm{DIAB}^* + \mathrm{RES}^* \tag{2}$$

where $\mathrm{PV}^*$ and $\mathbf{v}^*$ denote the error PV and wind field, respectively. The first three terms of Eq. 2 represent the contributions of adiabatic upper-level dynamics to the PV error tendency and the fourth term the contribution of diabatic processes. The last term contains the residual contribution (e.g due to frictional effects, numerical diffusion etc.) and seems to be systematically reducing forecast error (Baumgart et al., 2018). As in our case a substantial direct contribution of diabatic processes to the amplification of $\mathrm{PV}^*$ at 1800 UTC 24 Sep 2018 is very unlikely, we focus on the first three terms of Eq. 2. In our case the *-terms denote differences between clusters E and W, and the first term represents the advection of the PV field in cluster W by the difference wind and the second term the advection of the PV difference field by the difference wind. The second term is likely much smaller as the PV differences are still very small so early in the forecast. The third term is the advection of the PV difference by the wind in cluster W, which advects individual difference features along the flow in cluster W. Note that strictly speaking, Eq. 2 only applies to the PV difference between forecast and analysis (the PV error) or between two forecasts. Here, we are looking at the PV difference of the means of two ensemble clusters, but material PV conservation does not hold

necessarily for average fields. Therefore this equation has to be applied with caution and is used here only for qualitative argumentation. However, it helps to explain the relevant processes responsible for the amplification of the PV difference at 1800 UTC 24 Sep 2018.

The above argumentation implies that a substantial part of the amplification of the PV difference at 1800 UTC 24 Sep 2018 is due to the first and the third term on the r.h.s. of Eq. 2. If the difference wind is additionally separated into a geostrophic component $\mathbf{v_g}^*$ and an ageostrophic component $\mathbf{v_a}^*$ we arrive at the following approximation for the tendency of the not yet normalized PV difference for this specific time step and location:

$$\frac{\partial \mathrm{PV}^*}{\partial t} \approx -\mathbf{v_g}^* \cdot \nabla_\theta \mathrm{PV_W} - \mathbf{v_a}^* \cdot \nabla_\theta \mathrm{PV_W} - \mathbf{v_W} \cdot \nabla_\theta \mathrm{PV}^* \tag{3}$$

where the subscript W denotes the fields in cluster W. Figure 9 shows the normalized PV differences at the beginning and the end of the 6 h time interval during which the amplification takes place. Additionally, the geostrophic and the ageostrophic difference wind fields together with the PV field and the jet streak in cluster W are shown. As discussed above, it is reasonable to assume that, for the given synoptic situation, the positive PV difference between 40°W and 50°W at 1200 UTC 24 Sep (Fig. 9a,c) is advected by the wind field in cluster W (see third term in Eq. 3) and amplified (by the first two terms in Eq. 3), resulting in the positive PV difference 6 h later with a larger amplitude and spatial extent further downstream (Fig. 9b,d). The magnitude of the amplification can be estimated as about 1 standard deviation when using the normalized differences and about 1 PVU when using absolute differences (not shown). The geostrophic difference wind field shows a cyclonic pattern around the positive PV difference (Fig. 9a,b), essentially leading to a retrogressive propagation of the PV difference relative to the flow in cluster W, i.e. counteracting the advection of the PV difference. This is comparable to the effect of the error wind associated with the upper-level PV anomalies described in Baumgart et al. (2018). However, the cyclonic geostrophic difference wind field is not fully symmetric and results in larger PV advection on the upstream side of the PV difference than downstream and, hence, contributes to the amplification of the positive PV difference. Larger in this case is the contribution of the ageostrophic wind (see Fig. 9c,d). At 1200 UTC 24 Sep large ageostrophic difference winds almost perpendicular to PV isolines are present just downstream of the positive PV difference, near the left exit of the emerging jet streak are . Six hours later, ageostrophic difference winds with magnitudes up to $5\,\mathrm{m\,s^{-1}}$ coincide with the region where the positive PV difference is largest, but also with the jet streak and large PV gradients. Also further east large ageostrophic difference winds are present within the emerging negative PV difference. The effect of the ageostrophic wind is contained in the upper-level divergent wind in Baumgart et al. (2018). However, they attributed it mainly to the divergent outflow above a region with intense latent heat release, whereas in this case the divergent wind is strongly related to the ageostrophic circulation in the jet streak.

It can be concluded that the initial amplification of the positive PV difference at 1800 UTC 24 Sep 2018, especially the peak values, can be well explained by differences in the ageostrophic circulation in combination with the strong PV gradients associated with the emerging jet streak.

## 6 Effect of PV streamer position on cyclone formation and evolution

In the previous section we identified the dynamical pathway leading to the uncertainty in the position of the PV streamer at day 3 of the ensemble forecast, and here we investigate how this uncertainty affects the subsequent cyclone formation and precipitation patterns in the Mediterranean. Finally, the occurrence of medicanes in each cluster is diagnosed and links to upper-level PV and low-level $\theta_e$ discussed.

### 6.1 Synoptic development over the Mediterranean

To examine the diverging synoptic development of the three clusters, the evolution of cluster-mean upper-level PV, sea level pressure, and surface precipitation is analyzed. Figure 10 shows the upper-level PV scenarios for clusters W, C and E and how they translate into distinct low-level cyclogenesis scenarios. Prior to the formation of the PV streamer from the narrow trough over eastern Europe (Fig. 10a,e,i), the differences between the three clusters are still small, but significant on the upstream side of the narrow trough (indicated by the teal contours in Fig. 10a,i). This is the same region where the largest negative normalized PV differences occur (Fig. 8c,d). While the 2-PVU contour representing the trough in cluster C is very similar to the analysis (black contour), the contour is slightly shifted to the west compared to the analysis in cluster W, and the trough is too narrow on the upstream side resulting in an eastward displacement of the 2-PVU contour in cluster E at this location. After the narrow PV streamer has formed, the differences between the clusters become more obvious (Fig. 10b,f,j). The shape and position of the PV streamer in cluster C is still very close to the analysis, whereas in cluster W the tip of the streamer is thinner and extends more to the west, and in cluster E it is shifted to the east. In these regions, clusters W and E significantly differ from cluster C. This is not surprising as the clustering was specifically designed to focus on these differences. At 1200 UTC 27 Sep 2018, i.e. the time of cyclogenesis in the analysis, a PV cutoff has formed in all clusters (Figs. 10c,g,k) and the differences in the scenarios over the Mediterranean are very prominent. While in cluster C the cutoff is located south of Italy in the Central Mediterranean (as in the analysis), cluster W exhibits a much weaker cutoff further to the west over Tunisia, and cluster E shows a stronger cutoff shifted to the east. In all clusters the developing surface cyclones are located slightly east of the cutoff (cyclone centres of individual ensemble members are shown in black dots). Hence, in cluster C the cyclones are located close to Benghazi (as in the analysis, indicated by the teal star), in cluster W close to Tripoli, and in cluster C over Crete.

One day later (Fig. 10d,h,l), the cutoff in the analysis has decayed into smaller patches. In cluster C the cutoff has clearly weakened (PV values $< 3\,\mathrm{PVU}$), in cluster W it has fully decayed, and in cluster E it is still very prominent and strong (PV values $> 6\,\mathrm{PVU}$), indicating substantial differences in latent heat release, which is a major factor contribution to the erosion of PV cutoffs (e.g. Portmann et al., 2018). In both clusters C and E, the vertical structure of the system has become more barotropic, i.e. the high upper-level PV and the surface cyclone are vertically aligned.

The surface cyclones in cluster E show a very different behavior than in clusters W and C (see box plots in the individual panels of Fig. 10). First, cyclogenesis occurs earlier and takes place in a pre-existing low pressure area over the Levantine Sea. At 1200 UTC 26 Sep 2018, 9 out of 18 members have a cyclone identified in the Levantine Sea close to Cyprus, whereas in clusters W and C most cyclones form later in the southern part of the Central Mediterranean Sea. Second, cyclones in cluster

E are on average much weaker than in clusters W and C. The pre-existing cyclones over the Levantine Sea deepen slightly
when they interact with the PV cutoff but – with the exception of two cases – weaken again when the system becomes more
barotropic. Cyclones in clusters W and C intensify more strongly and reach their highest intensities later. It can be concluded
that the eastward shift of the PV streamer (cluster E) leads to a non-linear response of the evolution of the surface cyclones,
whereas the westward shift (cluster W) just results in slightly weaker surface cyclones with a similar cyclogenesis and intensity
evolution than in cluster C.

The shifts in the PV streamer position and resulting differences in cyclone evolution also result in different precipitation scenar-
ios. The accumulated cluster-mean precipitation during the three-day period after the formation of the PV streamer (between
1800 UTC 26 Sep 2018 and 0000 UTC 30 Sep 2018) clearly differs between the three clusters (Fig. 11a-c; differences are sta-
tistically significant, see supplementary material). In all clusters the precipitation mainly occurs in the immediate surrounding
of the cyclone tracks (red lines), with maximum values in the Central Mediterranean Sea near 20°W in cluster C and about
500 km further southwest and northeast in clusters W and E, respectively. Cluster E is associated with the smallest precipitation
area, whereas the areas of clusters C and W are of similar size. The precipitation pattern in cluster C matches best with the
short-term forecasts (Fig. 11d), but the amounts are smaller. This shows that the precipitation amounts derived from the short-
term forecasts are substantially higher than in most ensemble members. In fact, the value of the area-averaged accumulated
precipitation in the box from 30 to 40°N and 5 to 30°E derived from the short-term forecasts is close to the $90^{th}$ percentile of
the ensemble forecast. The variability among members within each cluster is large, as seen from accumulated precipitation for
the members with the highest (Fig. 11 e,f,g) and the lowest (Fig. 11 h,i,j) area-averaged precipitation in each cluster. Member
28 (Fig. 11f) shows the best agreement with the precipitation derived from short-term forecasts and also the cyclone track
shows good agreement with the track in the analyses. Note that in all ensemble members the tracks stop at 0000 UTC 30 Sep
at the latest due to limited data availability (see section 2.2), which explains why no ensemble member follows the track in the
analyses after reaching Greece. The members with lowest precipitation in clusters W and C have very short cyclone tracks and
substantially less precipitation than the members with highest precipitation, indicating large variability within these clusters.
In cluster E, there is less variability, as the precipitation maximum for both ensemble members shown is located over eastern
Greece and of similar amplitude.

This section has shown that the uncertainties in the PV streamer's zonal position over the Mediterranean directly resulted
in uncertainties in the location of cyclogenesis and the amount and location of precipitation. The eastward shift of the PV
streamer led to a different type and timing of cyclogenesis and substantially weaker cyclone intensities, whereas the westward
shift mainly shifts the location of cyclogenesis and peak precipitation.

## 6.2 Cyclone thermal structure and link to upper-level PV and low-level equivalent potential temperature

In this section, the occurrence of medicanes in each ensemble member is diagnosed and connected to differences in the evolu-
tion of upper-level PV and low-level $\theta_e$. Medicanes are identified as cyclones with a deep warm core in the CPS (see section
2.2). Subsequently, for each cyclone and timestep of its life-cycle $\theta_e$ between 850 and 950 hPa is averaged within a 250 km
radius (as shown for the analysis in Fig. 3e), hereafter referred to as LLTHE. The motivation to look at this quantity comes

from the analysis in section 3, which showed anomalously high $\theta_e$ values at 900 hPa in the cyclogenesis region immediately before strong convection and large scale ascent occurred, the cyclone rapidly intensified and the upper-level PV cutoff eroded. High $\theta_e$ values at low levels indicate the potential for strong cross-isentropic upward transport associated with latent heating, which is relevant for the erosion of the PV cutoff. In addition, the 50% grid points with the highest PV values on 325 K within a radius of 750 km around the cyclone centre are averaged to account for the positive upper-level PV anomaly, subsequently referred to as ULPV. The larger radius is chosen because the highest values of upper-level PV are usually expected west of the cyclogenesis area. Each cyclone can now be positioned in a LLTHE-ULPV diagram. Figures 12a,b show the geographical location of cyclogenesis in each ensemble member and the position in the LLTHE-ULPV diagram, respectively. The markers provide information about the vertical thermal structure of the cyclones, i.e. all cyclones that once fulfill the medicane criterion have a white centres and the marker size is proportional to the maximum intensity of the upper-level warm core.

Cluster C (blue markers) produces most medicanes (see also Table 1) and the strongest upper-level warm cores. Cyclogenesis occurs mostly at the Lybian coast around Benghazi. In the LLTHE-ULPV diagram, the medicanes in cluster C are characterized by high LLTHE values up to 332 K and ULPV values between 1.5 and 3 PVU. One medicane (with a weak upper-level warm core) and three other cyclones in this cluster exhibit a late cyclogenesis (blue diamonds) over the Central Mediterranean Sea. The non-medicane cyclones are characterized by very low LLTHE and high ULPV values.

Half of the cluster W cyclones (green markers) are medicanes that have a substantially weaker upper-level warm core than medicanes in cluster C. Cyclogenesis is generally shifted to the west, but all medicanes in this cluster occur close to medicanes in cluster C, showing that there is a preferred region for the formation of medicanes in this case. In the LLTHE-ULPV diagram, the medicanes in cluster W are placed similarly to the medicanes in cluster C but with a tendency to lower ULPV values. The non-medicanes in this cluster have either low LLTHE or low ULPV values, with the exception of two members, one of them is member 12 (see below).

Most of the cyclones in cluster E form close to Crete and earlier than in the other clusters. There are only two medicanes in this cluster and the stronger one forms close the medicanes in cluster C. In the LLTHE-ULPV diagram, most members are characterized by low LLTHE values and all members with an early cyclogenesis (squares) by very low ULPV values. The reason for this is simply that the upper-level PV anomaly is still far away at cyclogenesis in these members. Cyclones with later cyclogenesis have higher ULPV values.

Two days after cyclogenesis, the 34 cyclones that are still existing are mostly located in the northern part of the Central Mediterranean Sea (clusters W and C) or the southern part of the Aegean Sea (cluster E) and the positions of the medicanes vary strongly (Fig. 10c). In the LLTHE-ULPV diagram (Fig. 10d) the cyclones cluster into two distinct groups. All medicanes (with one exception), including the one in the operational analysis, are positioned in a group with low ULPV values (well below 2 PVU) and LLTHE values between 320 and 330 K. Overall, they have experienced a reduction in both, ULPV and LLTHE within the first 2 days after cyclogenesis (with very few exceptions). This is a strong indication, that high LLTHE values favored substantial latent heat release and cross-isentropic upward transport that led to the erosion of the upper-level PV anomaly, i.e. a reduction of ULPV similar to what was observed in the operational analysis (see section 3). This is also consistent with the cluster mean evolution shown in Fig. 10. The second group contains only members of cluster E and is characterized by low

LLTHE and high ULPV (between 3 and 6 PVU). This indicates that latent heating did not reach high enough to erode the PV

cutoff, which can be connected to the low LLTHE values. These low values can be understood because cyclogenesis occurred closer to the northern coast of the Mediterranean and as a result, the low-level air parcels were less exposed to the sea surface in order to be moistened strongly by latent heat fluxes, as it occurred in the operational analysis (see supplementary material). Hence, the erosion of the upper-level PV anomaly appears as a necessary ingredient for the formation of a medicane in this case. For most medicanes this erosion can be connected to high low-level $\theta_e$ at cyclogenesis.

There are a few members, where the cyclones do not follow this storyline. Some of them are now briefly discussed. In member 10, which belongs to cluster C, the cyclone follows the pattern observed for most cyclones in cluster E. This is explained by the fact that member 10 is a borderline member of cluster C (as mentioned in section 4, visible as the easternmost blue member in the histogram in Fig. 5). In fact, the borderline member in cluster E (westernmost red member in Fig. 5) is member 43, which is also a special case in Fig. 12. Another special case is member 12, which appears in a region in the LLTHE-ULPV diagram

that is favorable for medicanes, but does not develop a medicane. A detailed investigation of the upper-level PV evolution indicates, that the PV cutoff is rapidly advected away from the cyclone towards the southwest immediately after cyclogenesis, indicating high vertical wind shear. Low vertical wind shear has already been identified as a synoptic condition of medicane formation (Tous and Romero, 2013). On the contrary, the cyclone in member 28 forms at low LLTHE values compared to other medicane members, but nevertheless the ULPV is eroded and the cyclone develops into a medicane. The upper-level warm

core becomes about as strong as in the analysis and the medicane shows a similar track and precipitation pattern (member with most precipitation in cluster C, see Fig. 11f). The maximum intensity of the surface cyclone however remains 5 hPa above the value in the analysis. In this special case, the reduction in ULPV is not connected to a reduction of high LLTHE values. The last special case discussed here is member 14. In this member, the cyclone in general follows the pattern in cluster E but becomes a medicane, albeit with a very weak upper-level warm core. At the time of the formation of the upper-level warm core, the PV

cutoff exhibits a ring-like shape, with PV values around 2 PVU above the cyclone centre and much higher PV values around it (not shown). Hence, it is possible that the PV cutoff was only substantially eroded in the centre, enough for the formation of a weak upper-level warm core. These special cases illustrate the variety of systems that can emerge in this specific synoptic situation, the complexity of the cyclone dynamics and the limits of the LLTHE-ULPV framework to understand medicane formation.

It can be concluded that the uncertain position of the PV streamer is a major factor that determines the uncertainty in the cyclone formation and evolution in this case and that the central PV streamer position provides the most favorable synoptic environment for a medicane to form. However, storm-internal processes like convection, which is parametrized in the IFS model, and the detailed interaction between upper and lower levels have not been discussed in this study. They may further limit predictability of the cyclone and its meso-scale characteristics, including surface winds and precipitation.

**Table 1.** Number of medicanes in each cluster (bold font).

|  | cluster W | cluster C | cluster E |
|---|---|---|---|
| **# medicanes** | **6** | **15** | **2** |
| out of | 12 | 19 | 18 |

## 7  Conclusions

The basis of this study was an ECMWF operational ensemble forecast that showed large uncertainties in the position of a PV streamer over the Mediterranean and the subsequent development of Medicane Zorbas in September 2018. The ensemble members were clustered into three distinct scenarios according to the position of the PV streamer at day 3 of the forecast. The differences between these scenarios were used to provide new insight into, on the one hand, the origin, amplification and propagation of forecast uncertainties in Rossby waves, and, on the other hand, the link between uncertainties in the large-scale flow and the potential formation of a medicane.

The uncertainties in the position of the PV streamer after 72 h forecast lead time could be clearly linked to the amplification of uncertainties in a jet streak over the North Atlantic in the first 18 h of the forecast. Uncertainties in the ageostrophic circulation combined with the presence of large isentropic PV gradients associated with the jet streak contributed strongly to this amplification. To some extent, the uncertainty could be traced back to relatively large-scale uncertainties in the initial conditions on the stratospheric side of an upper-level jet streak over the Gulf of Saint Lawrence. They were advected along the dynamical tropopause before they amplified in the jet streak over the North Atlantic, propagated into the Mediterranean, eventually resulting in the uncertain position of the PV streamer and, as a result, the uncertain development of Zorbas. The contributions of diabatic airstreams, such as warm conveyor belts, or the direct diabatic PV modification were negligible for the uncertainty amplification in this case. The described amplification due to the ageostrophic circulation in a jet streak could be also relevant in other cases for the initial uncertainty growth that can then further amplify and propagate downstream, especially in strom track regions. Further case studies as well as more climatological analyses are needed to quantify the relevance of this process.

It was shown that the uncertain position of the PV streamer was a major factor limiting the predictability of Medicane Zorbas for both its location and its vertical thermal structure. An eastward shift of the PV streamer in the ensemble forecast led to a non-linear response in the surface cyclone evolution. This is particularly interesting as a comprehensive analysis of the predictability of PV streamers over the North Atlantic and the Mediterranean in the ECMWF ensemble forecasts showed that there is a tendency for eastward displacement in the forecasts, compared to the analysis (Wiegand and Knippertz, 2014). The central (i.e. correct) position of the PV streamer provided the best synoptic conditions for the formation of a strong medicane. These conditions were characterized by high values of low-level equivalent potential temperature and high upper-level PV in the cyclogenesis region. The subsequent rapid erosion of the high upper-level PV was identified as a necessary condition for medicane formation in this case. However, it has to be kept in mind that other potentially relevant factors were not analyzed, such as vertical wind shear and mid-tropospheric humidity (as identified by Tous and Romero, 2013), or the details of the con-

vective processes. In a subsequent study we plan to investigate how these factors limit the predictability of Zorbas' life-cycle after cyclogenesis.

Since the seminal work of Lorenz (1969), the growth of very small uncertainties on convective scales to large-scale uncertainties, so-called upscale error growth, has been discussed as a process limiting atmospheric predictability (e.g. Zhang et al., 2007). However, recent studies suggested that the practical limits of atmospheric predictability often come from uncertainties on much larger scales, even if they are very small compared to the average kinetic energy on that scale (e.g. Durran and Weyn, 2016). This study provides an illustrative example that large-scale initial uncertainties in an upper-level jet streak over North

America, which are of small amplitude relative to the background kinetic energy, can influence the forecast uncertainty of a very intense cyclone in the Mediterranean.

The results and approach of this study are relevant for understanding uncertainties in high-impact weather in the Mediterranean region beyond the relatively rare phenomenon of medicanes. For example, PV streamers are known to be typically connected to Alpine lee cyclogenesis and heavy precipitation on the Alpine south side (Tafferner, 1990; Massacand et al., 1998), and it has

been shown that the predictability of such heavy precipitation events is highly sensitive to the substructure of the PV streamer (Fehlmann et al., 2000; Fehlmann and Quadri, 2000), which is well in agreement with the results of this study. Studying the sensitivity of high-impact weather to the structure and position of PV streamers gains additional relevance considering that the ECMWF ensemble prediction system is strongly underdispersive for PV streamer situations over the North Atlantic and the Mediterranean (Wiegand and Knippertz, 2014). Further research could therefore investigate how uncertain PV streamer

positions come about and affect the predictability for other high-impact weather events.

*Data availability.*  All data is available from the authors upon request

*Author contributions.*  JJGA sparked this work by discovering the uncertain forecast of Zorbas. RP and JJGA designed the basic idea for the study. JJGA carried out the analyses that required the computation of the CPS and RP all remaining analyses. MS and HW helped with the data access and handling and gave important guidance and useful inputs during the whole project. RP and JJGA prepared the manuscript and

all authors gave critical feedback that helped to improve the article.

*Competing interests.*  The authors declare that they have no conflict of interest.

*Acknowledgements.*  We thank the reviewers Ron McTaggart-Cowan, Florian Pantillon, and one anonymous reviewer for their detailed feedback that helped to improve the manuscript. RP acknowledges funding from the ETH research grant ETH-0716-2. JJGA has been funded through the PhD-grants BES-2014-067905, EEBB-I-18-12841 for short research stays, and by grant CGL2017-89583-R by the Spanish Min-

istry of Science, Innovation and University, and co-funded by the European Social and Regional Development Funds. RP thanks Emmanouil Flaounas for insightful discussions about medicanes.

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

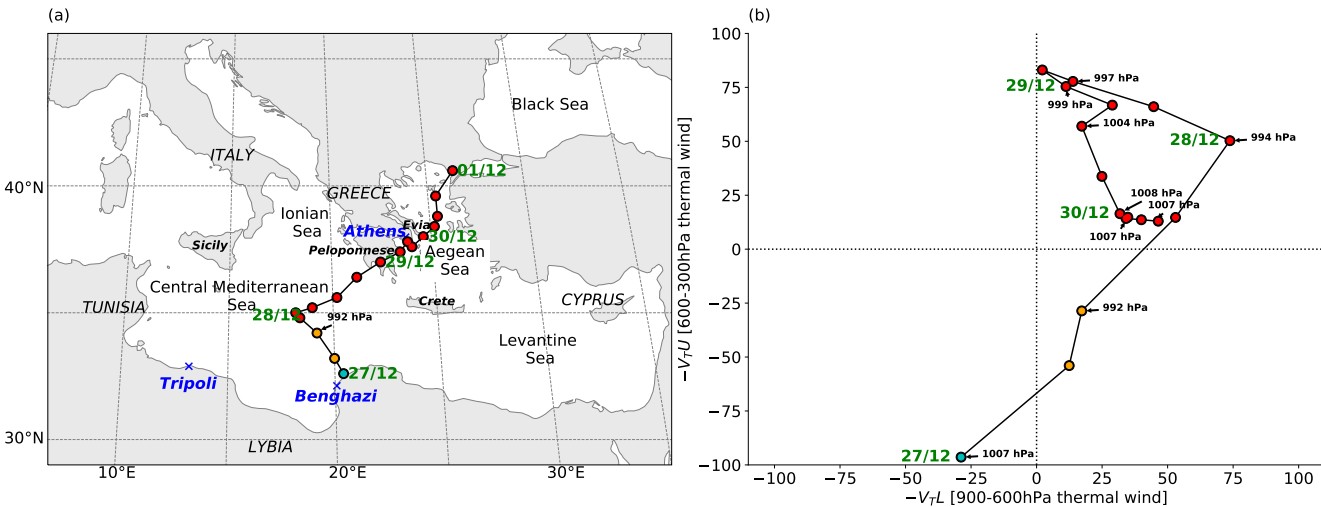

**Figure 1.** (a) Track of Medicane Zorbas (circles and black line) and (b) cyclone phase space diagram derived from the ECMWF operational analyses. Cyclone positions are colored according to the quadrant in the CPS diagram (blue: cold core, orange: shallow warm core, red: deep warm core). Black numbers indicate the minimum sea level pressure (hPa) of the cyclone at this particular time of its life cycle and green numbers the day (in Sep 2018) and time (hours UTC) .

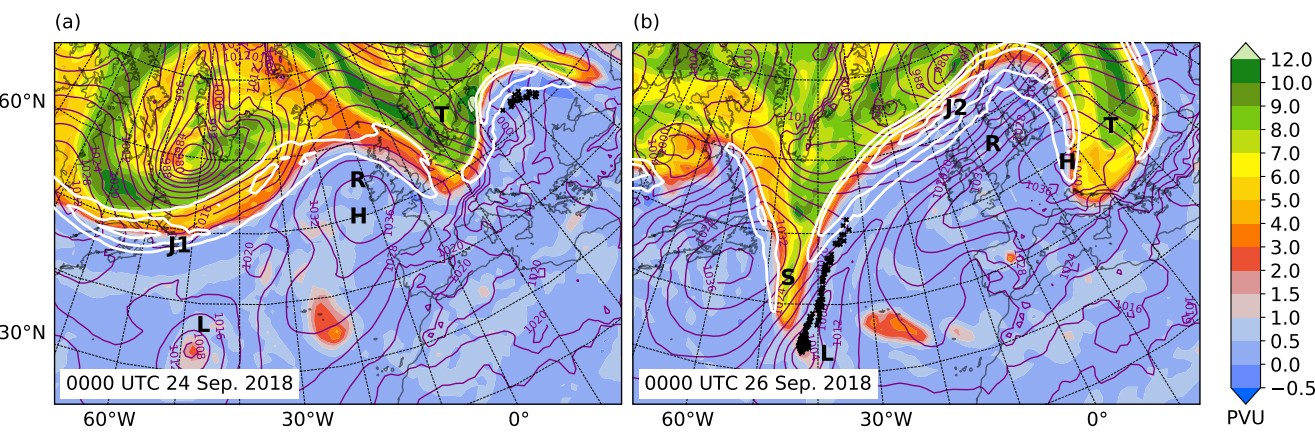

**Figure 2.** Synoptic situation over the Euro-Atlantic sector before the formation of the PV streamer over the Mediterranean. PV (shaded, in PVU) and wind speed (white contours, in m s$^{-1}$ on 325 K, intersection points of warm conveyor belts (ascent rate of more than 600 hPa in 48 h) with the 325 K isentrope (black crosses), and sea level pressure (purple contours, in hPa) at (a) 0000 UTC 24 Sep and (b) 0000 UTC 26 Sep 2018.

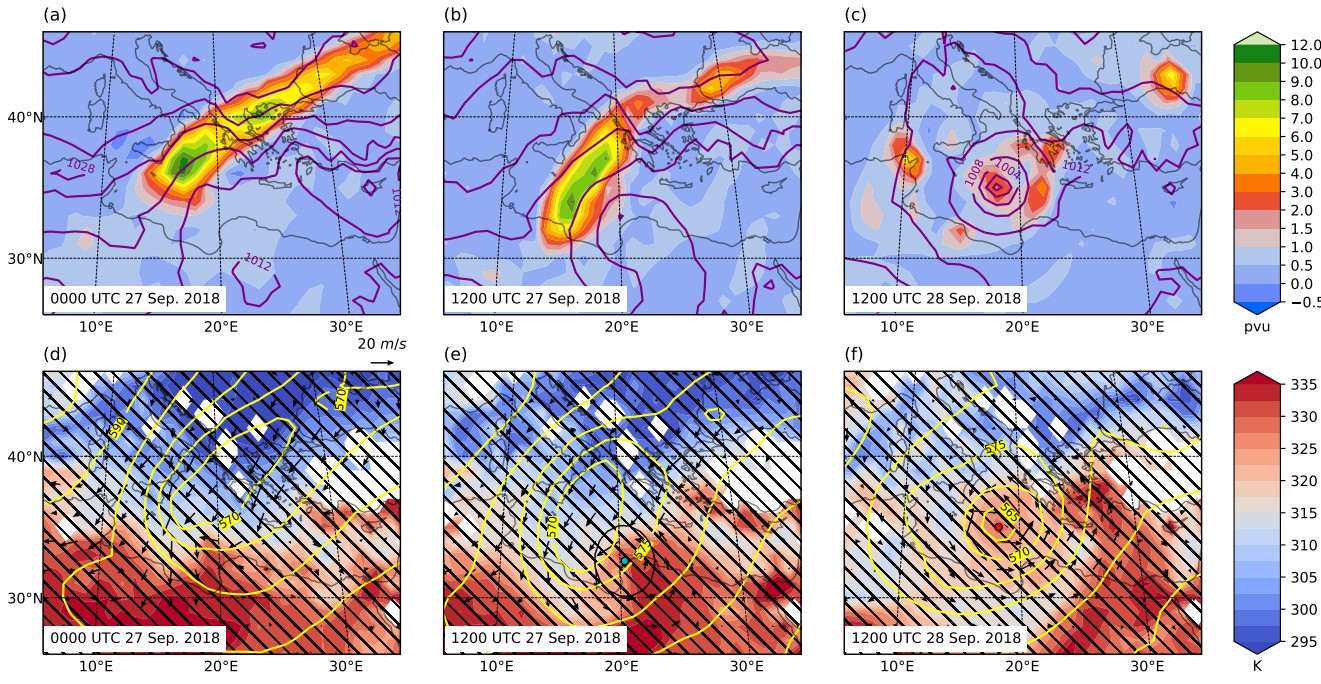

**Figure 3.** Synoptic situation over the Mediterranean after the formation of the PV streamer. (a-c) PV on 325 K (shaded, in PVU; this level corresponds approximately to the 300-350 hPa pressure levels in this region) and sea level pressure (purple contours, in hPa) and (d-f) equivalent potential temperature ($\theta_e$, shaded, in K) and wind vectors (black arrows) on 900 hPa, and geopotential height on 500 hPa at (a,d) 0000 UTC 27 Sep, (b,e) 1200 UTC 27 Sep, (c,f) 1200 UTC 28 Sep 2018. The hatched areas in (d-f) show regions where $\theta_e$ on 900 hPa is anomalously high (at least one standard deviation larger than climatology) with respect to the Sep-Oct ERA-Interim climatology for the period 1979 – 2017. The 325 K level

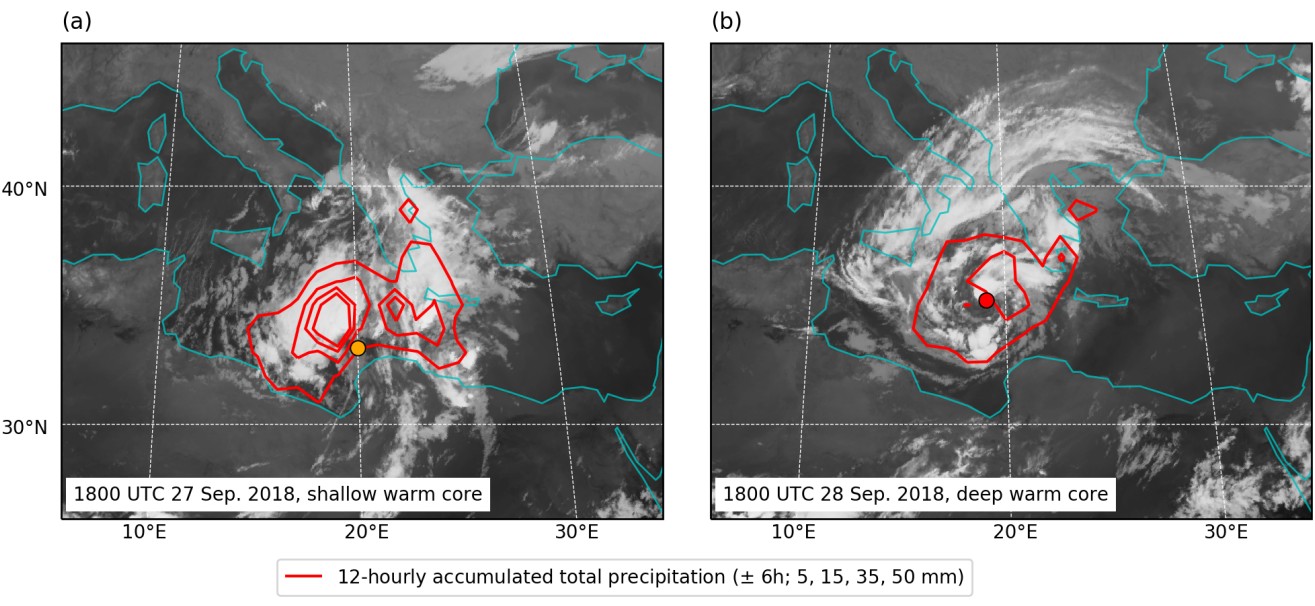

**Figure 4.** Infrared channel 9 (10.8 μm) of MSG SEVIRI provided by EUMETSAT (grey shading) and total precipitation accumulated over the previous and following 6 hours based on ECMWF operational short-term forecasts (red contours, 5, 20, 35 and 50 mm) at (a) 1800 UTC 27 Sep and (b) 1800 UTC 28 Sep. Cyclone positions are marked with circles and colored according to the thermal structure of the cyclone (as in Fig. 1).

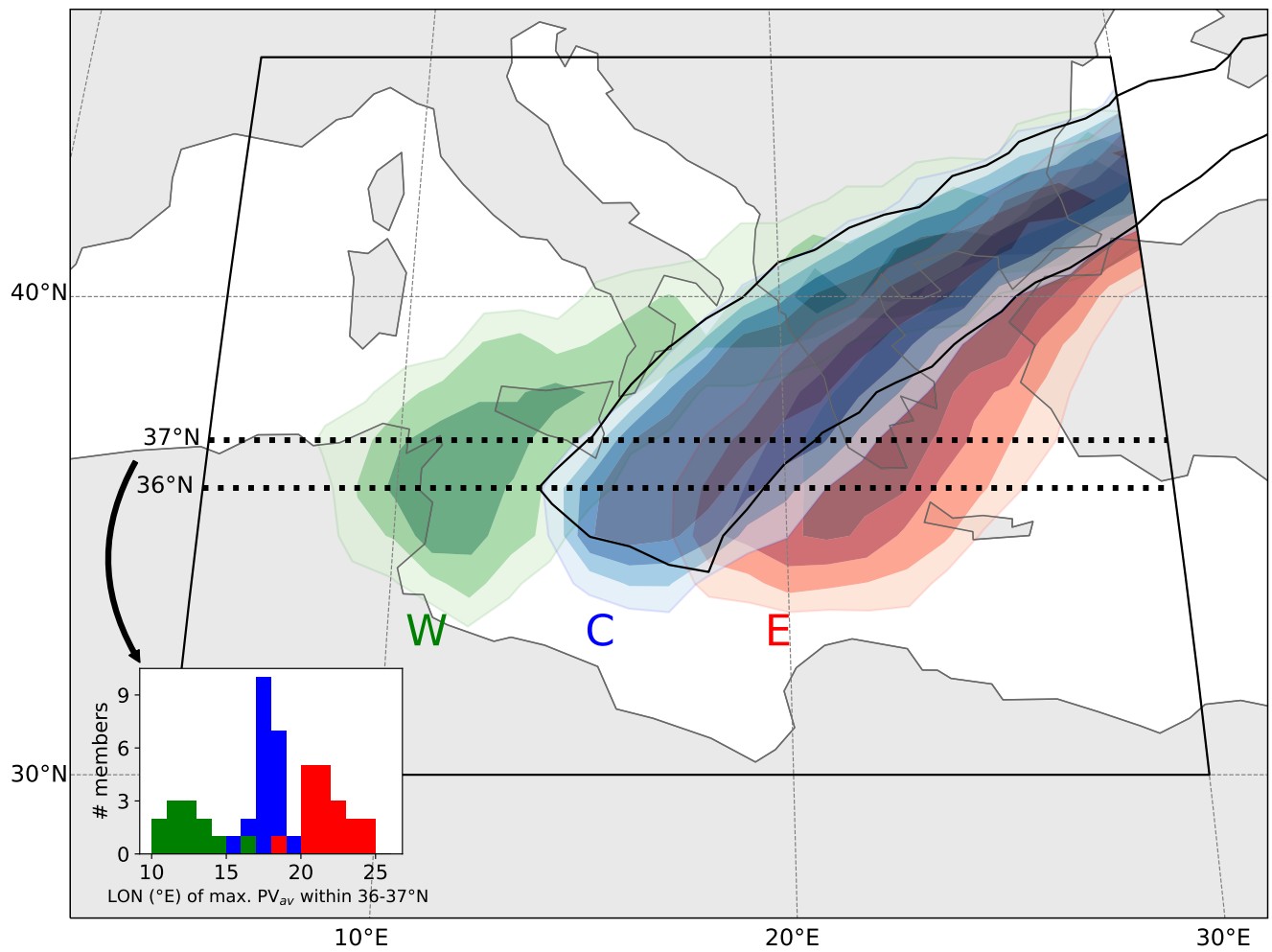

**Figure 5.** Clustering of ensemble members (initialized at 0000 UTC 24 Sep 2018) according to the position of the upper-level PV streamer in the Mediterranean at 0000 UTC 27 Sep 2018. Colors show frequencies of $PV_{av} \geq 2$ PVU (shading, every 20%) for each cluster (blue: cluster C, green: cluster W, red: cluster E) and the black line the $PV_{av} = 2$-PVU contour in the operational analysis. The region considered for the clustering is shown by the black box (see text for details).

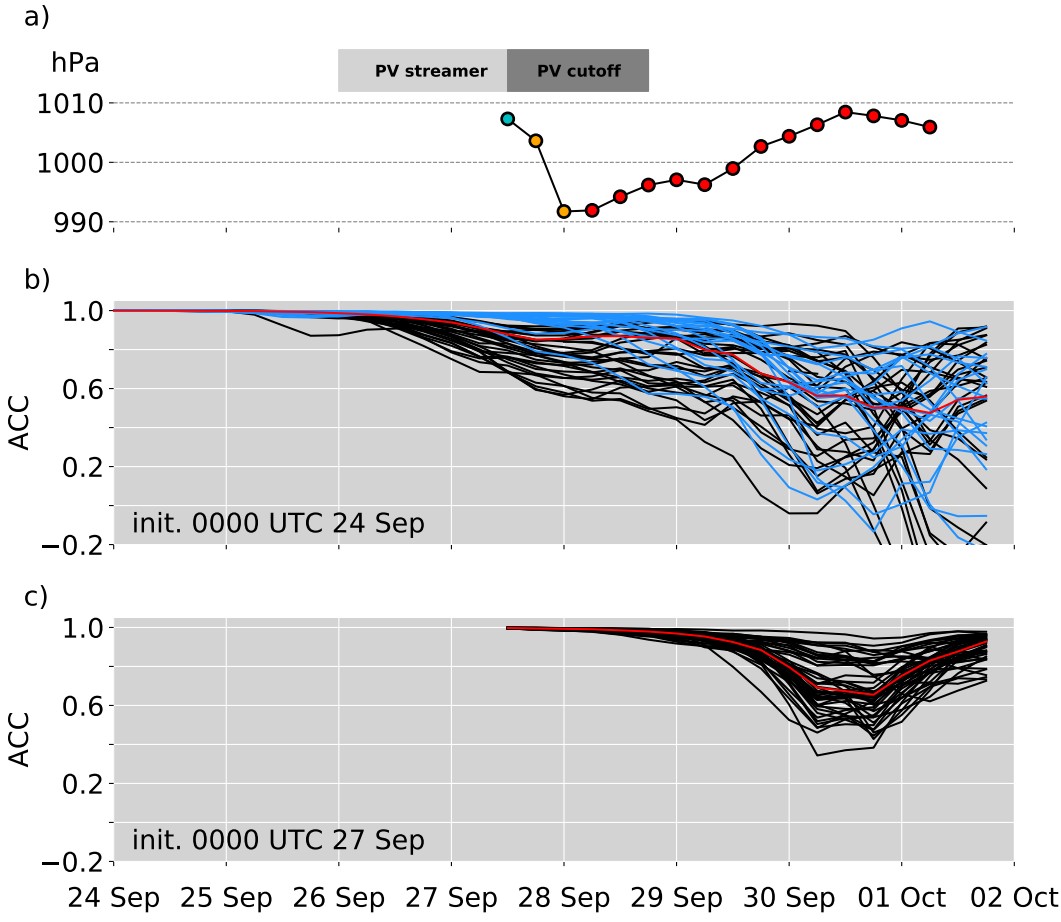

**Figure 6.** (a) Temporal evolution of synoptic elements discussed in this study. Grey boxes indicate times when the PV streamer or cutoff on 325 K is present in the analysis and the solid line shows the evolution of the minimum sea level pressure of the cyclone, colors indicate the cyclone stage as identified from the CPS (cold-core: blue, shallow warm-core: orange, deep warm-core: red). (b) and (c) Temporal evolution of the anomaly correlation coefficient of geopotential height at 500 hPa in the Mediterranean box (see Fig. 5) for each ensemble member (black lines for clusters W and E, blue lines for cluster C) and the median (red line) of the ensemble forecasts initialized at (b) 0000 UTC 24 Sep 2018 and (c) 0000 UTC 27 Sep 2018.

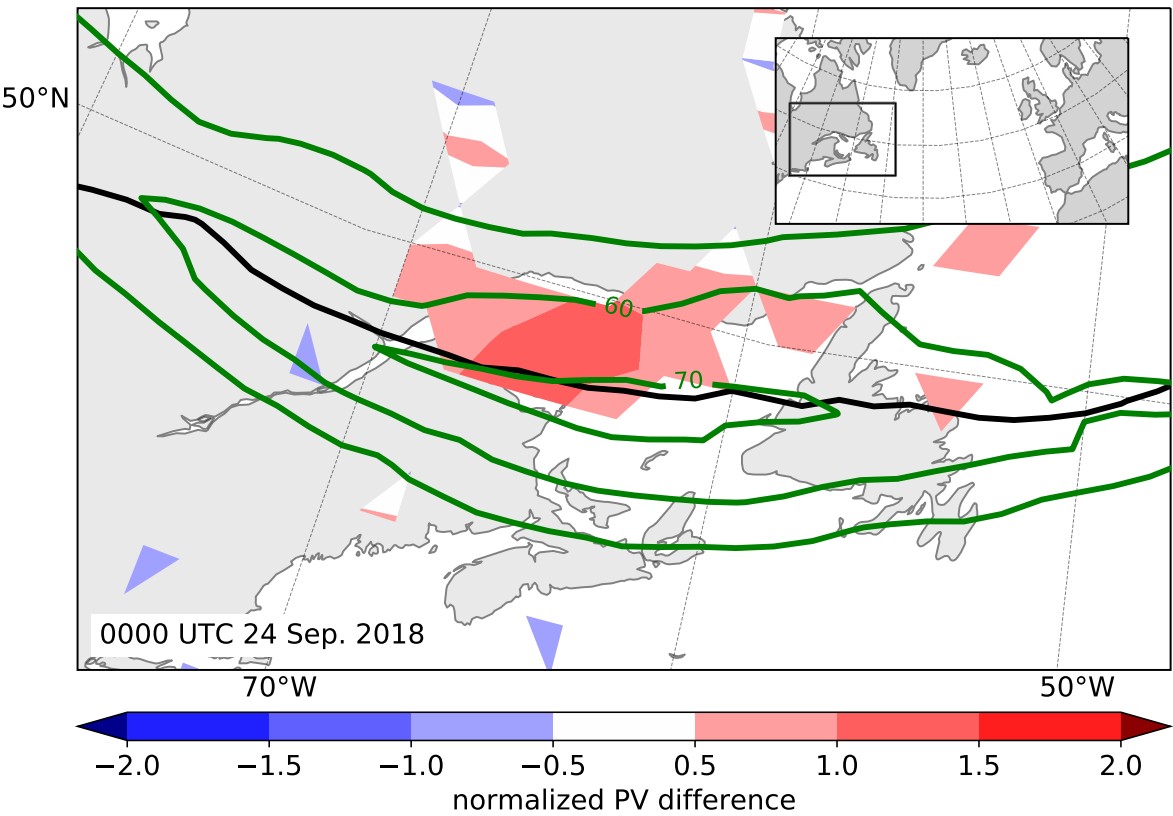

**Figure 7.** Normalized differences of PV on 325 K betweens cluster E and W (shaded), 2-PVU contour (black) and wind speed (green contours for 50, 60, and 70 m s$^{-1}$) on 325 K derived from the operational analysis centered over the Gulf of Saint Lawrence at 0000 UTC 24 Sep 2018, i.e. at the initialization time of the ensemble forecast. Normalized PV differences are not significant on the $\alpha_{\text{fdr}} < 0.1$ level.

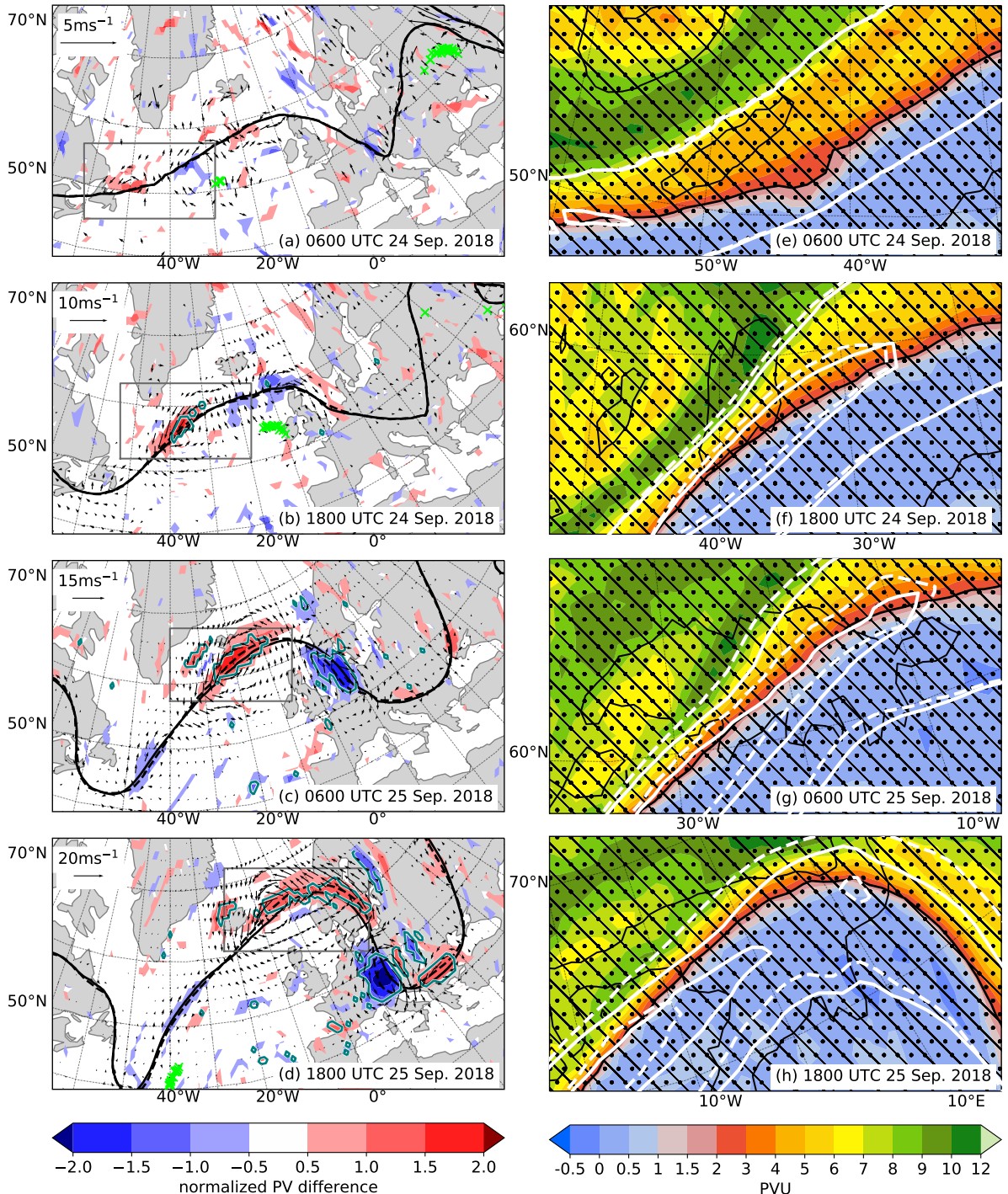

**Figure 8.** (a-d) Normalized PV differences between clusters E and W (shaded), full difference winds (arrows, only if larger than $1 \, \text{m s}^{-1}$, reference vectors is in top left of panels), 2-PVU contour (black lines ) of clusters E (solid) and W (dashed), and intersection points of warm conveyor belt air parcels on 325 K (green crosses) from 0600 UTC 24 Sep to 1800 UTC 25 Sep 2018, every 12 hours. Regions with statistically significant PV differences ($\alpha_{\text{fdr}} < 0.1$) are marked with **30**al contours. (e-h) PV (shaded, in PVU) and 2-PVU contour (black) on 325 K and 6-hourly accumulated total precipitation (2-10 mm in hatches, >10 mm in stippling) from the operational analysis at the same times as (a-d). Additionally, average wind speeds on 325 K (white contours, 50 and 70 $\text{m s}^{-1}$) are shown for clusters E (solid) and W (dashed).

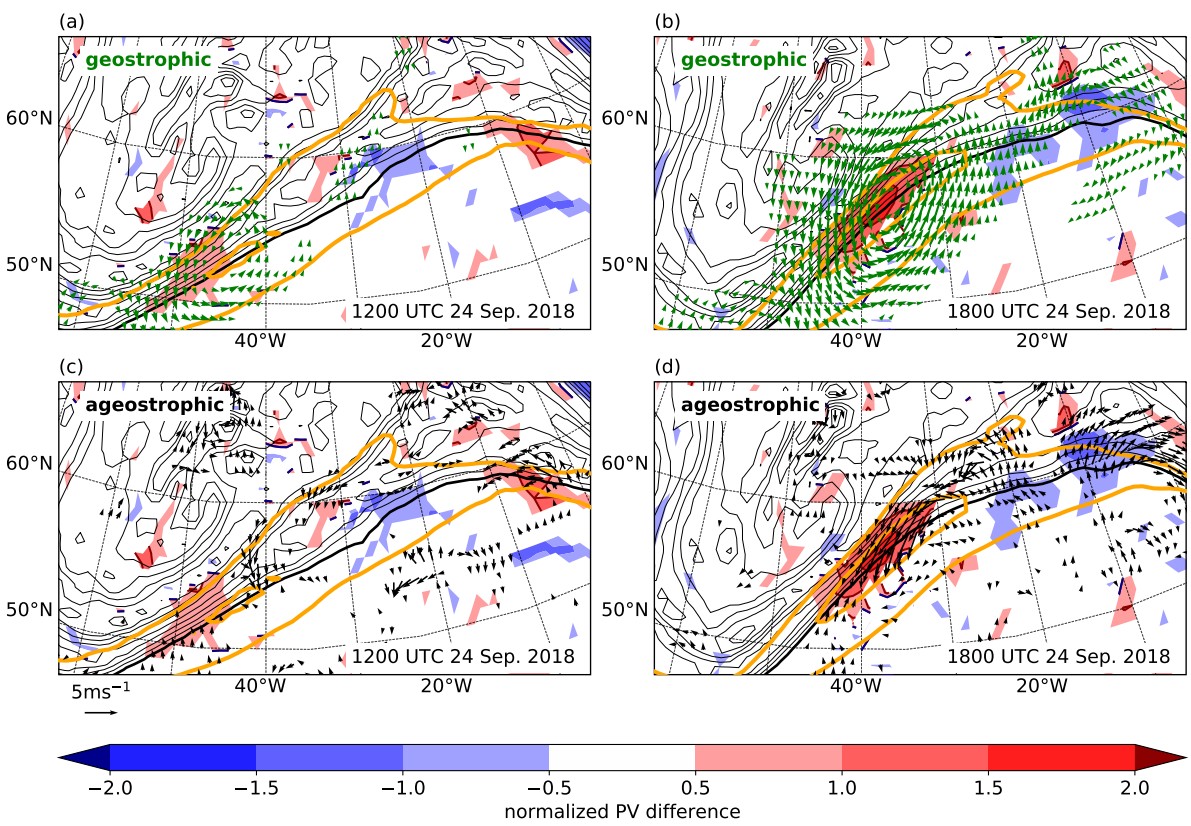

**Figure 9.** Normalized differences of PV on 325 K between clusters E and W (shaded, as in Figs. 7 and 8) and the contributions of (a,b) the geostrophic (green arrows) and (c,d) the agestrophic wind (black arrows) to the difference winds on 250 hPa (which corresponds approximately to the pressure level of the 325 K isentrope at this location) at (a,c) 1200 UTC 24 Sep 2018 and (b,d) 1800 UTC 25 Sep 2018. Additionally, wind speeds on 325 K (orange contours, 50 and 70 m s$^{-1}$) derived from the operational analysis are shown in each panel.

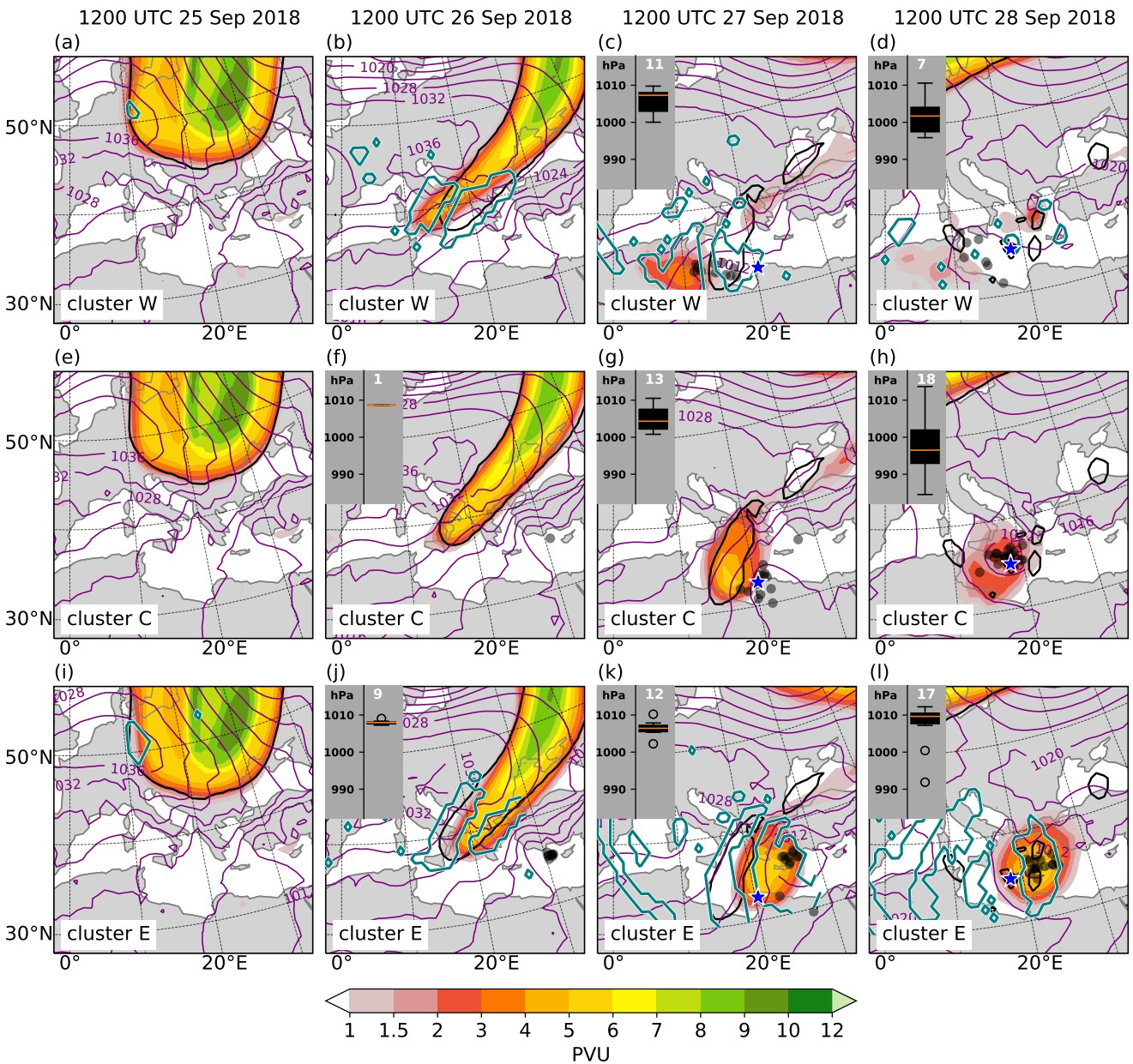

**Figure 10.** Cluster-mean PV on 325 K (shaded, in PVU), cluster-mean sea level pressure (purple contours, every 4 hPa), analysis 2-PVU contour on 325 K (black), cyclone positions (as identified with the method described in section 2.2) in each ensemble member (black dots), and in the operational analysis (blue star) for clusters W, C, and E (panels a-d, e-h, and i-l)) from 1200 UTC 25 Sep to 1200 UTC 28 Sep 2018 every 24 h. Insets at the top left of the panels show box plots of minimum sea level pressure of the cyclones in each cluster and white numerals indicate the number of cyclones. Additionally, at 1200 UTC 25 Sep 2018, regions where the differences to cluster C of the PV field on 325 K are statistically significant on the $\alpha_{\mathrm{fdr}}$=0.1 level are shown for clusters W and E as teal contours in (a-d, i-l).

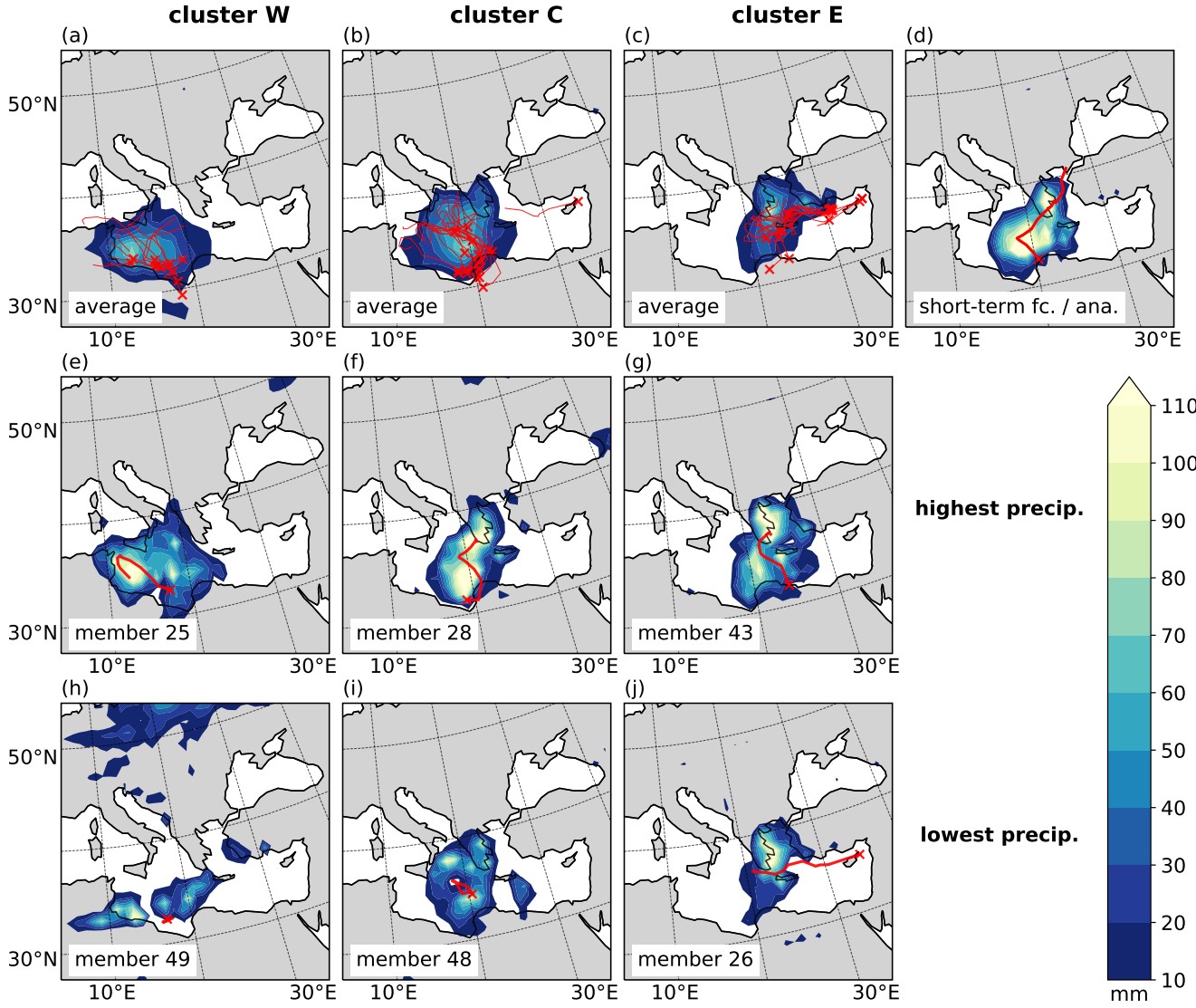

**Figure 11.** Accumulated precipitation (shading, in mm) from 1800 UTC 26 Sep to 0000 UTC 30 Sep 2018 for (a-c) cluster means of clusters W, C, and E; (d) the short-term forecasts; (e-g) the members in each cluster with the highest and (h-j) the lowest area-averaged accumulated precipitation (average over 30-40°N and 5-30°E). Additionally, red crosses indicate cyclogenesis and red lines cyclone tracks for (a-c) each member within the cluster, (d) the operational analysis and (e-j) the corresponding ensemble member.

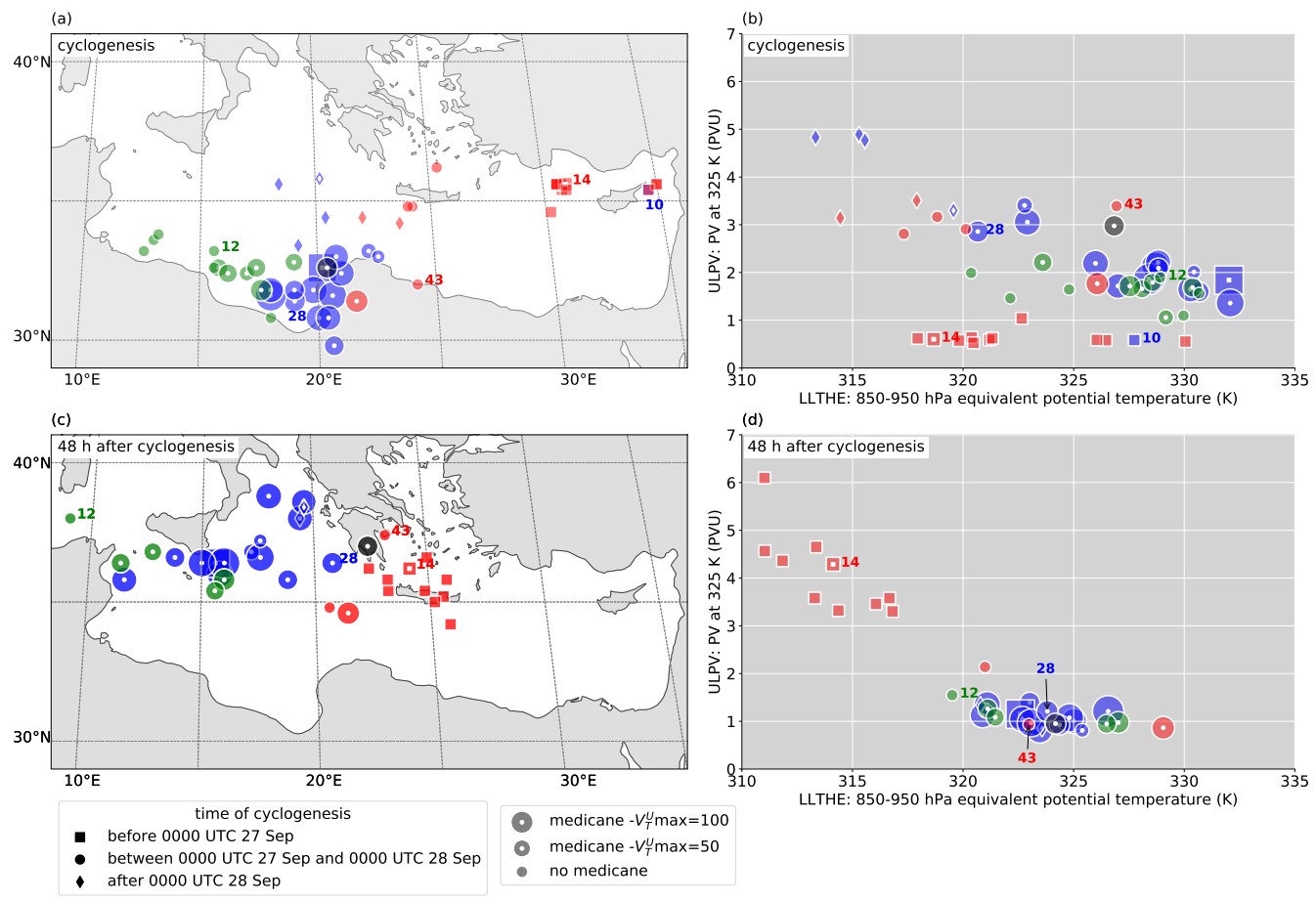

**Figure 12.** (a,c) Geographical maps of cyclone positions and (b,d) LLTHE-ULPV diagram for cyclogenesis (upper panels) and 48 h after cyclogenesis (lower panels). Marker shape corresponds to the time of cyclogenesis (see legend below panel c), colors to the cluster (W: green, C: blue, E: red, analysis: black), and, for medicanes (markers with white centre), the size to the maximum intensity of the upper-level warm core.