# Peer review of "How an uncertain short-wave perturbation on the North Atlantic wave guide affects the forecast of an intense Mediterranean cyclone (Medicane Zorbas)"

_Weather and Climate Dynamics, 2019_

## Referee Comment (RC1) · Ron McTaggart-Cowan (Referee) · 1 Sep 2019

Review of "Medicane Zorbas: origin and impact of an uncertain potential vorticity streamer" by Portmann et al.

**Background**

The authors investigate the predictions of a September 2018 medicane in the ECMWF ensemble system. They identify ensemble members that have differing day-3 representations of a PV streamer involved in the storm's development. They track these differences back to small differences in the initial conditions and show the progression of PV spread in association with an anticyclonic Rossby wave break. The remainder of the analysis focuses on the interaction between the PV streamer and the developing cyclone: distinct precipitation and storm structures are identified in the different sub-ensembles.

Reductions in predictive skill associated with the development of sub-synoptic systems in the Mediterranean region are an important subject for investigation. Similarly, the limits of predictability imposed by PV streamer evolution and interactions between such features and lower-tropospheric circulations are not well understood. Despite these interesting fundamental underpinnings, the current submission suffers from a large number of flaws in organization, preparation and analysis. Although each of these may not be considered fatal in isolation, a significant amount of effort will be required to address all of them thoroughly. Any revised submission of this investigation will necessarily be heavily modified and constitute a new piece of work. I hope that the authors will find the comments below useful as they pursue this research.

Reviewer: Ron McTaggart-Cowan

**General Comments**

1. The manuscript lacks organization and logical flow. This extends from the highest level of structure (sections and subsections), down to the paragraph and even sentence level. It makes the manuscript difficult to read and follow because concepts and details are disjointed, scattered and frequently repeated throughout the text. Ordinarily, I would consider these kinds of organizational issues relatively minor and possibly within the domain of the authors' discretion; however, in this case they seriously detract from the work and make it very difficult for readers to follow the investigation. Addressing there problems will involve the rewriting of much of the text, but will yield a more focused manuscript that will likely be shortened by at least 1-2 pages.
    a. High Level
        i. The structure of the introduction is ineffective. It begins with a very cursory review of tropical transition and medicanes (including PV streamers), then switches to a Rossby wave discussion that returns to error amplification twice, and then goes back to a very short summary of Zorbas. The latter also includes thesis questions and an outline of the remainder of the study that lacks section information or internal references.

ii. I do not think that the decision to replace the standard "Data and Methods" section with "Operational ECMWF Products" is effective. It means that additional data sources (satellite imagery, lightning detection, etc) have to be described in the text body, descriptions that seriously distract from the analysis when they occur. The same is true of methodological details (e.g. LAGRANTO, which is introduced twice) and the entire discussion of tracking and the CPS (L378-L392). Descriptions of all of these sources and techniques should be centralized in a "Data and Methods" section.

iii. There are numerous forward-references to section 6 throughout the earlier sections of the manuscript. While appropriate internal referencing is a useful tool, these repeated forward-references are likely an indication of poor manuscript organization, especially when they underpin important elements of the analysis (for example, the CPS referenced in sections 3 and 4 but never shown despite a reference to "Fig. 4a", which does not exist in the submission; L159). I think that the synoptic analysis (ideally shortened from its current length by enhanced focus) should include a discussion of the medicane itself, including the CPS. I understand that the medicane is not the intended focus of the work as is repeatedly stated in the text; however, the reader could be excused for thinking that it is because of (1) the title, (2) the multiple introduction subsections that deal with TC-like features, (3) the statement on L317 that the investigation of the "development of a medicane-like system" is an objective of section 6.2, and (4) the pervasive forward-referencing to the CPS analysis in the text.

b. Medium Level

i. Each section should begin with an introduction of the section contents so that the reader has an idea of how the section fits into the larger narrative of the work. A section should not begin with a description of data used in specific figures, as does section 3. Please revise each section introduction to ensure that the reader is logically guided through the work.

ii. Each paragraph should begin with an introduction to the paragraph contents, and should conclude with a statement that relates back to the material introduced. There are very few paragraphs in the manuscript that follow this basic structure. A particularly clear example of a paragraph that ranges far too broadly occurs on L50-64. The paragraph (and subsection) starts with a description of initial condition uncertainty, moves into ensembles more generally, then into PV error growth, and ends with a discussion of tropical cyclones. This lack of focus makes the study very difficult to follow despite the fact that the investigation itself is relatively straight-forward. In this case, the subject of error growth reappears

in the paragraph starting on L73, which further adds to the confusion of the discussion. Please do not simply rewrite the paragraphs noted in this comment: the structure of virtually all paragraphs in the manuscript needs to be reconsidered and revised, an effort that will lead to rewriting of large portions of the work. The readability problems induced by the lack of logical internal paragraph are more than aesthetic in this case, and are serious enough that they significantly reduce the potential impact of this work.

    iii. I do not think that summary paragraphs at the end of a section are useful, particularly given the large number of relatively short sections in this submission. For example, the summary on L163-L167 is redundant with analysis undertaken in the previous page of the manuscript. Please remove summary paragraphs (they also appear at the ends of sections 5 and 6.1) in favour of making the text itself direct, clear and readable (see item 1.c.ii below).

  c. Low Level

    i. Reference to caption-level figure details within the text is highly distracting. For example, the fact that precipitation is shown in red solid contours in Fig. 1 is referenced three times in section 3 (once erroneously as "blue contours"; L130), while the fact that QG vertical motion is shown in red contours is referenced twice in the same section. These plotting details are described in the caption, and their appearance in the text detracts from the flow of the analysis. Figure and panel references should be enough to guide readers through the discussion. Please consider removing caption information from the text body throughout the submission.

    ii. The writing style is too informal and lacks the precision required for scientific text. For example, the outline of the manuscript is described as "a journey" on L101, the analyses in section 6.2 "hint" at airstreams (L321), and the first person plural ("we") is used heavily throughout the submission. The introduction of section 6.2 (L320-324) can be summarized as, "we don't do anything thoroughly here, but we show stuff that's different and make some guesses about what that means; then in the next section we do it properly". I don't think that that kind of introduction (or approach to the analysis) will make readers want to continue to invest their time in the rest of the section. Throughout the submission, irrelevant details clutter the text (e.g. does it matter that 1800 UTC 26 September is "in the evening of the same day" on L126?), and ill-defined concepts reappear throughout the analysis (e.g. the "C-shaped" PV cutoff with a "dent" and "dent structure" on L138, 141 and 146, respectively). Every effort should be made to make the text succinct and readable, so that the analysis does not get lost in superfluous details and unnecessary bridging statements.

2. I think that cyclogenesis in cluster 3 is really interesting, but that the discussion in the current study misses the opportunity to capitalize on its uniqueness (I do not think that section 6.3 is sufficient in this respect). It looks to me like this is an excellent example of a nonlinear response / bifurcation leading to a real limit on predictability. Clusters 1 and 2 are simply phase shifts of the same cyclogenesis event. From a guidance perspective, both are reasonably useful at least in terms of situational awareness. Cluster 3, however, looks to me like the development of a different cyclone. There's an 850 hPa circulation south of Turkey in all of the groups at 1200 UTC 26 September (Fig. 8, column 2). In fact, a cyclone has already formed in this region in many of the cluster 3 members and one of the cluster 1 members (also shown in Fig. 9a). In groups 1 and 2, the low between Crete and Cyprus disappears as the PV tail promotes development along the African coast. In group 3, the pre-existing cyclone intensifies and fractures the PV streamer as the low retrogresses towards Crete (Fig. 8, column 3). By 1200 UTC 28 September, the medicane lies in the central Mediterranean in groups 1 and 2, but it is a completely different storm that is centered on Crete in group 3 (this differs from the interpretation implied by discussions on L286-L290 and L303-304 of the submission). So the relatively small difference in the location of the PV streamer axis (a linear change from west to east of the observed location) leads to a highly nonlinear response in the form of development of a new cyclone (groups 1 and 2) or intensification of an existing circulation (group 3). (Note that a couple of centers form over northern Africa in group 3 at 1200 UTC 27 Sept – Fig. 8k; these are cases in which the response to the change in PV streamer position is linear.) The theory that group 3 is fundamentally different from the others is supported by the precipitation patterns and tracks (Fig. 9; noting that the large track jumps between North Africa and Crete are unlikely to be accurate) parcel trajectories (Fig. 10) and parcel properties (Fig. 11). Because a nonlinear response / bifurcation is known to impose strong limits on predictability, identifying and describing such behaviour in this case would be an important outcome of this work. I hope that some of the length reductions achieved by improving the manuscript's focus and organization can be invested in a much more thorough analysis of this possibility.

3. The values of QG vertical motion seem too small to be very meaningful despite being described as "strong" in the text (L441). Vertical motions of ~0.5 mm/s and 1 mm/s (0.005 and 0.01 Pa/s) are plotted in Fig. 2, while values of up to ~5 mm/s (0.05 Pa/s) are plotted in Fig. 7. These values are all well within the typical rms of QG vertical motion at midlatitudes and mean vertical motions across the globe (Stepanyuk et al. 2017). If these calculations and plots are correct, then the vertical motion forcing from the upper levels is almost irrelevant to the real vertical circulations in most cases. Such weak vertical motions would need to be sustained in-place for many hours/days to have any appreciable impact on moistening or stability. For example, air in the peak ascending region in Fig. 2c ascends <10 hPa in a day in response to QG forcing, an ascent rate that is dwarfed by the 600 hPa ascent in the rising parcels near the centre. If the calculations are correct, then the relevance of the PV streamer to ascent and cyclogenesis needs to be seriously reconsidered in this case, an exercise that will likely lead to

conclusions that are completely different from those arrived at by the current submission.

4. The motivation for the case study approach adopted by the study is weakened by passages that highlight case-to-case variability, and is not supported by a clear statement of the useful aspect of the case study framework. The dominance of case-to-case variability is particularly emphasized on L38 and L84, with the latter appearing to be a direct criticism of the case study as a useful analytic tool. It is good to identify the limitations of the adopted investigation technique, but this criticism should be balanced with a clear description of what the case study approach can provide that other types of analysis (e.g. climatology) cannot.

5. The analysis of vertical coupling in section 5 is not quantitative enough to be included in the study. Despite significant discussion of Fig. 7e-h (L241-253), the strongest conclusion that is draw is that it is "most likely" that baroclinic instability is active. Even this conclusion appears to over-reach the analysis given that no baroclinic growth rates were computed. Given that the Icelandic low is not the focus of this investigation and that the left-hand column of Fig. 7 shows a convincing evolution of short-wave anomaly growth, I think that the right-hand column of Fig. 7 and the associated discussions should be removed. If this analysis is to be retained, then there needs to be a real quantification of baroclinic coupling and associated growth rates [note that the 12-18h time scale is very rapid for pure baroclinic growth, which typically has a doubling time scale on the order of a day (Hakim, Encyclopedia of the Atmospheric Sciences) and suggests that moist processes are likely to be very important].

6. The study of PV error growth by Baumgart et al. (2018) is referenced in the introduction, but not in section 5, where the left-hand column of Fig. 7 bears a striking resemblance to Fig. 3 of that work (albeit with a compressed time frame). The discussion of the importance of non-linear upper-level Rossby wave dynamics here follows closely that of Baumgart et al. (2018), so much of this description could be replaced by citations and comparisons. The Torn (2015) normalized difference is a useful measure, so compressing section 5 to focus on that metric in the context of the Baumgart et al. (2018) interpretation of this process would allow for a dramatic shortening of this section and serve to place this submission in the context of investigations by other groups.

7. Assessing the significance of the differences discussed in section 5 is important; however, the technique and in-text descriptions should be revised. Wilks (2016) provides a description of problems with the multiple-testing technique (as adopted in this study), which can lead to over-confident statements about significance. Please consider using the false discovery rate here. Additionally, the level at which the differences are considered significant is not identified in the text, and only appears in the Fig. 7 caption (is 0.05 used throughout?). Note that there is currently a reconsideration of the use of the term "significant", which appears to be leaning in favour of providing p-values rather than definitive statements about significance. I'm not very familiar with that discussion, but it might be of interest to consider during revision.

8. I am surprised not to see any references to Wiegand and Knippertz (2014), who study the representation and predictive skill of anticyclonic RWB and PV

streamer formation over the Mediterranean region in the ECMWF ensemble (i.e. an earlier version of the same system used here). That work seems so directly relevant to this study (including the conceptual diagram in Fig. 10 of that paper) that it should be leveraged heavily in this investigation, particularly in terms of putting the forecast uncertainty in this case in a broader context.

9. The numbering of clusters forces readers to remember the mapping: 1 is centered, 2 is west and 3 is east. Why not call the clusters C, W and E? Then the Fig. 8 rows could be reordered to W, C, E so that there's a progression in the columns rather than having the PV streamer location (and eventual cyclone location) jumping around.

10. Throughout the study, the "surface cyclone" is discussed by the 850 hPa heights are shown. Showing 850 hPa winds is useful, but I don't see anywhere in the manuscript that the 850 hPa heights are essential to the analysis. I think that all plots that currently show 850 hPa heights should be replaced with mean sea level pressure for consistency with the text.

11. Throughout the study, short-range ECMWF forecasts are used to estimate precipitation accumulations. To avoid model biases and potential "twinning", it would be preferable to use an independent product. The GPM IMERG is readily available and would be a better choice for this study than stitched-together IFS forecasts.

12. Advection of cold air over warm Mediterranean waters is identified as a factor that increases latent heat fluxes and promotes convection; however, this effect is not quantified in the current investigation. The OAFlux dataset covers the period of interest and is readily available for this kind of study. Please consider supporting the claims made in the manuscript with an analysis of OAFlux (or equivalent) surface flux estimates. An augmented surface flux analysis may particularly interesting if model-predicted fluxes are found to be very different between groups 1/2 and group 3 (see item 2 above). Such an analysis is essential if the categorical statements about surface fluxes currently found in the conclusions (L429) are to be retained.

13. The manuscript really needs to be clear about whether the medicane itself is a focus of the study. In multiple passages, it is stated explicitly that the medicane is not going to be investigated as part of this work (e.g. L97, L161, L320). However, much of section 6 is dedicated to the evolution of the medicane, including trajectory and CPS analyses. The title of the manuscript also emphasizes the storm morphology and will attract readers interested in medicanes. It feels as though the work was initially focused entirely on the PV streamer, and that "mission creep" has led to the introduction of more storm-scale-relevant material. Please reconsider the statements that disavow the relevance of the medicane structure for this work in an effort to remove what seems like a fairly important internal inconsistency in the manuscript.

14. Why are the ECMWF data coarsened to 1°, and how is it done? The result is very poor resolution in the graphics, and if it not done carefully, the operation could result in aliasing. Is a conservative remapping used? This is a particularly important question for the precipitation field, where the difference between

sampling/interpolation and remapping/aggregation can be enormous when the degradation of resolution is so large.

15. Most published works do not consider "medicane" a proper noun (and it is therefore not capitalized). This is analogous to "hurricane", which is only capitalized when a specific storm is discussed (e.g. "Some think that Hurricane Katrina was a category 3 hurricane at landfall"). Consider using lower case "medicane" throughout except in named reference to Medicane Zorbas.

16. The terms "air mass", "airstream" and "parcel" seem to be confused in relation to trajectory analyses (L139 and section 6.2). An "airstream" is a loosely defined concept, but I think that it would be represented by a high density of air parcel trajectories in a limited area. Then the phrase "trajectories of the airstreams" (Fig. 10 caption) doesn't make sense unless the airstream (a feature in storm-relative coordinates) is somehow tracked over time. Similarly, trajectories do not track "air masses" (L139), but parcels. The difference is important, because it is unlikely that all parcels in an "air mass" are ascending near the cyclone centre.

17. The trajectory analysis in section 6.2 is incomplete. The suggestion that moistening is occurring because of surface latent heat fluxes (L345-346) implies that the parcels are in contact with the surface; however, the vertical position of the parcels is never shown. It is also possible for parcels to be moistened by evaporation of falling precipitation or by turbulent mixing. It is therefore not demonstrated that enhanced surface fluxes are responsible for the moisture changes in groups 1 and 2. The same is true for the potential temperature analysis on L346-348: surface fluxes are only one possible reason for potential temperature increases, and only influence parcels if they are in contact with the surface (even at above-surface levels in the boundary layer, the moistening/heating mechanism would be turbulent flux convergence rather than surface fluxes per se). The lack of information about the trajectories makes it impossible for reviewers or future readers to confirm the validity of the conclusions drawn at the end of this section (L356-370).

18. Section 6.2 ends with a set of suppositions and conjectures based on an incomplete trajectory analysis (see previous item) climatological behaviour. As a result, terms such as "could favour" and "might support" are used instead of definitive statements. If the analysis and descriptions in this section cannot be made robust enough to be able to conclude these statements definitively, then this section should be removed.

19. The description of the CPS (L386-392) is insufficiently detailed to allow independent confirmation of the results (a requirement for publication). Because of the small scales of medicane structures, the hurricane-based radii are usually reduced for studies of Mediterranean storms. Was the same done here, or were the original hurricane-based values used?

20. I don't understand the "deep warm core" (DWC) analysis in Fig. 12. Take groups 2 and 3, for example. They have 12 and 18 members, respectively. The average number of DWC in group 2 is 7.2, and 7.0 in group 3 according to Fig. 12. That number is "per ensemble member", so multiplying by the relevant ensemble size yields 7.2*12=86.4 for group 2 and 126 for group 3. However, the total number of DWC steps for group 2 is given as 43, and that for group 3 is given at just 14 at

the bottom of the plot. In the text (L404) the reader is told to consider the group-3 DWC analysis "with caution, due to the small sample size". However, the average number of DWC steps per ensemble member is as large in group 3 as it is in group 2: why is the sample size so small? There seems to be something about the number of **sequential** DWC steps ("duration"), but that is never clearly stated in the text or caption. What is wrong with my interpretation of the DWC analysis?

21. Throughout the text, equivalent potential temperature gradients are used to identify both baroclinic zones and moisture gradients [L125, L132, L137 (where the 850 hPa theta-e is inappropriately used to identify a "weak surface cold front") L153 and elsewhere]. Strictly, neither of these is guaranteed by a theta-e gradient, which may arise as a result of either in isolation. If baroclinicity is important, then potential temperature (or temperature on an isobaric surface) should be shown. If moisture is important, then it should be shown. Theta-e is a very useful quantity for assessing convective potential and is a useful way to identify the warm sector for the trajectory analysis, but it does not replace the more basic fields for questions of baroclinicity and moisture.

22. There are a lot of very specific geographical references throughout the text, probably more than there need to be. I'm a geographer, but I still found myself having to look for specific place-names on maps. It would be very useful to have a new Fig. 1 that shows (at least) the storm track and labels for all place names referred to in the text.

23. The conclusions of the study are not supported by the evidence provided in the text:
    a. The "clustering" technique is not rooted in a mathematic definition and fails to guarantee the separation of the members into distinct "scenarios" as stated in the text (e.g. at L481). Is it true that there are three "distinct scenarios"? I agree that there are two (see General Comment 2), but I don't see why there are three. Groups 1 and 2 are distinguished only by the fractional overlap of the PV streamer, and there was no demonstration that there is any sort of heterogeneity in overlap space. This is a weakness in the analysis that results from the failure to use a true clustering analysis, and the decision to rely on a classification heuristic. There is no guarantee that group 1 and 2 events are separate from each other in any kind of meaningful way, and selection of a different overlap threshold (70%, for example) would result in the progressive reclassification of members from one group to the next. To demonstrate the presence of different scenarios, a true cluster analysis should be performed, and the optimal number of groups should be identified (e.g. using the "elbow method).
    b. There was no analysis of the near-surface flow induced by the PV streamer, so how is the conclusion about induced advection (L432-434) supported by the evidence provided in the submission? Particularly given the limited spatial extent of the streamer immediately prior to cyclogenesis, it is possible that the induced near-surface flow is very strong. For the arguments regarding air parcel modification by surface fluxes, the parcels approaching the centre in groups 1 and 2 must be in contact with the

surface, putting them as far as possible from the upper-tropospheric streamer.

    c.   I cannot see what part of the analysis is used to conclude that the group-1 PV streamer was better able to "maintain the cyclonic circulation" (unclear whether this refers to the upper- or lower-level flow) than the group-2 or group-3 features (L434-438). There appears to have been a rigorous analysis behind this statement (something that determines the number of members that meet a "condition"), but I don't know what section this analysis was described in.

    d.   The increase in the amplitude of the cyclonic PV anomaly from about -0.5 PVU to beyond -2.5 PVU (combined with a rapid areal expansion) over the 24-h period ending at 1800 UTC on 25 September (Fig. 7b and d) is "rapid" as stated on L442. However, as noted in item 4 above, this growth rate appears to exceed that expected for typical midlatitude baroclinic growth. It is highly likely that moist processes are involved, but because no estimates of growth rates are made in this study, it is impossible to know. It is therefore also inappropriate to conclude that the observed growth is "as expected from baroclinic instability" (L442) because the expected value remains unknown in the context of this work.

    e.   It is unclear to me what part of the analysis demonstrates that "the contributions of diabatic airstreams … were negligible for the uncertainty amplification in this case" (L444-445). The non-conservative evolution of the PV streamer was remarkable in this case (Fig. 2), and the impact of diabatic PV reduction in WCB outflow on ridge amplification during the upstream RWB (Fig. 7a-d) was not analyzed in the study, as far as I can tell. This statement about the role of diabatic process on forecast uncertainty (L444-445) is very strong, inconsistent with previous work, and needs to be clearly supported by the presented analysis.

**Minor Comments**

There are a relatively large number of grammatical errors in the submission, which I have not itemized here because of the major reworking of the text that will be required to address the issues identified above.

1. [L50] It is not clear why the introductory reference to paramterization uncertainty is useful here, where initial condition uncertainty is described in the subsequent passage. I would suggest starting this paragraph with "A major source …".

2. [L51] I don't think that "slight uncertainties in initial conditions **typically** grow" (my emphasis), because the majority of uncertainties in any given analysis project onto decaying modes in the atmosphere (Privé and Errico 2013). I think that it would be more precise to say something like, "Slight uncertainties in the initial conditions that project onto the growing modes of the atmosphere can increase in amplitude during the forecast and potentially …". You could also just replace "typically" with "can" in the current phrase.

3. [L87-L94] Suggest dropping this subsection in favour of the analysis in section 3.

4. [L95-L97] Having a clear set of objectives is a good idea, but these questions are framed in a way that is too complex to make them useful for the reader (e.g.

"what is a and what of b leads to c and d in e"). Suggest simplifying or removing these questions.

5. [L99-L105] Provide a standard outline with section references.
6. [L108] How are the ensemble members "perturbed": initial conditions, stochastic physics, SPPT, etc?
7. [L111 and elsewhere] The word "data" is plural, so "data are available", etc.
8. [L115] What climatology is used for the ACC calculation?
9. [L122] Reference to a URL is inappropriate. At the very least, an access date needs to be provided. Consider including lightning strike information on the plots, rather than making reference to external information that may not be permanently available.
10. [L152] How is conditional instability identified in this analysis?
11. [L159] Figure 4a does not exist and Fig. 4 is not the CPS.
12. [L207] Reference to Fig. 7 is out of order.
13. [L221] At what level are the differences significant?
14. [L231-L232] This sentence doesn't make sense: does the amplitude "propagate" at a different speed from the difference? Are you differentiating between phase speeds and group velocities here? Please rephrase to make this clearer.
15. [L263] The section title should be much clearer, and not read like a news headline.
16. [L274] Three different time references begin this sentence. Please determine whether it is the time relative to streamer extension, Gregorian date/time, or forecast time that is most relevant here and stick to this description of the first column of Fig. 8.
17. [L278] I don't see that cluster 3 trough is "clearly" shifted to the east of the analysis at 1200 UTC 25 September (Fig. 8i). Instead, I see a trough that is too narrow, notably on the upshear flank over Germany.
18. [L281] Why isn't significance plotted here as in the first column?
19. [L304-L305] This looks like more than just smoothing of the ensemble mean. Because averaging is a linear operation, the area-averaged ensemble-mean precipitation should match the observed values if the ensemble does not under-predict rainfall.
20. [L297] Are these SLP changes computed from the central pressures of the ensemble members, of from the ensemble mean? The search for a minimum central pressure is not a linear operation, so the results will likely be sensitive to the method. Particularly given the broad spatial distribution of group-2 centres, some/much/all of this apparent weakening may simply be the dilution of the ensemble mean if the ensemble averaging is done first.
21. [L312-315] The four lines of hypothesis here would be much better invested in the actual analysis rather than this forward-referenced supposition (I recommend the removal of this whole paragraph as noted in item 1.b.iii above).
22. [L317 and L320-L321] There seems to be an internal inconsistency here. On L317 the objective of the section is stated to be "to investigates … subsequent development of a medicane-like system". However, on L320-321 you state that you "do not identify low-level warm cores directly and do not investigate their formation in detail". Because the warm core is one of the primary structural

ingredients that distinguishes medicanes from typical Mediterranean cyclones (considering the CPS), these two statements seem to be in direct conflict.

23. [L321] What do you mean that you don't identify warm cores directly? The CPS-based warm-core detection is the basis for a large part of section 6.3.

24. [L344] Do you mean a larger increase in specific humidity in groups 1 and 2, than in group 3? This sentence suggests the opposite, likely because the intended target of the pronoun "this" is unclear (although the construction suggests group 3).

25. [L353-L355] What is the physical relevance of this comparison?

26. [L393-L399] Why bother with a set of conjectures right before performing the actual analysis? A far more direct approach would be to explain why the fractions of medicanes in each group differ, based on the analysis presented in earlier sections. The conjectures do nothing to build suspense for the big "reveal" of Table 1, and just serve to consume five lines of text unnecessarily.

27. [L399-L401] This text contains every number shown in Table 1, without offering any physical insight. Choose to present these numbers either within the text, or in a table, but not both.

28. [L413-415] How is it concluded that "the detailed interaction between the surface cyclone and the upper levels become limiting factors" for predictability? Why can't internal storm processes or air-sea exchanges be the limiting factors? Those processes have not been investigated or ruled out as limits on storm structure predictability in this analysis, as far as I can tell.

29. [Fig. 6] Can the map resolution be increased a bit? (Similar in Fig. 7 zooms.)

30. [Fig. 6] At what level are the contours significant?

31. [Fig. 6] The means are too similar to be usefully distinguished on the plot. Consider plotting the full ensemble mean only rather than solid and dashed contours.

32. [Fig. 7] Is that a reference vector between (d) and the colour bar? If so, it should be highlighted and described in the caption. If it isn't, then one should be added.

33. [Fig. 10] Use a fixed domain to ease comparison between panels.

34. [Fig. 10] What does the colour-coding of the trajectories represent (the last sentence of the caption is not clear about what is indicated "in colors")? Are different members assigned random colours? Why are there fewer cyclone positions in the groups than members within the groups? Are there multiple trajectories ending at the same point because of the degradation of the grid resolution? If so, there should be some way to represent the number of overlapping triangles (potentially the size of the triangle).

35. [Fig. 10] How does the maximum "percentage of ensemble members with an airstream occurring at the specific grid point" occur outside of the trajectory envelope? For example, the maximum departure frequency in Fig. 10b occurs poleward of any trajectory. Is it because these trajectories are actually averages of many trajectory calculations? If so, then there must be some unusual spatial distributions to obtain density maxima away from the means. How many trajectories are computed in each member?

36. [Fig. 11] Why are radii the best way to identify the blue and green lines? It would be clearer to label the blue line "center" and the green line "warm sector" because the radii are technical details rather than relevant features.

**References**

Privé, N. C. and Ronald M. Errico (2013) The role of model and initial condition error in numerical weather forecasting investigated with an observing system simulation experiment, Tellus A: Dynamic Meteorology and Oceanography, 65:1, DOI: 10.3402/tellusa.v65i0.21740

Stepanyuk, O., Jouni Räisänen, Victoria A. Sinclair & Heikki Järvinen (2017) Factors affecting atmospheric vertical motions as analyzed with a generalized omega equation and the OpenIFS model, Tellus A: Dynamic Meteorology and Oceanography, 69:1, 1271563, DOI: 10.1080/16000870.2016.1271563.

Wilks, D.S., 2016: "The Stippling Shows Statistically Significant Grid Points": How Research Results are Routinely Overstated and Overinterpreted, and What to Do about It. *Bull. Amer. Meteor. Soc.,* **97**, 2263–2273, https://doi.org/10.1175/BAMS-D-15-00267.1

Wiegand, L. and P. Knippertz, 2014: Equatorward breaking Rossby waves over the North Atlantic and Mediterranean region in the ECMWF operational Ensemble Prediction System. Quart. J. Roy. Meteor. Soc., **140**, 58-71.

---

## Referee Comment (RC2) · Florian Pantillon (Referee) · 14 Sep 2019

Review of "Medicane Zorbas: Origin and impact of an uncertain potential vorticity streamer" by Raphael Portmann, Juan Jesús González-Alemán, Michael Sprenger, and Heini Wernli.

The paper investigates the large-scale dynamics that led to the formation of a tropical-like cyclone over the eastern Mediterranean in late September 2018, which was characterized by high forecast uncertainty in the operational ECMWF ensemble prediction system. A potential vorticity streamer issued from an anticyclonic Rossby wave breaking over eastern Europe was key in the Medicane dynamics. Two clusters of ensemble members with zonal shift of the streamer can be tracked along the Rossby wave guide

back to initial conditions over North America. The evolution of the streamer into an upper-level cut-off low then controls the surface cyclogenesis, the stability and the advection of warm moist air that all support the Medicane formation.

Hybrid cyclones in general and Medicanes in particular are current sources of vivid discussions in atmospheric dynamics and objects of broad interest in the Mediterranean community. Contributions to better understand their dynamics and predictability are thus welcome and the paper presents interesting new material based on sound methods and high-quality figures. However, it suffers two major shortcomings: possible contributions from small-scale dynamics are largely ignored, although they at least partly explain Medicane formation, and the manuscript needs reorganization, as already pointed out by Referee 1. These shortcomings are linked somehow, as the tropical transition of the cyclone is actually assessed at the very end of the paper only. They are described below, as well as (many) specific comments.

The paper thus requires substantial revision before it can be considered for publication in Weather and Climate Dynamics.

General comments

Scales: as stated in the introduction, "the relative role of positive upper-level PV anomalies and air-sea interaction for the intensification of Medicanes is currently debated, as well as the question to which degree they are dynamically similar to tropical cyclones". The paper focuses on the synoptic scale and is based on model forecasts that do not explicitly resolve convection. This is fine but (1) the focus should be explicitly stated, (2) the limitation should be kept in mind throughout the paper and (3) the results should contribute to the current debate.

Organisation: as already pointed by Referee 1, the structure of the paper is unsatisfactory. Please better organize the Sections, make sure important concepts are introduced early in the paper (then stick to the terminology) and methods are described in the appropriate section, and avoid referring to later sections. In particular, show the

warm core structure early in the paper, and comprehensively, based on the analysis for instance; in the present form the reader must wait until the last subsection of the last results section to learn the cyclone actually developed a warm-core structure.

Specific comments

Title: the position and depth of the PV streamer exhibit some uncertainty in the ensemble forecast but the streamer itself is not uncertain; the link between PV streamer and Medicane could be more explicit.

Abstract

l. 3-4 This statement is not clearly supported. l. 5 "uncertain" is not properly used here (see comment on title). l. 7-8 "demonstrated", "the dominant source": not necessarily. See comments below. l. 9 Twice "strong(ly)". l. 12 More details about the two air streams and their key role?

1 Introduction

l. 19-25 All references relate to the North Atlantic, which should either be explicitly stated or extended to other oceanic basins. l. 19-20 ET could also be mentioned here. l. 26-33 It is unclear what is the difference between subtropical, tropical-like and hybrid cyclones, if there is one at all. And do not they by definition undergo tropical transition? l. 29 "air-sea feedback" is not precise enough. l. 32 This "may" result in high damage, as Medicanes often remain over sea. l.35-39 Confusing what "they" and "their" refer to. l. 41-42 Forecast uncertainty and the link with process understanding and ensemble forecasting needs better introduction. l. 42-49 This appears too early and several keywords are not introduced yet (Zorbas, warm core, practical predictability, . . .).

l. 50 The transition should be smoother between 1.1 and 1.2. l. 63 Lamberson et al. studied "extratropical" cyclone Joachim. l. 64-65 The link between the predictability of breaking Rossby waves and Medicanes is far from obvious and need more details; it was extensively explored for a case study in September 2006: Chaboureau, J. , Pantillon, F. , Lambert, D. , Richard, E. and Claud, C. (2012), Tropical transition of a Mediter-ranean storm by jet crossing. Q.J.R. Meteorol. Soc., 138: 596-611. doi:10.1002/qj.960 Pantillon, F. , Chaboureau, J. , Lac, C. and Mascart, P. (2013), On the role of a Rossby wave train during the extratropical transition of hurricane Helene (2006). Q.J.R. Mete-orol. Soc., 139: 370-386. doi:10.1002/qj.1974 Pantillon, F.P., J. Chaboureau, P.J. Mas-cart, and C. Lac, 2013: Predictability of a Mediterranean Tropical-Like Storm Down-stream of the Extratropical Transition of Hurricane Helene (2006). Mon. Wea. Rev., 141, 1943–1962, https://doi.org/10.1175/MWR-D-12-00164.1 l. 67 Who are "they"? l. 67-69 The mentioned studies do not clearly attribute forecast busts to initial uncertain-ties rather than to the representation of diabatic processes. Both error sources would thus better be described together here. l. 79 Which process?

l. 86 Even if only few case studies exist, there are certainly more than the two cited here. l. 89 Some basic information about the case study are needed here (what, where, when). And where does the name "Zobras" come from? l. 91 Please explicit "short lead times". l. 94 Which PV streamer? Either detail or remain general; referring to Section 3 does not help. l. 96 "leads" or "led" l. 100-106 Detailing Sections 2, 3, 4, . . . might be required.

2 Operational ECMWF products

l. 110 Why 46 members only? What is the operational short-term forecast? l. 112 Forecast data is actually available at higher frequency. l. 115 What is the reference for computing ACC?

3 Synoptic overview

Figures 1-3 Zooming in on the region of interest would be very helpful to follow the discussion. Large-scale dynamics play an important role but, e.g., the Irish and Red Seas are not relevant. Consider then merging the three figures to avoid jumping from one to the other. l. 135-136 The spiral-like structure is hardly seen. Or do you mean the frontal band? Spiral often refers to a tropical structure. Again, zooming in would

help. l. 139 Fig. 2C l. 140-141 Move to the methods section. l. 144, 146 What are "they"? l. 139-149 Is it convection and/or large-scale ascent? The 600 hPa in 24 h criteria suggests the latter, while lightning suggests the former. The ECMWF model cannot actually resolve convection but you could check whether the precipitation is issued from the convection scheme or not. l. 135-149 There is a general confusion in the paragraph between what was stated by previous studies and what happens here. l. 152 Fig. 3d l. 152 Clarify the precipitation is from model data; using different colors would help distinguishing the $-11$, 8, 15 and 21 mm (6h) contours on Fig. 1. l. 153 How can you know it is due to conditional instability? l. 156-157 How do you discern a warm seclusion from a warm core? l. 157-160 and 166-167 There is not enough evidence at that point to claim a tropical transition. Relying here on Sec. 6.3 is not a good idea and there is no Fig. 4a. Either show more details or keep for later. l. 162 This would be worth showing!

4 Ensemble clustering according to position of PV streamer

l. 170-172 The sentence presents essential information but needs more support: why 00 UTC 27 Sep? Why 00 UTC 24 Sep? Is it perhaps the combination of valid time and initialization time resulting in largest spread? Can we see this somewhere? "Shown in Fig. 2 six hours earlier" is not too convincing. l. 172-174 Referring to a later Section to motivate the present one is surprising. l. 174-176 Please move technical details to the Methods section. l. 178 Average of 320K and 330K levels or are there additional levels in between? l. 178 Why set PV<2pvu to zero? l. 180 Remind there are 50 members? l. 190-191 Again, referring to a later Section is surprising. l. 192-193 "Decrease" rather than "drop"? Are these values of ACC particularly low? And why not color clusters 2 and 3 in green and red on Fig. 5 as on Fig. 4? l. 198 Errors in the shape of the PV streamer have not been discussed yet, only the zonal shift. l. 199 Which characteristics are relevant? l. 200-204 ACC may not account for the cyclone at all, at least the link is not showed yet. The link with the PV streamer is not obvious either. Consider adding Z500 on one of Figs 1-3.
5 PV streamer scenarios emerge from initial condition uncertainties and baroclinic amplification

l. 205-206 The jet streak has not been mentioned before. Consider either adding the large-scale dynamics leading to the PV streamer in Section 3 or, at least, shortly describing these dynamics here and motivating why they are the focus of the following analysis. l. 209-220 Please move to the Methods. Can you say some words about deltaPV values, e.g., is there a threshold that indicates bi-modal distribution? l. 221-223 While the normalized PV difference is clearly highlighted, PVU and wind contours barely differ between clusters and are not discussed at all. l. 230 The separation between clusters is hardly seen at that point. l. 238-239 Remind Fig. 7d; better "stronger anticyclonic wave breaking" than "westward phase shift and larger amplitude"? l. 242-254 The description of Fig. 7e-h is difficult to follow and Fig. S1 suggests that differences in omega and Z850 are hardly significant in the region of interest. Consider removing altogether. l. 244, 250 Either show or omit potential temperature. l. 255-263 This interpretation is meaningful and consistent with displayed material overall but (1) the formulation partly sounds speculative ("strong QG forcing", "uncertain low-level wave", "exponential growth", "strong vertical coupling") and (2) ensemble members differ not only in their initial conditions but also in their physical parametrisations or any other perturbations implemented by ECMWF to increase ensemble spread.

6 Diverging synoptic development impacts Medicane predictability

l. 268 What is a "Medicane-like" cyclone?

l. 270 The subsection provides a synthetic summary of the dynamics of all clusters, but what is the variability between members within each cluster? l. 275-276 Mention the analysis PV is depicted by the black contour. l. 277 What is meant by "exactly the ones"? l. 278-279 There is no visible difference in PV between clusters in Fig. 8 a, e, i. l. 288 Fig. 2c; slightly different time. l. 288-291 Mention the cyclone in individual members is depicted by dots. l. 295 Differences are substantial but not necessarily

due to latent heat release (only). l. 297-298 Clarify these are mean values. l. 305 Fig. 9d l. 306-306 The smoothing effect due to averaging makes the comparison difficult for precipitation; how do individual members look like? You could e.g. compute PDFs of accumulated or instantaneous precipitation for each member and the analysis. l. 314-316 These arguments are too speculative and are better left to Sec. 6.3.

l. 318 Again, what is a Medicane-like system? l. 318-325 This paragraph is confusing and must be rewritten/streamlined. How do you define a low-level warm core, a warm seclusion and a Medicane-like system, and why do you focus on two air streams? l. 326-327 The analysis tool Lagranto belongs to the methods and is already mentioned above. l. 330 Is the warm core formation shown somewhere? l. 345 "weaker" not "stronger" increase. l. 346 In clusters 1 and 2. l. 347 The Mediterranean Sea is not an ocean. l. 353-356 This would likely better fit at the beginning of the paragraph. l. 363 Closer to the coast but the region remains the eastern Mediterranean. l. 363-371 The discussion is speculative so far, as the cyclone thermodynamics have (still!) not been documented yet. There is also a general confusion between warm core, warm seclusion, warm sector and tropical structure.

l. 377-394 This all belongs to the Methods. What radius is used to compute CPS metrics? l. 394-399 I expect clusters 2/3 to show more favourable low-level/high-level forcing but not necessarily to produce a stronger/weaker Medicane. l. 400 Avoid introducing an additional name ("DWC"), which adds confusion, better stick to the terminology used up to that point. l. 400-402 What about the two other CPS metrics, symmetry and low-level warm core? The upper-level warm core metric might be contaminated by te presence of the PV streamer/cutoff. And what about the cyclone intensity? l. 403-404 But cluster 3 produces stronger upper-level warm cores than cluster 2, which contradicts the other results and interpretation. l. 407-409 The three-day long sustained deep warm core (Fig. 5a) appears unprecedented. Can you provide CPS diagrams for the analysis? l. 412-413 Why? l. 414-416 What about convection?

7 Conclusions

l. 427-418 Again, what is a subtropical cyclone, a tropical-like system or a Medicane? l. 431 More details about this "first case"? l. 432 Which process? l. 435-438 This is not shown here. l. 445-446 How do you know this? l. 446-447 Which process, baroclinic instability? l. 455-457 Ensemble forecasts are computed with different perturbation methods thus the error growth cannot be attributed to initial conditions only here. l. 458-460 . . .and convection and its organisation.

l. 462 As the used data is from ECMWF essentially, it could be stated how to access it.

References

Providing DOIs or URLs for all papers would be helpful.

Figures

Moving all figures to the end of the paper would ease the review. Fig. 5: it is unclear which date relates to which tick mark. Fig. 7 appears before Fig. 6. Fig. 6: consider changing the color scale to [-1,5; 1,5] and plotting coast lines at higher definition. Fig. S1: the title should refer to Fig. 7 not 6.

---

## Referee Comment (RC3) · Anonymous Referee #3 · 17 Sep 2019

This interesting paper deals with the predictability of the Medicane Zorba, which affected the central Mediterranean in September 2018. ECMWF ensemble forecasts are used in this effort. The limit in the predictability of the cyclone is analyzed and discussed in connection with the upper-level PV, which also affected the low-level evolution.

This is one of the first paper that clearly identifies the relevance of PV features in the predictability of Medicanes. The results are a relevant contribution in the field; however, the analysis should be substantially improved. Some points are indicated hereafter.

Major points:

- Line 27, Line 154-157: I would like to see some clarifications about the definition of

[Figure]

Medicanes. Although there is no general consensus, in most of the literature (e.g., Miglietta et al., 2011; Picornell et al., 2014; Cavicchia et al., 2014), a Medicane is considered as an extra-tropical cyclone that acquires a symmetric, deep warm core in the Mediterranean region. At the same time, the presence of a deep, warm core is not always an indication of tropical-like processes going on: as discussed in Fita and Flounas (2018) and Mazza et al. (2017), a deep warm core is not necessarily associated with a WISHE mechanism, but it can also be induced by a warm seclusion. However, Miglietta and Rotunno (2019) have shown that the intensification of the same cyclones discussed in the latter two papers cannot be explained without considering the sea surface fluxes and the latent heat release, in analogy with the WISHE mechanism typical of tropical cyclones. For these reasons, I suggest to remove "occasionally" (Line 27) and the sentence "Warm seclusion have been previously linked to Medicane formation" (Lines 155-156). Also, at Line 317 and 321 you "investigate potential precursors of a low-level warm core": however, the low-level warm core is not relevant for the following development of the cyclone in itself (see also Line 365, 396), but because of the high values of equivalent potential temperature that are responsible for potential instability and favor the development of convection at later times. - Figure 10: I found the understanding and interpretation of Figure 10 quite difficult; in particular, I did not understand if the backtrajectories you show are averages over all the ensemble members, since this is not mentioned in the Figure caption and not clearly reported in the text (Line 334); also, the presence of high percentages far from the plotted trajectories (purple shading) is counterintuitive; - Line 350-355: for a more comprehensive analysis of the trajectories in Fig. 10, some information should be included about the change of height along them;

Minor points: - Line 115: please provide the definition of ACC; - Line 130: red instead of blue; - Lines 143-145: the role of upper level PV anomaly in the generation of Medicanes is also discussed in Miglietta et al. (2017); - Line 201: were instead of is; - Line 202: severely instead of severly; - Line 207: please can you provide an approximate indication of the height the isentropic = 325 K corresponds to? - Line 214: why do you

give less weight to the regions of strong gradients? - Line 221: explain why "stratospheric" side; - Line 311: favorably instead of favorable; - Line 317 and elsewhere: Medicane or tropical-like, not Medicane-like! - Figure 5 caption: why do you use only 46 members for the second ensemble? - Figure 6: change contour line colors to facilitate interpretation; - Figure 7 caption: please indicate that the black contour refers also to captions (a-d); - Figure 8 caption: (e,i) instead of (e,f); the black contours around the teal patches create confusion; - Supplement material, Line 9: "The results show that significant differences of QG are located in the region of strong QG on 1800 UTC 24 Sep 2018": it does not seem to be the case, at least at that time.

REFERENCES: Cavicchia, L., von Storch, H., Gualdi, S. (2014). Mediterranean tropical-like cyclones in present and future climate. J. Clim. 27, 7493–7501. Miglietta, M.M., Moscatello, A., Conte, D., Mannarini, G., Lacorata, G. and Rotunno, R. (2011). Numerical analysis of a Mediterranean hurricane over south-eastern Italy: sensitivity experiments to sea surface temperature. Atmospheric Research, 101, 412–426. Miglietta, M.M., Cerrai, D., Laviola, S., Cattani, E. and Levizzani, V. (2017). Potential vorticity patterns in Mediterranean "hurricanes". Geophysical Research Letters, 44, 2537–2545. Picornell, M. A., J. Campins, and A. Jansà (2014). Detection and thermal description of Medicanes from numerical simulation, Nat. Hazards Earth Syst. Sci., 14, 1059–1070.

---

## Author Comment (AC1) · 28 Nov 2019

Paper wcd-2019-1

**"Medicane Zorbas: Origin and impact of an uncertain potential vorticity streamer"**

Raphael Portmann, Juan Jesús González-Alemán, Michael Sprenger, and Heini Wernli

**Response to the Reviewers' comments:**

We thank the reviewers for their many critical and constructive comments that helped improving the manuscript. All points have been carefully considered and will lead to the following important changes compared to the original revision:

- We better explain the objectives of the study, which focuses on the synoptic-scale processes involved in the formation of this intense cyclone.
- The introduction is restructured, following the suggestions by the reviewers.
- The paper contains a separate data and methods section.
- The cyclogenesis of 'Zorbas' is introduced in more detail.
- The analysis of the error amplification and propagation is improved, and a few aspects of our previous analysis are omitted.
- The final part of the study (cyclogenesis in the ensemble members) is completely rewritten, also based on additional analyses.

Below are the detailed replies to the individual comments (in blue).

**Reviewer 1 (Ron McTaggart-Cowan)**

Background

The authors investigate the predictions of a September 2018 medicane in the ECMWF ensemble system. They identify ensemble members that have differing day-3 representations of a PV streamer involved in the storm's development. They track these differences back to small differences in the initial conditions and show the progression of PV spread in association with an anticyclonic Rossby wave break.

The remainder of the analysis focuses on the interaction between the PV streamer and the developing cyclone: distinct precipitation and storm structures are identified in the different sub-ensembles.

Reductions in predictive skill associated with the development of sub-synoptic systems in the Mediterranean region are an important subject for investigation. Similarly, the limits of predictability imposed by PV streamer evolution and interactions between such features and lower-tropospheric circulations are not well understood. Despite these interesting fundamental underpinnings, the current submission suffers from a large number of flaws in organization, preparation and analysis. Although each of these may not be considered fatal in isolation, a significant amount of effort will be required to address all of them thoroughly. Any revised submission of this investigation will necessarily be heavily modified and constitute a new piece of work. I hope that the authors will find the comments below useful as they pursue this research.

General Comments

1. The manuscript lacks organization and logical flow. This extends from the highest level of structure (sections and subsections), down to the paragraph and even sentence level. It makes the manuscript difficult to read and follow because concepts and details are disjointed, scattered and frequently repeated throughout the text. Ordinarily, I would consider these kinds of organizational issues relatively minor and possibly within the domain of the authors' discretion; however, in this case they seriously detract from the work and make it very difficult for readers to follow the investigation. Addressing these problems will involve the rewriting of much of the text, but will yield a more focused manuscript that will likely be shortened by at least 1-2 pages.

a. High Level

i. The structure of the introduction is ineffective. It begins with a very cursory review of tropical transition and medicanes (including PV streamers), then switches to a Rossby wave discussion that returns to error amplification twice, and then goes back to a very short summary of Zorbas. The latter also includes thesis questions and an outline of the remainder of the study that lacks section information or internal references.

We agree that the introduction could be better structured. We will streamline and rewrite parts of the introduction and provide section information when presenting the outline of the study.

ii. I do not think that the decision to replace the standard "Data and Methods" section with "Operational ECMWF Products" is effective. It means that additional data sources (satellite imagery, lightning detection, etc) have to be described in the text body, descriptions that seriously distract from the analysis when they occur. The same is true of methodological details (e.g. LAGRANTO, which is introduced twice) and the entire discussion of tracking and the CPS (L378-L392). Descriptions of all of these sources and techniques should be centralized in a "Data and Methods" section.

We agree and now include a proper "data and methods" section that introduces most methodological aspects of the study.

iii. There are numerous forward-references to section 6 throughout the earlier sections of the manuscript. While appropriate internal referencing is a useful tool, these repeated forward-references are likely an indication of poor manuscript organization, especially when they underpin important elements of the analysis (for example, the CPS referenced in sections 3 and 4 but never shown despite a reference to "Fig. 4a", which does not exist in the submission; L159). I think that the synoptic analysis (ideally shortened from its current length by enhanced focus) should include a discussion of the medicane itself, including the CPS. I understand that the medicane is not the intended focus of the work as is repeatedly stated in the text; however, the reader could be excused for thinking that it is because of (1) the title, (2) the multiple introduction subsections that deal with TC-like features, (3) the statement on L317 that the investigation of the "development of a medicane-like system" is an objective of section 6.2, and (4) the pervasive forward-referencing to the CPS analysis in the text.

We agree that the medicane itself should be more in the focus of the synoptic overview. We shorten some aspects of the synoptic overview, but add the CPS and the track of Medicane Zorbas.

b. Medium Level

i. Each section should begin with an introduction of the section contents so that the reader has an idea of how the section fits into the larger narrative of the work. A section should not begin with a description of data used in specific figures, as does section 3. Please revise each section introduction to ensure that the reader is logically guided through the work.

We agree that the text should be written in such a way that it guides the reader as elegantly as possible through the study. We consider this when rewriting the section introductions.

ii. Each paragraph should begin with an introduction to the paragraph contents, and should conclude with a statement that relates back to the material introduced. There are very few paragraphs in the manuscript that follow this basic structure. A particularly clear example of a paragraph that ranges far too broadly occurs on L50-64. The paragraph (and subsection) starts with a description of initial condition uncertainty, moves into ensembles more generally, then into PV error growth, and ends with a discussion of tropical cyclones. This lack of focus makes the study very difficult to follow despite the fact that the investigation itself is relatively straight-forward. In this case, the subject of error growth reappears in the paragraph starting on L73, which further adds to the confusion of the discussion. Please do not simply rewrite the paragraphs noted in this comment: the structure of virtually all paragraphs in the manuscript needs to be reconsidered and revised, an effort that will lead to rewriting of large portions of the work. The readability problems induced by the lack of logical internal paragraph are more than aesthetic in this case, and are serious enough that they significantly reduce the potential impact of this work.

We apologize for the obviously in parts confusing writing style in the original submission. However, we argue that the rule that *every* paragraph should begin with an introduction to the paragraph contents might be too restrictive. Again, we carefully reconsider the structure of the entire text in order to increase readability.

iii. I do not think that summary paragraphs at the end of a section are useful, particularly given the large number of relatively short sections in this submission. For example, the summary on L163- L167 is redundant with analysis undertaken in the previous page of the manuscript. Please remove summary paragraphs (they also appear at the ends of sections 5 and 6.1) in favour of making the text itself direct, clear and readable (see item 1.c.ii below).

We removed unnecessary summary paragraphs at the end of sections, which just repeated what has been discussed before.

c. Low Level

i. Reference to caption-level figure details within the text is highly distracting. For example, the fact that precipitation is shown in red solid contours in Fig. 1 is referenced three times in section 3 (once erroneously as "blue contours"; L130), while the fact that QG vertical motion is shown in red contours is referenced twice in the same section. These plotting details are described in the caption, and their appearance in the text detracts from the flow of the analysis. Figure and panel references should be enough to guide readers through the discussion. Please consider removing caption information from the text body throughout the submission.

We agree that the original version contained to many caption-level figure details. However, we are still convinced that, in some cases, they can help the reader to quickly grasp the relevant

information from the figures. Therefore, we carefully re-consider the usefulness of caption information in the text and remove them where possible.

ii. The writing style is too informal and lacks the precision required for scientific text. For example, the outline of the manuscript is described as "a journey" on L101, the analyses in section 6.2 "hint" at airstreams (L321), and the first person plural ("we") is used heavily throughout the submission. The introduction of section 6.2 (L320-324) can be summarized as, "we don't do anything thoroughly here, but we show stuff that's different and make some guesses about what that means; then in the next section we do it properly". I don't think that that kind of introduction (or approach to the analysis) will make readers want to continue to invest their time in the rest of the section. Throughout the submission, irrelevant details clutter the text (e.g. does it matter that 1800 UTC 26 September is "in the evening of the same day" on L126?), and ill-defined concepts reappear throughout the analysis (e.g. the "C-shaped" PV cutoff with a "dent" and "dent structure" on L138, 141 and 146, respectively). Every effort should be made to make the text succinct and readable, so that the analysis does not get lost in superfluous details and unnecessary bridging statements.

Here we ask the reviewer to please consider that we are not native speakers and, in some cases, we don't realize that our language becomes too informal. We thought that the "C-shaped" cutoff would be a useful terminology, but obviously it isn't. We improve the text and tried to avoid too informal terminology.

2. I think that cyclogenesis in cluster 3 is really interesting, but that the discussion in the current study misses the opportunity to capitalize on its uniqueness (I do not think that section 6.3 is sufficient in this respect). It looks to me like this is an excellent example of a nonlinear response / bifurcation leading to a real limit on predictability. Clusters 1 and 2 are simply phase shifts of the same cyclogenesis event. From a guidance perspective, both are reasonably useful at least in terms of situational awareness. Cluster 3, however, looks to me like the development of a different cyclone. There's an 850 hPa circulation south of Turkey in all of the groups at 1200 UTC 26 September (Fig. 8, column 2). In fact, a cyclone has already formed in this region in many of the cluster 3 members and one of the cluster 1 members (also shown in Fig. 9a). In groups 1 and 2, the low between Crete and Cyprus disappears as the PV tail promotes development along the African coast. In group 3, the pre-existing cyclone intensifies and fractures the PV streamer as the low retrogresses towards Crete (Fig. 8, column 3). By 1200 UTC 28 September, the medicane lies in the central Mediterranean in groups 1 and 2, but it is a completely different storm that is centered on Crete in group 3 (this differs from the interpretation implied by discussions on L286-L290 and L303-304 of the submission). So the relatively small difference in the location of the PV streamer axis (a linear change from west to east of the observed location) leads to a highly nonlinear response in the form of development of a new cyclone (groups 1 and 2) or intensification of an existing circulation (group 3). (Note that a couple of centers form over northern Africa in group 3 at 1200 UTC 27 Sept – Fig. 8k; these are cases in which the response to the change in PV streamer position is linear.) The theory that

group 3 is fundamentally different from the others is supported by the precipitation patterns and tracks (Fig. 9; noting that the large track jumps between North Africa and Crete are unlikely to be accurate) parcel trajectories (Fig. 10) and parcel properties (Fig. 11). Because a nonlinear response / bifurcation is known to impose strong limits on predictability, identifying and describing such behaviour in this case would be an important outcome of this work. I hope that some of the length reductions achieved by improving the manuscript's focus and organization can be invested in a much more thorough analysis of this possibility.

Many thanks for emphasizing the strongly nonlinear effects seen in cluster 3. We agree that this is an interesting aspect and discuss it in more detail in the revised version of the paper.

3. The values of QG vertical motion seem too small to be very meaningful despite being described as "strong" in the text (L441). Vertical motions of ~0.5 mm/s and 1 mm/s (0.005 and 0.01 Pa/s) are plotted in Fig. 2, while values of up to ~5 mm/s (0.05 Pa/s) are plotted in Fig. 7. These values are all well within the typical rms of QG vertical motion at midlatitudes and mean vertical motions across the globe (Stepanyuk et al. 2017). If these calculations and plots are correct, then the vertical motion forcing from the upper levels is almost irrelevant to the real vertical circulations in most cases. Such weak vertical motions would need to be sustained in-place for many hours/days to have any appreciable impact on moistening or stability. For example, air in the peak ascending region in Fig. 2c ascends <10 hPa in a day in response to QG forcing, an ascent rate that is dwarfed by the 600 hPa ascent in the rising parcels near the centre. If the calculations are correct, then the relevance of the PV streamer to ascent and cyclogenesis needs to be seriously reconsidered in this case, an exercise that will likely lead to conclusions that are completely different from those arrived at by the current submission.

It is true that the values of QG omega shown here are very small compared to the full vertical motion. However, it was not intended and is also not "fair" to try to explain the full vertical motion with the QG omega shown here. Note that we show the QG omega, as forced by levels above 550 hPa (QG omega top) on 850 hPa. These values are expectedly much smaller than the full QG omega on 850 hPa or the QG omega top on higher levels (see also Fig. 1b in Davies (2015), which shows the effect of an isolated forcing region on the vertical velocity field in the surrounding atmosphere). We therefore do not intend to explain the full vertical motion with this variable, but the goal is to show the presence and location of the upper-level forcing.

However, we reconsider if it is important to show QG omega in both Figs. 2 and 7. If QG omega is still needed, we will make sure to put the values into context and make them better comparable to the values found by other studies.

4. The motivation for the case study approach adopted by the study is weakened by passages that highlight case-to-case variability, and is not supported by a clear statement of the useful aspect of the case study framework. The dominance of case-to-case variability is particularly emphasized on L38 and L84, with the latter appearing to be a direct criticism of the case study as a useful

analytic tool. It is good to identify the limitations of the adopted investigation technique, but this criticism should be balanced with a clear description of what the case study approach can provide that other types of analysis (e.g. climatology) cannot.

We agree and now better explain the motivation for the case study approach.

5. The analysis of vertical coupling in section 5 is not quantitative enough to be included in the study. Despite significant discussion of Fig. 7e-h (L241-253), the strongest conclusion that is draw is that it is "most likely" that baroclinic instability is active. Even this conclusion appears to over-reach the analysis given that no baroclinic growth rates were computed. Given that the Icelandic low is not the focus of this investigation and that the left-hand column of Fig. 7 shows a convincing evolution of short-wave anomaly growth, I think that the right-hand column of Fig. 7 and the associated discussions should be removed. If this analysis is to be retained, then there needs to be a real quantification of baroclinic coupling and associated growth rates [note that the 12-18h time scale is very rapid for pure baroclinic growth, which typically has a doubling time scale on the order of a day (Hakim, Encyclopedia of the Atmospheric Sciences) and suggests that moist processes are likely to be very important].

We agree that in the original submission it was not clearly shown that baroclinic instability is active.

We change Fig. 7(e-h) to show (based on the operational analysis) the synoptic setting in which the error growth is occurring, without claiming that baroclinic instability is active. Additionally, we add a Figure and extend the discussion about upper-level dynamics related to the jet streak, which more convincingly explains the error amplification.

6. The study of PV error growth by Baumgart et al. (2018) is referenced in the introduction, but not in section 5, where the left-hand column of Fig. 7 bears a striking resemblance to Fig. 3 of that work (albeit with a compressed time frame). The discussion of the importance of non-linear upper-level Rossby wave dynamics here follows closely that of Baumgart et al. (2018), so much of this description could be replaced by citations and comparisons. The Torn (2015) normalized difference is a useful measure, so compressing section 5 to focus on that metric in the context of the Baumgart et al. (2018) interpretation of this process would allow for a dramatic shortening of this section and serve to place this submission in the context of investigations by other groups.

We agree that Baumgart et al. (2018) could be referenced and some sentences of this section could be shortened. However, we still think that Fig. 7 needs to be discussed well, also because (a) we show a different measure than Baumgart et al. (2018), and (b) as you point out, the time frame is very different (we show lead times 6-42 h in 12-hourly time steps, whereas Baumgart et al. (2018) showed lead times 2-8 days every 2 days). We add a reference to Baumgart et al. (2018) in this section and shorten the text where possible.

7. Assessing the significance of the differences discussed in section 5 is important; however, the technique and in-text descriptions should be revised. Wilks (2016) provides a description of problems with the multiple-testing technique (as adopted in this study), which can lead to over-confident statements about significance. Please consider using the false discovery rate here. Additionally, the level at which the differences are considered significant is not identified in the text, and only appears in the Fig. 7 caption (is 0.05 used throughout?). Note that there is currently a reconsideration of the use of the term "significant", which appears to be leaning in favour of providing p-values rather than definitive statements about significance. I'm not very familiar with that discussion, but it might be of interest to consider during revision.

Thank you very much for pointing out this problem and mentioning the Wilks (2016) paper. We agree that, as we are using multiple-testing, a p-value correction is required to control the false discovery rate. We will correct the p-values using the Benjamini-Hochberg correction that is suggested by Wilks (2016) and make sure it is clear which p-values are used in the Figures.

8. I am surprised not to see any references to Wiegand and Knippertz (2014), who study the representation and predictive skill of anticyclonic RWB and PV streamer formation over the Mediterranean region in the ECMWF ensemble (i.e. an earlier version of the same system used here). That work seems so directly relevant to this study (including the conceptual diagram in Fig. 10 of that paper) that it should be leveraged heavily in this investigation, particularly in terms of putting the forecast uncertainty in this case in a broader context.

We apologize for not referencing this important study in the original version. This paper is now included and helps putting our case study in a broader context.

9. The numbering of clusters forces readers to remember the mapping: 1 is centered, 2 is west and 3 is east. Why not call the clusters C, W and E? Then the Fig. 8 rows could be reordered to W, C, E so that there's a progression in the columns rather than having the PV streamer location (and eventual cyclone location) jumping around.

Thanks, very good suggestion, which we adopted in the revised version.

10. Throughout the study, the "surface cyclone" is discussed by the 850 hPa heights are shown. Showing 850 hPa winds is useful, but I don't see anywhere in the manuscript that the 850 hPa heights are essential to the analysis. I think that all plots that currently show 850 hPa heights should be replaced with mean sea level pressure for consistency with the text.

We agree that 850 hPa geopotential heights are not ideal and SLP would be more appropriate. The plots will be changed accordingly.

11. Throughout the study, short-range ECMWF forecasts are used to estimate precipitation accumulations. To avoid model biases and potential "twinning", it would be preferable to use an

independent product. The GPM IMERG is readily available and would be a better choice for this study than stitched-together IFS forecasts.

No precipitation product is perfect for our analysis. The short-range IFS forecasts have the advantage that they are from the same model as the other data used. GPM data might also not be free of biases. Since the exact precipitation values are not essential for our study, we continue using the short-range IFS forecasts.

12. Advection of cold air over warm Mediterranean waters is identified as a factor that increases latent heat fluxes and promotes convection; however, this effect is not quantified in the current investigation. The OAFlux dataset covers the period of interest and is readily available for this kind of study. Please consider supporting the claims made in the manuscript with an analysis of OAFlux (or equivalent) surface flux estimates. An augmented surface flux analysis may particularly interesting if model-predicted fluxes are found to be very different between groups 1/2 and group 3 (see item 2 above). Such an analysis is essential if the categorical statements about surface fluxes currently found in the conclusions (L429) are to be retained.

We agree that the argument, that latent heat fluxes are active was not well supported in the original submission. Because of the general restructuring and in favour of a clearer focus of the paper, we remove the trajectory analysis in the ensemble members and instead provide a trajectory analysis including surface fluxes based on the operational analysis in the supplementary material, which is briefly discussed in the synoptic overview.

13. The manuscript really needs to be clear about whether the medicane itself is a focus of the study. In multiple passages, it is stated explicitly that the medicane is not going to be investigated as part of this work (e.g. L97, L161, L320). However, much of section 6 is dedicated to the evolution of the medicane, including trajectory and CPS analyses. The title of the manuscript also emphasizes the storm morphology and will attract readers interested in medicanes. It feels as though the work was initially focused entirely on the PV streamer, and that "mission creep" has led to the introduction of more storm-scale-relevant material. Please reconsider the statements that disavow the relevance of the medicane structure for this work in an effort to remove what seems like a fairly important internal inconsistency in the manuscript.

We apologize for the confusion about the focus of the study, which created the impression of internal inconsistency. The original idea was to submit a two-part paper, where the medicane would be the focus of the $2^{nd}$ part. Obviously, when we decided to first focus on this paper, we didn't manage to clearly explain the role of the medicane aspect.

We will clarify this aspect better. In particular, we clearly define what we consider as a medicane in this study (and distinguish between medicanes that undergo tropical transition and the ones that don't). We make clear that in this study, we focus on the large-scale conditions that

influence the predictability of a medicane. And we also make clear that we do not focus on the meso-scale dynamics that – once a medicane has formed – can lead to tropical transition.

14. Why are the ECMWF data coarsened to 1°, and how is it done? The result is very poor resolution in the graphics, and if it not done carefully, the operation could result in aliasing. Is a conservative remapping used? This is a particularly important question for the precipitation field, where the difference between sampling/interpolation and remapping/aggregation can be enormous when the degradation of resolution is so large.

We downloaded ECMWF analysis data on a 1° grid to be consistent with the resolution of the ensemble data. Such a coarsening of the ensemble data is required to cope with the huge amount of data. Note that, for all ensemble members, we download the 3D fields on all model levels, which is required to accurately calculate PV and trajectories. This data transfer needs to be done within a few hours after completion of the ensemble simulation, because eventually, fields are archived in MARS on a few pressure levels only. All grid interpolations were done with routines available in MARS.

15. Most published works do not consider "medicane" a proper noun (and it is therefore not capitalized). This is analogous to "hurricane", which is only capitalized when a specific storm is discussed (e.g. "Some think that Hurricane Katrina was a category 3 hurricane at landfall"). Consider using lower case "medicane" throughout except in named reference to Medicane Zorbas.

Thank you for the explanation! We changed the use of "medicane" accordingly.

16. The terms "air mass", "airstream" and "parcel" seem to be confused in relation to trajectory analyses (L139 and section 6.2). An "airstream" is a loosely defined concept, but I think that it would be represented by a high density of air parcel trajectories in a limited area. Then the phrase "trajectories of the airstreams" (Fig. 10 caption) doesn't make sense unless the airstream (a feature in storm-relative coordinates) is somehow tracked over time. Similarly, trajectories do not track "air masses" (L139), but parcels. The difference is important, because it is unlikely that all parcels in an "air mass" are ascending near the cyclone centre.

17. The trajectory analysis in section 6.2 is incomplete. The suggestion that moistening is occurring because of surface latent heat fluxes (L345-346) implies that the parcels are in contact with the surface; however, the vertical position of the parcels is never shown. It is also possible for parcels to be moistened by evaporation of falling precipitation or by turbulent mixing. It is therefore not demonstrated that enhanced surface fluxes are responsible for the moisture changes in groups 1 and 2. The same is true for the potential temperature analysis on L346-348: surface fluxes are only one possible reason for potential temperature increases, and only influence parcels if they are in contact with the surface (even at above-surface levels in the boundary layer, the moistening/heating mechanism would be turbulent flux convergence rather than surface

fluxes per se). The lack of information about the trajectories makes it impossible for reviewers or future readers to confirm the validity of the conclusions drawn at the end of this section (L356-370).

18. Section 6.2 ends with a set of suppositions and conjectures based on an incomplete trajectory analysis (see previous item) climatological behaviour. As a result, terms such as "could favour" and "might support" are used instead of definitive statements. If the analysis and descriptions in this section cannot be made robust enough to be able to conclude these statements definitively, then this section should be removed.

Reply to comments 16-18: Thanks for pointing out these weaknesses of the analysis. As mentioned above, we remove this trajectory analysis in favour of focusing on the most important aspects of our study (see reply to comment 12).

19. The description of the CPS (L386-392) is insufficiently detailed to allow independent confirmation of the results (a requirement for publication). Because of the small scales of medicane structures, the hurricane-based radii are usually reduced for studies of Mediterranean storms. Was the same done here, or were the original hurricane-based values used?

Consistently with previous studies (e.g. Gaertner et al., 2018), we have used a radius of 150 km. This is now mentioned in the revised version.

20. I don't understand the "deep warm core" (DWC) analysis in Fig. 12. Take groups 2 and 3, for example. They have 12 and 18 members, respectively. The average number of DWC in group 2 is 7.2, and 7.0 in group 3 according to Fig. 12. That number is "per ensemble member", so multiplying by the relevant ensemble size yields 7.2*12=86.4 for group 2 and 126 for group 3. However, the total number of DWC steps for group 2 is given as 43, and that for group 3 is given at just 14 at the bottom of the plot. In the text (L404) the reader is told to consider the group-3 DWC analysis "with caution, due to the small sample size". However, the average number of DWC steps per ensemble member is as large in group 3 as it is in group 2: why is the sample size so small? There seems to be something about the number of sequential DWC steps ("duration"), but that is never clearly stated in the text or caption. What is wrong with my interpretation of the DWC analysis?

Thank you for pointing out that this analysis has not been straightforward to follow. The missing piece is, that the analysis only includes ensemble members that actually have a deep warm core cyclone. So, the number of DWC steps has to be multiplied by the number of members in the considered cluster that have a DWC cyclone. This number can be read from Table 1. Hence, the sample size for group 2 is small because only 2 members form a deep warm core cyclone.

However, we most likely replace the Figure with another one that is clearer and additionally provides evidence for conclusions that are so far not very well supported (see your comment 23).

21. Throughout the text, equivalent potential temperature gradients are used to identify both baroclinic zones and moisture gradients [L125, L132, L137 (where the 850 hPa theta-e is inappropriately used to identify a "weak surface cold front") L153 and elsewhere]. Strictly, neither of these is guaranteed by a theta-e gradient, which may arise as a result of either in isolation. If baroclinicity is important, then potential temperature (or temperature on an isobaric surface) should be shown. If moisture is important, then it should be shown. Theta-e is a very useful quantity for assessing convective potential and is a useful way to identify the warm sector for the trajectory analysis, but it does not replace the more basic fields for questions of baroclinicity and moisture.

We agree that it is more appropriate to look at potential temperature and humidity when the focus is on baroclinicity or humidity. We reconsider the importance of discussing baroclinicity and moisture gradients for the storyline of the manuscript. Where needed, we show potential temperature or specific humidity, respectively.

22. There are a lot of very specific geographical references throughout the text, probably more than there need to be. I'm a geographer, but I still found myself having to look for specific place-names on maps. It would be very useful to have a new Fig. 1 that shows (at least) the storm track and labels for all place names referred to in the text.

We agree with this point and include a new Fig. 1 that shows the storm track and labels for relevant places.

23. The conclusions of the study are not supported by the evidence provided in the text:

a. The "clustering" technique is not rooted in a mathematic definition and fails to guarantee the separation of the members into distinct "scenarios" as stated in the text (e.g. at L481). Is it true that there are three "distinct scenarios"? I agree that there are two (see General Comment 2), but I don't see why there are three. Groups 1 and 2 are distinguished only by the fractional overlap of the PV streamer, and there was no demonstration that there is any sort of heterogeneity in overlap space. This is a weakness in the analysis that results from the failure to use a true clustering analysis, and the decision to rely on a classification heuristic. There is no guarantee that group 1 and 2 events are separate from each other in any kind of meaningful way, and selection of a different overlap threshold (70%, for example) would result in the progressive reclassification of members from one group to the next. To demonstrate the presence of different scenarios, a true cluster analysis should be performed, and the optimal number of groups should be identified (e.g. using the "elbow method).

We agree that generally, and for climatological or operational purposes, an objective mathematical clustering is clearly preferable. However, the beauty of this specific case is that it is very clear to the reader that the uncertainty of the PV streamer prediction is large in terms of

the meridional position of its tip and its shape (both are somewhat connected) and that it is meaningful to separate the ensemble into a group where the meridional position of the PV streamer is more or less correct, another group that has the streamer too far to the east, and one that has it too far to the west. If you agree that groups 1 and 3 are different, then you could also agree that group 2 is different, because group 3 and group 2 are distinguished from group 1 in the exact same way (with group 3 having the PV centre of mass to the east and group 2 to the west of the analysis). We are very thankful that you pointed out the fact that cyclogenesis in group 3 shows a non-linear response to the PV streamer position (General comment 2), and that it shows also a clearly distinct surface cyclone scenario in response to the distinct PV streamer scenario. However, from the fact that the surface cyclone scenarios of group 1 and 2 are not as distinct as of 1 and 3 (albeit clearly shifted) we cannot conclude that there are no distinct PV streamer scenarios. We therefore argue that, in the present case, this kind of ad-hoc clustering is useful and suits the purpose of this case study. It is not at all guaranteed that an objective clustering provides more appropriate scenarios for this kind of investigation. For example, if we assumed that the overlap space is fully homogeneous and the PV streamers are shifted by constant distances from west to east, an objective clustering would not necessarily provide us with useful information how many clusters to choose. However, if we are interested in what the shift of the PV streamer does to the predictability of the surface cyclone, it would still be meaningful to split the ensemble into equal bins with one containing the westernmost PV streamers, one the more central PV streamers, and the third the easternmost PV streamer.

In the Zorbas case, we argue that the heterogeneity in the PV streamer distribution is captured very well by the ad-hoc clustering and an objective clustering technique is not required for this study.

We now add an inlet to Figure 4 (see below) of the initial submission which shows a histogram of longitude of the maximum PV value within the latitude band 36°-37°N. The three maxima in this histogram, representing each one of the clusters, hopefully convince you that – from the PV streamer perspective – there are three clearly distinct scenarios. Note that a clustering according to this measure would put a few boarderline members into another bin (see green and red "outliers"). However, we stick to the original clustering as this accounts for the uncertain positon/shape of the PV streamer on a larger domain.

[Figure]

b. There was no analysis of the near-surface flow induced by the PV streamer, so how is the conclusion about induced advection (L432-434) supported by the evidence provided in the submission? Particularly given the limited spatial extent of the streamer immediately prior to cyclogenesis, it is possible that the induced near-surface flow is very strong. For the arguments regarding air parcel modification by surface fluxes, the parcels approaching the centre in groups 1 and 2 must be in contact with the surface, putting them as far as possible from the upper-tropospheric streamer.

We agree that for this statement to be fully supported, a PV inversion would be necessary. From our general understanding however it is very sensible to assume that the PV streamer, as a strong upper-level PV anomaly, affects the lower-level circulation. Note that this conclusion is not about the air parcels modified by surface fluxes, but the ones that end up in the warm sector (which are never argued to experience surface fluxes, on the contrary they move almost adiabatically and without increase in specific humidity). However, as mentioned above, in response to above comments and for the sake of enhancing the focus of our study, we remove the trajectory analysis (see reply to your comment 12).

c. I cannot see what part of the analysis is used to conclude that the group-1 PV streamer was better able to "maintain the cyclonic circulation" (unclear whether this refers to the upper- or

lower-level flow) than the group-2 or group-3 features (L434-438). There appears to have been a rigorous analysis behind this statement (something that determines the number of members that meet a "condition"), but I don't know what section this analysis was described in.

We agree that composites may not be fully ideal/fair to draw this conclusion because spatial shifts could also result in low PV values in the composite. We therefore provide an additional analysis that better supports this statement and make sure the formulation is clearer. We also rephrase the conclusion.

d. The increase in the amplitude of the cyclonic PV anomaly from about -0.5 PVU to beyond -2.5 PVU (combined with a rapid areal expansion) over the 24-h period ending at 1800 UTC on 25 September (Fig. 7b and d) is "rapid" as stated on L442. However, as noted in item 4 above, this growth rate appears to exceed that expected for typical midlatitude baroclinic growth. It is highly likely that moist processes are involved, but because no estimates of growth rates are made in this study, it is impossible to know. It is therefore also inappropriate to conclude that the observed growth is "as expected from baroclinic instability" (L442) because the expected value remains unknown in the context of this work.

We agree that this conclusion has been a bit shaky. Note that the argument about baroclinic growth was (at least this was our intention) mainly made for the 6/12-h period from 12-18 UTC 24. Sept. After this time, we argued that non-linear barotropic dynamics is mostly responsible for the growth and downstream development. Still we agree that this is very rapid for (dry) baroclinic growth.

The reviews sparked further analyses of this aspect. As stated in the reply to General Comment 5 we will revise the analysis in Figure 7 and make sure the conclusions drawn from it are well supported.

e. It is unclear to me what part of the analysis demonstrates that "the contributions of diabatic airstreams ... were negligible for the uncertainty amplification in this case" (L444-445). The non-conservative evolution of the PV streamer was remarkable in this case (Fig. 2), and the impact of diabatic PV reduction in WCB outflow on ridge amplification during the upstream RWB (Fig. 7a-d) was not analyzed in the study, as far as I can tell. This statement about the role of diabatic process on forecast uncertainty (L444-445) is very strong, inconsistent with previous work, and needs to be clearly supported by the presented analysis.

We meant that, for the uncertainty amplification shown in Figure 7 (which is before the PV streamer forms), WCB outflow was not present in the vicinity of the region with strong error growth. This was based on a careful analysis of WCB trajectories, but not shown, because of the few WCB air parcels identified in the domain. Of course, you are right that the PV streamer evolution was non-conservative, but with our statement about the contribution of diabatic effects we did not aim to characterize the later times but the timesteps shown in Figure 7. Also, we

argue that it is not at all inconsistent with previous work, that direct modification by diabatic processes and diabatic airstreams are not very relevant for the amplification of forecast uncertainty in individual cases (see e.g. Baumgart et al. 2018). This of course, does not mean at all that we consider diabatic airstreams as generally not important for the amplification of forecast uncertainty. The attribution of the amplification of forecast uncertainties in Rossby waves to individual dynamical processes is a rather new research subject, but it seems that pure non-linear barotropic Rossby wave dynamics is very important (Davies and Didone 2013; Baumgart et al. 2018). Of course, diabatic airstreams can influence these dynamics, for example by placing a negative PV anomaly close to the wave guide. The absence of warm conveyor belts in the phase of the rapid error growth shown in Fig. 7 is a very strong indication that diabatic airstreams are not relevant for the uncertainty amplification in this case. However, this has not been discussed in the initial submission.

We now include the intersection points of WCB air parcels with 325 K in a revised version of Fig. 7 and/or in the synoptic overview. With this, we show that, even if the WCB was present at the later stage of the wave breaking, the few intersection points present in the early stage are far away from the region of the amplification and highlight this aspect when we discuss Figure 7.

Minor Comments
There are a relatively large number of grammatical errors in the submission, which I have not itemized here because of the major reworking of the text that will be required to address the issues identified above.

1. [L50] It is not clear why the introductory reference to parameterization uncertainty is useful here, where initial condition uncertainty is described in the subsequent passage. I would suggest starting this paragraph with "A major source ...".

2. [L51] I don't think that "slight uncertainties in initial conditions typically grow" (my emphasis), because the majority of uncertainties in any given analysis project onto decaying modes in the atmosphere (Privé and Errico 2013). I think that it would be more precise to say something like, "Slight uncertainties in the initial conditions that project onto the growing modes of the atmosphere can increase in amplitude during the forecast and potentially ...". You could also just replace "typically" with "can" in the current phrase.

3. [L87-L94] Suggest dropping this subsection in favour of the analysis in section 3.

4. [L95-L97] Having a clear set of objectives is a good idea, but these questions are framed in a way that is too complex to make them useful for the reader (e.g. "what is a and what of b leads to c and d in e"). Suggest simplifying or removing these questions.

5. [L99-L105] Provide a standard outline with section references.

Reply to comments 1-5: We adopt the suggestions in 2 and 5, and consider 1, 3 and 4 when rewriting the introduction (see your General Comment 1.a.i)

6. [L108] How are the ensemble members "perturbed": initial conditions, stochastic physics, SPPT, etc?

It is the standard ECMWF operational ensemble forecast. Because the details of how the ensemble is created are not relevant for the conclusions in this analysis, we will not go into detail here, but add a short statement clarifying this point.

7. [L111 and elsewhere] The word "data" is plural, so "data are available", etc.

Thanks, we change it accordingly.

8. [L115] What climatology is used for the ACC calculation?

It is daily mean Z500 values from ERA-Interim from 1979-2014. We add this information in the corresponding section.

9. [L122] Reference to a URL is inappropriate. At the very least, an access date needs to be provided. Consider including lightning strike information on the plots, rather than making reference to external information that may not be permanently available.

We add a map of lightning strikes to the supplementary material.

10. [L152] How is conditional instability identified in this analysis?

It is not directly identified. We remove "conditional".

11. [L159] Figure 4a does not exist and Fig. 4 is not the CPS.

Apologies for this typo, it should have been Figure 5a. We corrected it.

12. [L207] Reference to Fig. 7 is out of order.

Thank you, we corrected it.

13. [L221] At what level are the differences significant?

Always 0.05 is used in the initial submission. We make sure that it is always clear which threshold for the p-value is used in each figure.

14. [L231-L232] This sentence doesn't make sense: does the amplitude "propagate" at a different speed from the difference? Are you differentiating between phase speeds and group velocities here? Please rephrase to make this clearer.

15. [L263] The section title should be much clearer, and not read like a news headline.

16. [L274] Three different time references begin this sentence. Please determine whether it is the time relative to streamer extension, Gregorian date/time, or forecast time that is most relevant here and stick to this description of the first column of Fig. 8.

Reply to comments 14-16: Thank you for pointing out these unclarities. We consider them in the revisions.

17. [L278] I don't see that cluster 3 trough is "clearly" shifted to the east of the analysis at 1200 UTC 25 September (Fig. 8i). Instead, I see a trough that is too narrow, notably on the upshear flank over Germany.

Thanks for pointing this out. We agree and change the text accordingly.

18. [L281] Why isn't significance plotted here as in the first column?

For visibility reasons. In the later plots, large areas would be covered by the significance shading. The significance for all timesteps is shown in the supplementary material. We make the reference to the supplementary material clearer in the figure caption.

19. [L304-L305] This looks like more than just smoothing of the ensemble mean. Because averaging is a linear operation, the area-averaged ensemble-mean precipitation should match the observed values if the ensemble does not under-predict rainfall.

We do not agree, that area-averaged ensemble-mean precipitation should match the observed values. As long as the "observation" lies within the range of the ensemble, we don't think that there is an under-prediction. The area-averaged precipitation can be highly different between the ensemble members and as long as there are (even only very few) ensemble members that have equal or higher amounts of area-averaged precipitation than the "observed", the ensemble is fine (unless this is the case for most ensemble forecasts over many cases). We computed area-averaged accumulated precipitation for each member and the analysis over the study region and it shows that the value for the analysis is around the $90^{th}$ percentile of the ensemble.

We now add additional panels to Figure 9 that show the members with the highest and the lowest accumulated precipitation in each cluster in a box over the study region to illustrate the variability among members.

20. [L297] Are these SLP changes computed from the central pressures of the ensemble members, of from the ensemble mean? The search for a minimum central pressure is not a linear operation, so the results will likely be sensitive to the method. Particularly given the broad spatial distribution of group-2 centres, some/much/all of this apparent weakening may simply be the dilution of the ensemble mean if the ensemble averaging is done first.

Thanks for pointing this out. We computed it from the cluster mean. However, in the revised version we will compute the changes from the individual members and correct the numbers.

21. [L312-315] The four lines of hypothesis here would be much better invested in the actual analysis rather than this forward-referenced supposition (I recommend the removal of this whole paragraph as noted in item 1.b.iii above).

22. [L317 and L320-L321] There seems to be an internal inconsistency here. On L317 the objective of the section is stated to be "to investigates ... subsequent development of a medicane-like system". However, on L320-321 you state that you "do not identify low-level warm cores directly and do not investigate their formation in detail". Because the warm core is one of the primary structural ingredients that distinguishes medicanes from typical Mediterranean cyclones (considering the CPS), these two statements seem to be in direct conflict.

23. [L321] What do you mean that you don't identify warm cores directly? The CPS-based warm-core detection is the basis for a large part of section 6.3.

24. [L344] Do you mean a larger increase in specific humidity in groups 1 and 2, than in group 3? This sentence suggests the opposite, likely because the intended target of the pronoun "this" is unclear (although the construction suggests group 3).

25. [L353-L355] What is the physical relevance of this comparison?

26. [L393-L399] Why bother with a set of conjectures right before performing the actual analysis? A far more direct approach would be to explain why the fractions of medicanes in each group differ, based on the analysis presented in earlier sections. The conjectures do nothing to build suspense for the big "reveal" of Table 1, and just serve to consume five lines of text unnecessarily.

27. [L399-L401] This text contains every number shown in Table 1, without offering any physical insight. Choose to present these numbers either within the text, or in a table, but not both.

Reply to comments 21-27: Thanks for these helpful comments that highlight parts of the analysis and the text that require revisions. This part of the paper will be completely revised in the new version, considering these comments.

28. [L413-415] How is it concluded that "the detailed interaction between the surface cyclone and the upper levels become limiting factors" for predictability? Why can't internal storm processes or air-sea exchanges be the limiting factors? Those processes have not been investigated or ruled out as limits on storm structure predictability in this analysis, as far as I can tell.

We conclude that "sub synoptic-scale processes including the detailed interaction between surface cyclone and upper-levels become limiting factors". This does not exclude other sub-synoptic scale processes but maybe puts too much emphasis on the interaction of the surface cyclone and upper levels. We improve this sentence and also mention that internal storm dynamics/convection can be relevant.

29. [Fig. 6] Can the map resolution be increased a bit? (Similar in Fig. 7 zooms.)

Yes, we will do this.

30. [Fig. 6] At what level are the contours significant?

Again, 0.05. We make sure this information is clear.

31. [Fig. 6] The means are too similar to be usefully distinguished on the plot. Consider plotting the full ensemble mean only rather than solid and dashed contours.

We agree and change the Figure accordingly.

32. [Fig. 7] Is that a reference vector between (d) and the colour bar? If so, it should be highlighted and described in the caption. If it isn't, then one should be added.
Thanks, we adopt these suggestions.
33. [Fig. 10] Use a fixed domain to ease comparison between panels.

34. [Fig. 10] What does the colour-coding of the trajectories represent (the last sentence of the caption is not clear about what is indicated "in colors")? Are different members assigned random colours? Why are there fewer cyclone positions in the groups than members within the groups? Are there multiple trajectories ending at the same point because of the degradation of the grid resolution? If so, there should be some way to represent the number of overlapping triangles (potentially the size of the triangle).

Each ensemble member has a different color.

35. [Fig. 10] How does the maximum "percentage of ensemble members with an airstream occurring at the specific grid point" occur outside of the trajectory envelope? For example, the maximum departure frequency in Fig. 10b occurs poleward of any trajectory. Is it because these trajectories are actually averages of many trajectory calculations? If so, then there must be some unusual spatial distributions to obtain density maxima away from the means. How many trajectories are computed in each member?

The "average trajectories" represent means of several trajectories (~12, depending on the location) for each member. Therefore, it is possible that there is a density maximum (of the all actual trajectories) away from the starting position of the "average trajectory". We realize that this analysis is a bit confusing.

36. [Fig. 11] Why are radii the best way to identify the blue and green lines? It would be clearer to label the blue line "center" and the green line "warm sector" because the radii are technical details rather than relevant features.

**Reply to Comments 33-36:** As mentioned above, we remove the trajectory analysis to achieve a better focus of the paper and instead provide a trajectory analysis based on the operational analysis in the supplementary material (which we discuss in the synoptic overview).

References
Privé, N. C. and Ronald M. Errico (2013) The role of model and initial condition error in numerical weather forecasting investigated with an observing system simulation experiment, Tellus A: Dynamic Meteorology and Oceanography, 65:1, DOI: 10.3402/tellusa.v65i0.21740

Stepanyuk, O., Jouni Räisänen, Victoria A. Sinclair & Heikki Järvinen (2017) Factors affecting atmospheric vertical motions as analyzed with a generalized omega equation and the OpenIFS model, Tellus A: Dynamic Meteorology and Oceanography, 69:1, 1271563, DOI: 10.1080/16000870.2016.1271563.

Wilks, D.S., 2016: "The Stippling Shows Statistically Significant Grid Points": How Research Results are Routinely Overstated and Overinterpreted, and What to Do about It. Bull. Amer. Meteor. Soc., 97, 2263–2273, https://doi.org/10.1175/BAMS-D-15-00267.1

Wiegand, L. and P. Knippertz, 2014: Equatorward breaking Rossby waves over the North Atlantic and Mediterranean region in the ECMWF operational Ensemble Prediction System. Quart. J. Roy. Meteor. Soc., 140, 58-71.

**Reviewer 2 (Florian Pantillon)**

The paper investigates the large-scale dynamics that led to the formation of a tropical- like cyclone over the eastern Mediterranean in late September 2018, which was characterized by high forecast uncertainty in the operational ECMWF ensemble prediction system. A potential vorticity streamer issued from an anticyclonic Rossby wave breaking over eastern Europe was key in the Medicane dynamics. Two clusters of ensemble members with zonal shift of the streamer can be tracked along the Rossby wave guide back to initial conditions over North America. The evolution of the streamer into an upper-level cut-off low then controls the surface cyclogenesis, the stability and the advection of warm moist air that all support the Medicane formation.

Hybrid cyclones in general and Medicanes in particular are current sources of vivid discussions in atmospheric dynamics and objects of broad interest in the Mediterranean community. Contributions to better understand their dynamics and predictability are thus welcome and the paper presents interesting new material based on sound methods and high-quality figures. However, it suffers two major shortcomings: possible contributions from small-scale dynamics are largely ignored, although they at least partly explain Medicane formation, and the manuscript needs reorganization, as already pointed out by Referee 1. These shortcomings are linked somehow, as the tropical transition of the cyclone is actually assessed at the very end of the paper only. They are described below, as well as (many) specific comments.

The paper thus requires substantial revision before it can be considered for publication in Weather and Climate Dynamics.

General comments

Scales: as stated in the introduction, "the relative role of positive upper-level PV anomalies and air-sea interaction for the intensification of Medicanes is currently debated, as well as the question to which degree they are dynamically similar to tropical cyclones". The paper focuses on the synoptic scale and is based on model forecasts that do not explicitly resolve convection. This is fine but (1) the focus should be explicitly stated, (2) the limitation should be kept in mind throughout the paper and (3) the results should contribute to the current debate.

Organisation: as already pointed by Referee 1, the structure of the paper is unsatisfactory. Please better organize the Sections, make sure important concepts are introduced early in the paper (then stick to the terminology) and methods are described in the appropriate section, and avoid referring to later sections. In particular, show the warm core structure early in the paper, and comprehensively, based on the analysis for instance; in the present form the reader must wait until the last subsection of the last results section to learn the cyclone actually developed a warm-core structure.

Thanks for these comments. We obviously failed to clearly state the focus and intention of the study and the organisation of the Paper was not as appealing as we thought.

When revising the manuscript, we will make sure that the focus is clearly stated and kept in mind throughout the study. In particular we state the definition of a medicane as suggested by the current literature, even if it is debated. In particular, we mention that there are cyclones classified as medicanes that do not seem to have any tropical dynamics while others undergo proper tropical transition. We then define how we identify medicanes in this study (which is by the presence of a deep warm core). This allows for a clearer terminology, which we make sure is consistently used throughout the text. We will state clearly that the focus of the study is to investigate the predictability of the medicane in the early stage (the formation of the deep warm core) and not the later phase, when Zorbas acquires more tropical-like appearance. Specifically, we emphasize that we link the differing position of the PV streamer in the clusters with differing probability for the formation of a deep warm core

We now show the CPS of Zorbas based on the operational analysis early in the paper and make clear which part of the life cycle we are looking at and why.

We also provide a standard data and methods section, and extend the synoptic overview section.

Specific comments

Title: the position and depth of the PV streamer exhibit some uncertainty in the ensemble forecast but the streamer itself is not uncertain; the link between PV streamer and Medicane could be more explicit.

Thanks for pointing this out. We change the title of the paper, most likely as follows:

Medicane Zorbas: Origin of an uncertain potential vorticity streamer position and impact on cyclone formation.

Abstract
l. 3-4 This statement is not clearly supported. l. 5 "uncertain" is not properly used here (see comment on title).

We change the wording and make sure it is clear that mainly the PV streamer position/shape was uncertain.

l. 7-8 "demonstrated", "the dominant source": not necessarily. See comments below.

Thanks for this comment. See replies to your comment below.

l. 9 Twice "strong(ly)".

l. 12 More details about the two air streams and their key role?

Reply to above two comments: Thanks, we rephrase and reconsider the content abstract after the revisions.

1 Introduction
l. 19-25 All references relate to the North Atlantic, which should either be explicitly stated or extended to other oceanic basins.

We add two more references of studies in other ocean basins.

l. 19-20 ET could also be mentioned here.

We agree that ET is an important process, but not really essential for this study. Therefore, we decided to not mention ET.

l. 26-33 It is unclear what is the difference between subtropical, tropical-like and hybrid cyclones, if there is one at all. And do not they by definition undergo tropical transition?

We agree that the wording is unclear here. We now try to make a better distinction between the different terms in the introduction.

l. 29 "air-sea feedback" is not precise enough.

We make this more precise by mentioning the WISHE mechanism that becomes active, when tropical transition occurs.

l. 32 This "may" result in high damage, as Medicanes often remain over sea.

l.35-39 Confusing what "they" and "their" refer to.

l. 41-42 Forecast uncertainty and the link with process understanding and ensemble forecasting needs better introduction.

l. 42-49 This appears too early and several keywords are not introduced yet (Zorbas, warm core, practical predictability, . . .).
l. 50 The transition should be smoother between 1.1 and 1.2.

l. 63 Lamberson et al. studied "extratropical" cyclone Joachim.

l. 64-65 The link between the predictability of breaking Rossby waves and Medicanes is far from obvious and need more details; it was extensively explored for a case study in September 2006: Chaboureau, J. , Pantillon, F. , Lambert, D. , Richard, E. and Claud, C. (2012), Tropical transition of a Mediterranean storm by jet crossing. Q.J.R. Meteorol. Soc., 138: 596-611. doi:10.1002/qj.960

Pantillon, F., Chaboureau, J., Lac, C. and Mascart, P. (2013), On the role of a Rossby wave train during the extratropical transition of hurricane Helene (2006). Q.J.R. Meteorol. Soc., 139: 370-386. doi:10.1002/qj.1974

Pantillon, F. P., J. Chaboureau, P. J. Mascart, and C. Lac, 2013: Predictability of a Mediterranean Tropical-Like Storm Downstream of the Extratropical Transition of Hurricane Helene (2006). Mon. Wea. Rev., 141, 1943–1962, https://doi.org/10.1175/MWR-D-12-00164.1

l. 67 Who are "they"?

l. 67-69 The mentioned studies do not clearly attribute forecast busts to initial uncertainties rather than to the representation of diabatic processes. Both error sources would thus better be described together here.

l. 79 Which process?

l. 86 Even if only few case studies exist, there are certainly more than the two cited here.

l. 89 Some basic information about the case study are needed here (what, where, when). And where does the name "Zorbas" come from?

l. 91 Please explicit "short lead times".

l. 94 Which PV streamer? Either detail or remain general; referring to Section 3 does not help. l. 96 "leads" or "led" l. 100-106 Detailing Sections 2, 3, 4, . . . might be required.

Thanks for all the above comments that point out unclarities, inconsistency, and missing depth in the introduction. We completely revise the introduction and consider these valuable points.

2 Operational ECMWF products
l. 110 Why 46 members only? What is the operational short-term forecast?

For technical reasons, four ensemble members were missing in the data we downloaded operationally. And as this forecast is not in the core of the study, we thought that 46 members were sufficient. The operational short-term forecast are based on the first 12 hours of forecasts started at 00 UTC and 12 UTC each day and are used to get an estimate of actual precipitation. We now include all ensemble members of this forecast and explain better what we mean with operational short-term forecast.

l. 112 Forecast data is actually available at higher frequency.

This is true, but not with the full vertical resolution which is needed to compute potential vorticity appropriately. Note that we download ensemble data on full model levels right after forecast completion because much fewer (pressure) levels are in MARS.

l. 115 What is the reference for computing ACC?

It is daily mean Z500 values from 1979-2014. We add this information in the corresponding section.

3 Synoptic overview
Figures 1-3 Zooming in on the region of interest would be very helpful to follow the discussion. Large-scale dynamics play an important role but, e.g., the Irish and Red Seas are not relevant. Consider then merging the three figures to avoid jumping from one to the other.

l. 135-136 The spiral-like structure is hardly seen. Or do you mean the frontal band? Spiral often refers to a tropical structure. Again, zooming in would help.

l. 139 Fig. 2C

l. 140-141 Move to the methods section.

Thanks, we put methodological aspects into the methods section.

l. 144, 146 What are "they"?

l. 139-149 Is it convection and/or large-scale ascent? The 600 hPa in 24 h criteria suggests the latter, while lightning suggests the former. The ECMWF model cannot actually resolve convection but you could check whether the precipitation is issued from the convection scheme or not.
We agree that the 600 hPa in 24h ascending air parcels are not ideal to show here.

We quantify the contribution of convective precipitation and mention it in the text. Additionally, we show lightnings in the supplementary material

l. 135-149 There is a general confusion in the paragraph between what was stated by previous studies and what happens here.

l. 152 Fig. 3d

l. 152 Clarify the precipitation is from model data; using different colors would help distinguishing the −11, 8, 15 and 21 mm (6h) contours on Fig. 1.

l. 153 How can you know it is due to conditional instability?
We do not check directly what type of instability occurs.

We remove the "conditional".

l. 156-157 How do you discern a warm seclusion from a warm core?

l. 157-160 and 166-167 There is not enough evidence at that point to claim a tropical transition. Relying here on Sec. 6.3 is not a good idea and there is no Fig. 4a. Either show more details or keep for later.

Thanks, we agree.

As commented above, we now show the CPS early in the paper and make sure we have a consistent wording.

l. 162 This would be worth showing!

This paper doesn't aim to deal with the tropical transition of Zorbas and discussing the eyewall formation would be clearly beyond the scope of this study. For the sake of focus, we don't show this aspect of Zorbas and leave it for further studies.

**Reply to all above comment for section 3 with no direct replies:** Thanks, these comments are all valuable to improve the text and/or figures. We consider them when revising the paper.

4 Ensemble clustering according to position of PV streamer
l. 170-172 The sentence presents essential information but needs more support: why 00 UTC 27 Sep? Why 00 UTC 24 Sep? Is it perhaps the combination of valid time and initialization time resulting in largest spread? Can we see this somewhere? "Shown in Fig. 2 six hours earlier" is not too convincing.

These times were chosen because the PV steamer position showed this "nice" tri-modal behaviour. We add a sentence that states the motivation for these initial and valid times.

l. 172-174 Referring to a later Section to motivate the present one is surprising.

l. 174-176 Please move technical details to the Methods section.

l. 178 Average of 320 K and 330 K levels or are there additional levels in between?

It's every 5 K, so we average 320,325,330 K.

We make sure this is clear in the revised text.

l. 178 Why set PV < 2 pvu to zero?

This was done to "mask" out the troposphere and just use stratospheric PV air when averaging PV vertically. This gives a field that is sensitive to the depth and PV values within the PV streamer. For example, we get the same values if the PV streamer is present just at one level with a PV value of 6 PVU or at three levels with each PV values of 2 PVU. Another way to look at it is that we want to cluster the ensemble members according to the PV streamer, and therefore reduce the contribution of tropospheric PV values to the averaged field.

l. 180 Remind there are 50 members?

l. 190-191 Again, referring to a later Section is surprising.

l. 192-193 "Decrease" rather than "drop"? Are these values of ACC particularly low? And why not color clusters 2 and 3 in green and red on Fig. 5 as on Fig. 4?

Thanks for this suggestion. ACC values are not particularly low. We did not color clusters 2 and 3 because the focus here is to show the better performance of cluster 1. If clusters 2 and 3 are added the plot is less easy to read. We adopt the suggestion to use "decrease" but decided to not colour clusters 2 and 3.

l. 198 Errors in the shape of the PV streamer have not been discussed yet, only the zonal shift.

Thanks for this comment. The shape is somewhat included in the sense that if the shape of the PV streamer is completely wrong, the overlap would be too low to satisfy the criterion for cluster 1. The overlap can be low because of a shift or because of a different shape, or of course, a combination. The member that has been excluded from the analysis because it did not fit into a cluster actually shows no zonal shift at the tip of the streamer but a very special shape, such that the overlap is not large enough.

l. 199 Which characteristics are relevant?

l. 200-204 ACC may not account for the cyclone at all, at least the link is not showed yet. The link with the PV streamer is not obvious either. Consider adding Z500 on one of Figs 1-3.

Thanks for this comment. We agree that Z500 may not account for the cyclone very much. But it is reasonable to assume that Z500 is linked to upper-level PV, which is what is needed for the argumentation in this section. We now add Z500 in the synoptic overview.

**Reply to all above comment for section 4 with no direct replies:** Thanks, these comments all contain valuable suggestions to improve the text and/or figures. We consider them when revising the paper.

5 PV streamer scenarios emerge from initial condition uncertainties and baroclinic amplification l. 205-206 The jet streak has not been mentioned before. Consider either adding the large-scale dynamics leading to the PV streamer in Section 3 or, at least, shortly describing these dynamics here and motivating why they are the focus of the following analysis.

This is a good idea. We now include a Figure that provides an overview of the large-scale dynamics leading to the PV streamer.

l. 209-220 Please move to the Methods. Can you say some words about delta PV values, e.g., is there a threshold that indicates bi-modal distribution?

Delta PV values are just normalized differences and say something about how different the clusters are relative to the ensemble spread. We move this part to the methods and make sure the reader understands the meaning of delta PV values.

l. 221- 223 While the normalized PV difference is clearly highlighted, PVU and wind contours barely differ between clusters and are not discussed at all.

We agree and use ensemble mean or analysis wind contours now.

l. 230 The separation between clusters is hardly seen at that point.

We agree and only mention the PV difference in the text at this point.

l. 238-239 Remind Fig. 7d; better "stronger anticyclonic wave breaking" than "westward phase shift and larger amplitude"?
.
l. 242- 254 The description of Fig. 7e-h is difficult to follow and Fig. S1 suggests that differences in omega and Z850 are hardly significant in the region of interest. Consider removing altogether.

Thanks for these comments. We also received comments regarding this Figure from Reviewer 1. We completely revise Figure 7, especially the right hand panels, where we will present analysis fields with the goal to show the synoptic situation in which the error amplification takes place.

l. 244, 250 Either show or omit potential temperature.

l. 255- 263 This interpretation is meaningful and consistent with displayed material overall but (1) the formulation partly sounds speculative ("strong QG forcing", "uncertain low-level wave", "exponential growth", "strong vertical coupling") and (2) ensemble members differ not only in their initial conditions but also in their physical parametrisations or any other perturbations implemented by ECMWF to increase ensemble spread.

Thanks for this comment. Regarding point (2) we agree that the ensembles also differ in their physical parametrisations and errors might come from there but we can still see an initial condition uncertainty (at lead time 0, Figure 6) that seems to propagate into the North Atlantic and amplify there. We therefore argue that some of the uncertainty most likely comes from initial condition uncertainty. We will thoroughly revise this section and the analysis in Figure 7 and make sure our conclusions are less speculative and we acknowledge more clearly that we do not exclude contributions from model uncertainty.

**Reply to all above comments for section 5 with no direct replies:** Thanks, these comments point out unclarities or suggest an alternative wording. We consider them when revising the paper.

6 Diverging synoptic development impacts Medicane predictability
l. 268 What is a "Medicane-like" cyclone?

We will make sure our terminology is clearly defined in the introduction and consistently used throughout the text.

l. 270 The subsection provides a synthetic summary of the dynamics of all clusters, but what is the variability between members within each cluster?

We agree that the variability between members is not shown (except for the cyclone locations). We understand the goal of making clusters to reduce the information from all members into scenarios. The statistical significance test allows to draw the conclusion that the member within a cluster are really different from the members in another cluster. We now add information regarding the variability in cyclone intensities, their formation and upper level PV, most likely in the section where the results from the CPS analysis are presented.

l. 275-276 Mention the analysis PV is depicted by the black contour.

l. 277 What is meant by "exactly the ones"?

l. 278-279 There is no visible difference in PV between clusters in Fig. 8 a, e, i.

On the western side of the trough (marked by the p-values) slight differences are discernible. The point here is that the differences are maybe still small when comparing the full PV fields, but (as

shown with the PV difference plots) they are significant and they propagated into the trough from upstream.

l. 288 Fig. 2c; slightly different time.

l. 288-291 Mention the cyclone in individual members is depicted by dots.

l. 295 Differences are substantial but not necessarily due to latent heat release (only).

We agree that the differences in the PV cutoff evolution are not necessarily all due to latent heat release. However, if PV is eroded in cluster 1 and not in cluster 3, and as erosion of PV cutoffs is known to be related to substantial latent heat release, this is a clear indication that cluster 1 experiences more latent heating or at least, a different one (i.e. one in the vicinity of the PV cutoff).

l. 297-298 Clarify these are mean values.

l. 305 Fig. 9d

l. 306-306 The smoothing effect due to averaging makes the comparison difficult for precipitation; how do individual members look like? You could e.g. compute PDFs of accumulated or instantaneous precipitation for each member and the analysis.

Thanks for this comment. We will include a Figure and add a sentence that provides information about the internal variability within the clusters to make them better comparable to the analysis.

l. 314-316 These arguments are too speculative and are better left to Sec. 6.3.

l. 318 Again, what is a Medicane-like system?

l. 318-325 This paragraph is confusing and must be rewritten/streamlined. How do you define a low-level warm core, a warm seclusion and a Medicane-like system, and why do you focus on two air streams?

l. 326-327 The analysis tool Lagranto belongs to the methods and is already mentioned above.

l. 330 Is the warm core formation shown somewhere?

l. 345 "weaker" not "stronger" increase.

l. 346 In clusters 1 and 2.

l. 347 The Mediterranean Sea is not an ocean.

l. 353-356 This would likely better fit at the beginning of the paragraph.

l. 363 Closer to the coast but the region remains the eastern Mediterranean.

l. 363-371 The discussion is speculative so far, as the cyclone thermodynamics have (still!) not been documented yet. There is also a general confusion between warm core, warm seclusion, warm sector and tropical structure.

l. 377-394 This all belongs to the Methods. What radius is used to compute CPS metrics?

l. 394-399 I expect clusters 2/3 to show more favourable low-level/high-level forcing but not necessarily to produce a stronger/weaker Medicane.

l. 400 Avoid introducing an additional name ("DWC"), which adds confusion, better stick to the terminology used up to that point.

l. 400-402 What about the two other CPS metrics, symmetry and low-level warm core? The upper-level warm core metric might be contaminated by the presence of the PV streamer/cutoff. And what about the cyclone intensity?

Thanks for this comment. Although it could be worth looking at these additional aspects, the focus of this study is to investigate the factors affecting the formation of the deep warm core in Zorbas, which is a major characteristic of medicanes. The low-level warm core is indirectly included in the sense that it is a necessary requirement for a deep warm core. However, as the upper-level warm core is a distinguishing factor that separates so-called "medicanes" from subtropical cyclones, it is in the main focus. Regarding parameter B, this is a measure of the frontal nature of the cyclone, but analysing the frontal structures of Zorbas is beyond the scope of this paper. In order not to extend the analysis, we have decided not to analyse those parameters. However, we now additionally analyze the maximum cyclone intensity in each cluster.

l. 403-404 But cluster 3 produces stronger upper-level warm cores than cluster 2, which contradicts the other results and interpretation.

This is true, but cluster 3 produces only 2 cyclones with an deep warm core, so this average has to be taken with caution. We revise the deep warm core analysis and now show individual members rather than box plots to make the clusters better comparable.

l. 407-409 The three-day long sustained deep warm core (Fig. 5a) appears unprecedented. Can you provide CPS diagrams for the analysis?

Yes, we will provide the CPS of the analysis in the introduction.

l. 412-413 Why?

l. 414-416 What about convection?
We agree that convection can also be a relevant sub-synoptic scale process. This sentence was not meant to exclude other factors. We make this sentence clearer and mention that also internal storm dynamics/convection can be relevant.

**Reply to all above comment for section 6 with no direct replies:** Thanks, these comments point out unclarities or suggest small structural or content changes of the text. We consider them when revising the paper. Especially we will make sure that the terminology is clearer.

7 Conclusions
l. 427-418 Again, what is a subtropical cyclone, a tropical-like system or a Medicane?

l. 431 More details about this "first case"?

l. 432 Which process?

l. 435-438 This is not shown here.

l. 445-446 How do you know this?

It has also been pointed out by reviewer 1 that it is not clear how we arrive at the conclusion that diabatic air streams were not relevant for the uncertainty amplification. We will support this point better with the analysis (Figure 7) by providing more information about warm conveyor belts and precipitation.

l. 446-447 Which process, baroclinic instability?

l. 455-457 Ensemble forecasts are computed with different perturbation methods thus the error growth cannot be attributed to initial conditions only here.

Thank you for this comment. As already commented above, we agree that model errors can contribute to errors in this case. However, we show that a patch of uncertainty is present at initial time of the forecast that then propagates over the North Atlantic where the amplification takes place in its vicinity. Even if we cannot exclude model error here, we argue that the analysis shows strong indication, that initial condition uncertainty was very relevant in this case.

l. 458-460 . . .and convection and its organisation.

l. 462 As the used data is from ECMWF essentially, it could be stated how to access it.

The data used is not available long term from ECMWF with this high vertical resolution (which was required to compute PV and trajectories). We downloaded this data immediately after the event occured

**Reply to all above comment for section 7 with no direct replies:** Thanks, these comments point out unclarities or inconsistencies in the conclusions. We carefully consider them when revising the paper.

References
Providing DOIs or URLs for all papers would be helpful
Thank you for this comment. We will provide DOIs.

Figures
Moving all figures to the end of the paper would ease the review.

In the submitted document for discussion they are all at the end of the paper.

Fig. 5: it is unclear which date relates to which tick mark.

Fig. 7 appears before Fig. 6.

Fig. 6: consider changing the color scale to [-1,5; 1,5] and plotting coast lines at higher definition.

Fig. S1: the title should refer to Fig. 7 not 6.

**Reply to all above comment regarding the Figures:** Thanks, for these comments. We consider them when revising the paper.

**Reviewer 3**

This interesting paper deals with the predictability of the Medicane Zorba, which affected the central Mediterranean in September 2018. ECMWF ensemble forecasts are used in this effort. The limit in the predictability of the cyclone is analyzed and discussed in connection with the upper-level PV, which also affected the low-level evolution.

This is one of the first paper that clearly identifies the relevance of PV features in the predictability of Medicanes. The results are a relevant contribution in the field; however, the analysis should be substantially improved. Some points are indicated hereafter.

Major points:
Line 27, Line 154-157: I would like to see some clarifications about the definition of Medicanes. Although there is no general consensus, in most of the literature (e.g., Miglietta et al., 2011; Picornell et al., 2014; Cavicchia et al., 2014), a Medicane is considered as an extra-tropical cyclone that acquires a symmetric, deep warm core in the Mediterranean region. At the same time, the presence of a deep, warm core is not always an indication of tropical-like processes going on: as discussed in Fita and Flounas (2018) and Mazza et al. (2017), a deep warm core is not necessarily associated with a WISHE mechanism, but it can also be induced by a warm seclusion. However, Miglietta and Rotunno (2019) have shown that the intensification of the same cyclones discussed in the latter two papers cannot be explained without considering the sea surface fluxes and the latent heat release, in analogy with the WISHE mechanism typical of tropical cyclones. For these reasons, I suggest to remove "occasionally" (Line 27) and the sentence "Warm seclusion have been previously linked to Medicane formation" (Lines 155-156). Also, at Line 317 and 321 you "investigate potential pre- cursors of a low-level warm core": however, the low-level warm core is not relevant for the following development of the cyclone in itself (see also Line 365, 396), but because of the high values of equivalent potential temperature that are responsible for potential instability and favor the development of convection at later times.

Thank you for these explanations. We will make sure that in the revised paper we have a clearer definition of medicanes and that the subsequent argumentations are consistent with this definition.

Figure 10: I found the understanding and interpretation of Figure 10 quite difficult; in particular, I did not understand if the back trajectories you show are averages over all the ensemble members, since this is not mentioned in the Figure caption and not clearly reported in the text (Line 334); also, the presence of high percentages far from the plotted trajectories (purple shading) is counterintuitive.

Line 350-355: for a more comprehensive analysis of the trajectories in Fig. 10, some information should be included about the change of height along them.

Thanks for pointing out these unclarities in Figure 10 and the related analysis. As the other reviewers were also critical regarding this analysis and because it is not the centrepiece of the paper, we consider removing this analysis in favour of enhancing the focus of the paper. We provide a trajectory analysis based on the operational analysis in the supplement.

Minor points:
Line 115: please provide the definition of ACC

Yes we can do this.

Line 130: red instead of blue

Lines 143-145: the role of upper level PV anomaly in the generation of Medicanes is also discussed in Miglietta et al. (2017)

Thanks, we now include this reference.

Line 201: were instead of is

Line 202: severely instead of severly

Line 207: please can you provide an approximate indication of the height the isentropic = 325 K corresponds to?

Isentropic surfaces are generally not horizontal, they can intersect the surface towards the equator and be in the stratosphere at the pole. We will provide information about the approximate pressure level of the 325 K in the specific location over the North Atlantic that is discussed in this part of the text.

Line 214: why do you give less weight to the regions of strong gradients?

The idea of standardized anomalies is to quantify how different two clusters are, relative to the ensemble spread at a specific location. Just looking at absolute PV differences can result in high values just because the spread is high. But in this case, we are interested in regions where the clusters start to separate within the ensemble. As a result, regions of strong gradients are usually given less weight, because that's where the ensemble spread is usually large.

Line 221: explain why "stratospheric" side

This is because we use the 2 pvu contour to separate stratospheric and tropospheric air masses. The PV difference is located poleward of the 2 pvu contour and of the jet streak which is a region of stratospheric air.

Line 311: favorably instead of favourable

Line 317 and elsewhere: Medicane or tropical-like, not Medicane-like!

Figure 5 caption: why do you use only 46 members for the second ensemble?

Because this is what we downloaded operationally and for technical reasons, 4 members were missing in this forecast.
We now include the missing 4 members.

Figure 6: change contour line colors to facilitate interpretation

Figure 7 caption: please indicate that the black contour refers also to captions (a-d)

Figure 8 caption: (e,i) instead of (e,f); the black contours around the teal patches create confusion

**Reply to all above comments with no direct reply:** Thanks, for these comments that show wrong spelling or suggest Figure improvements. We consider them when revising the paper.

Supplement material, Line 9: "The results show that significant differences of QG are located in the region of strong QG on 1800 UTC 24 Sep 2018": it does not seem to be the case, at least at that time.

There are some patches of significant differences of QG omega in the region where QG omega is high. But we agree that this argumentation and analysis are a bit shaky. Also as response to comments from the two other reviewers, we revise the analysis shown in Figure 10. Most likely, in this new analysis we will not show QG omega anymore.

REFERENCES:
Cavicchia, L., von Storch, H., Gualdi, S. (2014). Mediterranean tropical-like cyclones in present and future climate. J. Clim. 27, 7493–7501.

Miglietta, M.M., Moscatello, A., Conte, D., Mannarini, G., Lacorata, G. and Rotunno, R. (2011). Numerical analysis of a Mediterranean hurricane over south-eastern Italy: sensitivity experiments to sea surface temperature. Atmospheric Research, 101, 412–426.

Miglietta, M.M., Cerrai, D., Laviola, S., Cattani, E. and Levizzani, V. (2017). Potential vorticity patterns in Mediterranean "hurricanes". Geophysical Research Letters, 44, 2537–2545.

Picornell, M. A., J. Campins, and A. Jansà (2014). Detection and thermal description of Medicanes from numerical simulation, Nat. Hazards Earth Syst. Sci., 14, 1059–1070.

---

## Referee Report (RR1)

**Review of "Medicane Zorbas: Origin of an uncertain potential vorticity streamer position and impact on cyclone formation" by Portmann et al.**

**Background**

This revised submission documents the prediction of Medicane Zorbas' development in the ECMWF ensemble initialized approximately three days before the event. The authors identify differences in member solutions depending on the position of the PV streamer that strongly influences the cyclone life cycle. They suggest that the differences in the streamer location over the Mediterranean are related to differences in member initial states over the Gulf of St. Lawrence. The authors also show that Zorbas' structure is sensitive to the position and structure of the PV anomaly as the storm intensifies.

The subject of the study remains very interesting and relevant. The major changes made by the authors have made certainly improved the analysis and presentation of the results. However, there are a number of issues with this version of the submission that remain to be addressed. I hope that these comments will be useful to the authors as they further revise this work.

Reviewer: Ron McTaggart-Cowan
Recommendation: Major Revisions

**General Comments**

1. The main body of the text (Abstract through Acknowledgements) is pushing 10 000 words, which is >30% longer than a standard submission. This makes the study a very long read, especially for anyone who also wants to dive into the supplemental material (a further seven pages, although mostly figures). This length combines with a persistent lack of flow to make the submission a very long read. This is unfortunate, because the subject of the manuscript is interesting and it would be a shame if paper did not get the attention that it deserves simply because it is difficult to get through. Consider the sections: "Zorbas life cycle", "PV streamer clusters", "initial differences", "PV streamer impact on development", "PV streamer impact on structure". The ordering of this set of section descriptions does not appear to follow any particular logic, leaving the reader to bounce around in both time and spatial scale (two useful ways that the content could be ordered). I think that reordering the manuscript would let the authors eliminate redundancies to reduce the length of the paper by 25% or more, a change that would be of real benefit to the study. An alternative that is discussed more in General Comment #6 is to remove section 5 completely (to be further developed into an independent study), a change that would limit length and improve the flow substantially.

2. As noted in my first review, I think that the manuscript tries to cover too much ground. This is exemplified by the main objectives of the study (L94-96), which expand quickly from two to four when the itemized list is broken apart into separable elements. As in the initial submission, it is noted that the storm's tropical transition "is not the focus of this study" (L188); however, the CPS (section 2.2) is used only to classify core thermal structure in different members, an exercise to which section 6.2 is entirely devoted. This kind of internal inconsistency makes it seem as though the manuscript is in the midst of an identity crisis, leaving the reader with without a clear picture of the true objectives of the work. One component of the analysis that could be dropped is section 5. Although I think that identifying the source of the uncertainty is

very interesting and possibly important, it adds significantly to the volume of information presented in the text and lacks the depth required of such an analysis. Removing this section would drop almost 2.5 pages out of the submission and free up figure space for images currently found in the supplement. It would also help to keep the focus of this manuscript on Zorbas itself, rather than the state far upstream.

3. I understand that you disagree with me on the formulation of a paragraph. However, I really do think that this manuscript would be very well served by reducing paragraph lengths throughout, such that each paragraph deals with a single subject. This will help to manage the breathless pacing of the text.

4. I don't understand or approve of the use of model precipitation as "truth" (L114-115), particularly when observational estimates are readily available (i.e. GPM). All modelling systems have precipitation spin-up problems, making the use of 0-6h forecasts particularly problematic. These data are degraded to a 1o grid, hopefully using conservative remapping; however, this is unclear in section 2.1. On L482-L84, the weaknesses of model precipitation estimates are highlighted. This acknowledgement of model QPF shortcomings (especially in convective cases) is inconsistent with the use of short-range forecasts as "truth". Please use an independent, observation-based QPE as the reference for this study.

5. Why use the full standard deviation in the denominator of Eq. 1? If this were a test statistic (it looks a lot like the two-sample t statistic), then the denominator would be the pooled standard deviation (i.e. the sum of the individual sample variances). I understand that the form in Eq. 1 has been used before in the literature, but if you consider two tightly clustered samples with small internal variances and a large difference between the sample means, then use of pooled variances will make the normalized difference much larger than that estimated by the current formulation. This may serve to highlight regions of systematic differences between the classes more effectively. Such a change will also bring Eq. 1 in line with standard assessments of differences between groups by comparing between-group differences with within-group variability (i.e. ANOVA).

6. I think that section 5 should be removed from this study and form the basis of a separate piece of work instead. The analysis of the error source is potentially very interesting, but (a) distracts from the discussion about Zorbas, and (b) is not rigorous enough for inclusion without a lot of additional investigation. Terms are dropped from the equations in this section without any quantitative justification: simply stating that terms are likely to be unimportant is not acceptable. The fact that the reduced form of the equation is then used to "imply" something about the relevant dynamics is specious reasoning (L324-325 and L331-333). Similarly, the decomposition into geostrophic and ageostrophic winds for some of the terms in Eq. 2 (becoming Eq. 3) is not physically justified, any more than a decomposition into any other vector components would have been (L325-327). The standard decomposition of the wind field under such conditions would be into nondivergent and irrotational components. The secondary circulation of a jet streak also has a strong projection onto the divergent wind, so it is not clear why geostrophy is preferable here, particularly given that it is related to a lower order of balance. The choice is important, however, because the current decomposition is used to conclude that "the described amplification [is] due to the ageosterophic circulation" (L500). I do not agree that this conclusion follows from the analysis presented in section 5. Indeed, the current analysis does not really serve to advance the main objectives of this study. A full study that looks carefully at the PV error growth in this case would be a nice follow-up to the current submission.

7. Throughout the text, the authors refer to vertical alignment of PV features as implying a "barotropic" structure. In a barotropic environment, there isobaric and isothermal surfaces are aligned, so there is no vertical wind shear. Given that wind shear is being used to diagnose core

structures in this study, I believe that the authors consider this to be an "equivalent barotropic" atmosphere.

**Specific Comments**

1. [L22] Suggest "increasing" to "intensifying".
2. [L27-30] This TT intro is too cursory to be useful to anyone not familiar with the concepts. Given that section 6.2 is dedicated to warm core formation, I think that this description should be improved.
3. [L30-31] The TT process involves far more than WISHE.
4. [L37] Suggest "rather" to "more".
5. [L40] Note smaller than typical hurricanes.
6. [L42] Not "high damage".
7. [L53-54] Strange inclusion of result in introduction.
8. [L54-55] Suggest citing Maier-Gerber et al. here.
9. [L60] Suggest "high" to "large".
10. [L61] Referent of "they" unclear.
11. [L64-70] Awkward transitions throughout paragraph.
12. [L75] Suggest "are" to "have been".]
13. [L88] Double period.
14. [L89] Remove "the".
15. [L92] Reconsider use of singular throughout this sentence.
16. [L103] Suggest "previous" to "prior".
17. [L106] "Section" misspelled.
18. [L129] Suggest, "The CPS uses three parameters to define thermal structure:"
19. [L135] Isn't there a better reason than "simplicity"? Could it be that none of the members differ because this is a larger-scale parameter? If there are large differences in B between members, then I think that it should be considered as an important dimension of the CPS.
20. [L157] This statement assumes that variance is large when gradients are large: not necessarily the case if the state is very strongly constrained by observations.
21. [L183-184] Please provide citations at the ends of each of these sentences.
22. [L191 and L193] Are these the same "R" features described as "moderate" and "large"?
23. [L199] Should be "events associated" I think.
24. [L203-204] Suggest simplifying the first two sentences.
25. [L214-217] I think that this discussion of the storm's vertical thermal and PV structure would really benefit from a cross-section through the centre.
26. [L222-223] It is really unclear in this discussion that the authors consider 38% a small number. Maybe this sentence should start with "Despite the prevalence of convection, only 38% ...".
27. [L222] What area is used here?
28. [L224] How does stretching affect the cyclone path here?
29. [L227] Why is the fraction of parameterized convection at later times relevant?
30. [L227] Could also add that the DWD analysis does not show any fronts associated with Zorbas at this time (it showed them at 1200 UTC 27 September, but not thereafter).
31. [L238] Add missing "the".
32. [L239] The previous section showed that the PV streamer was present, not that it was "crucial".

33. [L241]  It would be useful to say a couple of words here about the impact on Zorbas' outcomes in the members, otherwise this whole development of the streamer classifications happens without a clear motivation.
34. [L242]  Suggest "the important" rather than "this significant".
35. [L258]  Does not go to 73N.
36. [L259]  Don't use contractions in professional writing (yes, I know).
37. [L261]  Suggest "explaining" to "to explain".
38. [L276]  Is this J1 or J2 in Fig. 2?
39. [L279]  "Relative" to what?
40. [L280-284]  Suggest deleting to avoid pre-summarizing.
41. [L306-307]  Equation 1 has nothing to do with geostrophic winds, so why are the balance concepts introduced here?
42. [L314]  The diabatic heating term includes effects from radiation, turbulence, etc.  Why would they necessarily not be important here?
43. [L315]  Does using W as the PV "master" have an impact on the results?
44. [L317]  Hypothesizing about the size of terms is not acceptable.
45. [L327 and 329]  Why is there a mix between normalized and non-normalized PV error between Eq. 3 and Fig. 9?  It would be much simpler to pick one measure and stick to it.
46. [L336]  Is it not the absolute differences (PV*) that are of most interest here?
47. [L338]  How does the "cyclonic pattern around the positive PV difference" "counteract the advection of the PV difference"?  By definition, a circulation centred on a feature does not advect the feature.
48. [L378]  The W cutoff is over North Africa.  Could turbulent mixing not also be affecting PV at this level?
49. [L390]  Suggest "than" to "as".
50. [L397-399]  It may be true that most ensemble members have less precipitation than the short-range forecast; however, that is not a conclusion that can be drawn from the subensemble mean. Shifts in precipitation structure will lead to a "smearing" of the ensemble mean and a reduction in peak values.  The ensemble mean forecast should not look like the QPE unless there is no uncertainty in the event.
51. [L404-406]  Remove this sentence after plotting track only to the common final time.
52. [L404-409]  Why is the within-cluster variability important for this discussion?  Consider removing the members from Fig. 11 and associated discussion in the text.
53. [L421]  This is a complicated description for an introduction.  A simpler introduction here would be more effective, followed by the more thorough description on L450 (including citations).
54. [L438-439]  Run-on sentence.
55. [L450]  Remove extra comma.
56. [L458-459]  This conclusion seems to be a tautology.  The formation of an anticyclonic outflow layer is essential for any tropical cyclone (or medicane).  This layer will necessarily have an anticyclonic PV anomaly.  So the replacement of a cyclonic PV anomaly at upper levels with an anticyclonic one is simply a requirement for tropical transition that is not case-dependent.
57. [L460-479]  This member-by-member analysis is not sufficiently in-depth to be highly valuable for readers; however, it takes up a lot of space in the text.  I think that it would be fair to mention that there are some members whose evolution does not follow the story line, and that these members tend to be the ones that lie on the borders between the subensembles.  If this discussion is to be kept, then I think that it should be made more robust and moved to the supplemental material for (very) keen readers.
58. [L482]  Reverse "storm" and "internal".

59. [L484]  It is not clear what the precipitation analysis has yielded in terms of results that feed into the study conclusions.
60. [Table 1]  This is not a clear caption.
61. [L502]  What "process" is being referred to here?
62. [L519]  Is the Fig. 7 feature really "large-scale"?  It is larger than convective, but nowhere near the Rossby radius.
63. [L520]  This is the first appearance of the word "energy" in reference to the results of this study. Because the reader does not know the size of wind differences that contribute to the PV feature shown in Fig. 7, it is impossible for them to assess the size of differences compared to the background.
64. [L523-526]  We also showed that cyclogenesis and TT can occur in the lee of the Alps during trough passage (shameless self-promotion of McTaggart-Cowan et al. 2010a and b, but they're highly relevant to this discussion).
65. [L531]  I do not believe that making "all data available from the authors upon request" is aligned with the FAIR data policy adopted by WCD.
66. [Fig 1]  Suggest adding "at 6-hourly intervals in each panel" on the second line.
67. [Fig. 3]  Explain the circle in panel (e) (not referenced until L418).  Note that the reference vector is above panel (d).  The last sentence seems to be missing.  What are the white dots in the lower row?
68. [Fig. 6]  Is panel (a) really needed?  All of this information is available in earlier figures, and the grey boxes in this panel appear to have been subjectively determined.  Note that there are no blue lines in panel (c) as implied by the caption.
69. [Fig. 7]  What are the white areas?  Adding normalized differences in the inset would be useful so that the reader has a better sense of how unusual the patterns over the Gulf of St. Lawrence really are.  A cross-section across the jet core might be an interesting way to show the depth of this feature.
70. [Fig. 8]  Consider whether the right-hand column of plots is really needed.  These figures are complicated to interpret, barely referenced in the text, and do not appear to contribute to the discussion.  Suggest adding a semicolon at the end of the first line of the caption.
71. [Fig. 10]  Why are the blue contours discontinuous (especially in panels (j) and (k))?
72. [Fig. 11]  Suggest adding row labels as "mean", "largest" and "smallest".  Consider stopping the track in (d) at 0000 UTC 30 September as for the other panels.  This will simplify interpretation, text and caption.
73. [Fig. 12]  Consider adding a full legend with three primarily classifications ("before", "between" and "after"), each with three sub-classes ("medicane", "weak medicane" and "no medicane") so that all plot symbols are shown in the legend.  Does the marker size scale progressively with upper-level thermal wind strength?  What is the no-medicane threshold?

---

## Referee Report (RR2)

**Notes for WCD-0001, "How an uncertain short-wave perturbation on the North Atlantic wave guide affects the forecast of an intense Mediterranean cyclone (Medicane Zorbas)" by R. Portmann et al.**

**Background**

This is a massively improved manuscript. The authors have clearly taken the time that they needed to revise their approach to the study and present the material in a clear and logical way. The result is a document that is fun to read and easy to draw conclusions from.

I have one general concern regarding the robustness of the storyline for different (primarily earlier) initialization times, and a few specific questions that may require some adjustments to the analysis or text to address (notably Specific Comments # 28, 36, 37 and 59). However, once the authors have completed a set of relatively minor revisions in response to these remaining issues, I look forward to seeing this article in print. I truly appreciate the efforts that the authors have made to bring this manuscript up to a very worthy standard.

Reviewer: Ron McTaggart-Cowan

Recommendation: Minor Revisions

**General Comments**

1. This detailed analysis is presented for a single initialization time. Do you know if these results are robust for other (primarily earlier) initializations? Does the previous set of ensemble forecasts show a similar range of solutions for the PV streamer that map onto difference cyclone developments? I understand that repeating this analysis for other initializations would be a huge undertaking and make the work much longer than it is. However, can a subjective evaluation of the robustness of the result be made relatively easily and the results noted in the conclusions?

**Specific Comments**

1. [L10] Suggest "… amplified **on** the stratospheric side …".
2. [L24] Suggest the more direct "… investigates the impact of PV streamer position uncertainty on medicane development".
3. [L27] Would everyone agree that medicanes are **the** main meteorological threat in the Mediterranean region? There are other local wind hazards and there's the flooding on the Alpine south-side that can be pretty intense. I don't mind if you want to stick with the current wording, but perhaps consider adjusting this to something like, "… are therefore a leading meteorological threat …".

4.  [L34] Add "… a range of modelling case studies …" to make it clear that the errors are in a numerical model, not the real atmosphere.
5.  [L40] Replace "far equatorward reaching" with "high amplitude" or "meridionally extended".
6.  [L48] Suggest citing Clark et al. (2017) and Keller et al. (2019) when mentioning ET here.
7.  [L55 and L57] Wikipedia is an unstable reference, so ideally primary material should be cited here instead. If Wikipedia absolutely needs to be cited, then an access date should be provided.
8.  [L57] Suggest simplifying to "This cyclone belonged to a special class …".
9.  [L62] Suggest "significant damage" as more common.
10. [L62] The note that "sometimes they acquire the typical appearance of a hurricane" suggests that sometimes medicanes don't acquire hurricane-like characteristics. Given that medicanes are typically identified by morphology, this seems unlikely and therefore the statement should be strengthened.
11. [L73] Remove comma after "during ET".
12. [L97-107] This paragraph is too general to be particularly useful, and contains a defense of the experimental design that would be better placed in a data/methods section. It also disrupts the flow between the "study objectives" paragraph and the "outline" paragraph at the end of the introduction. I think that this paragraph should be removed.
13. [L131-L133] The TRMM mission ended in 2015, replaced by the GPM. Are the 2018 data used here processed by the legacy TRMM algorithm or the newer IMERG algorithm? The text here may be completely correct, but please confirm.
14. [L138-L139] The idea of intersection points between a trajectory and a layer is a bit strange. I would ordinarily have thought of a trajectory line intersecting a 2D plane (e.g. the 325 K surface). Doesn't the use of the 322.5 K lower boundary for the layer simply imply that the points are defined as the locations at which the ascending WCB trajectories cross the 322.5 K surface? The Fig. 1 caption suggests that the simpler 325 K surface definition is used, at odds with the description here. The Fig. 2 caption suggests that the 322.5 K surface is used instead. I realize that the difference between these is relatively small, but consistency is desirable.
15. [L140] Suggest "around **the** cyclogenesis position".
16. [L148] This use of the CPS does not seem to distinguish between convection- and seclusion-induced warm cores. Is there any risk that some of the members classified as medicanes by the CPS are the result of frontal seclusion, a process that is not associated with tropical transition? Might B be useful to distinguish between storms at the early stages of seclusion and those whose warm cores are diabatically generated?
17. [L172-L182] This is an excellent description of your testing technique: very well done.
18. [L184] Suggest reversing "provides first".
19. [Fig. 1] The PV streamer has already formed in (d), so the phrase "before the formation of the PV streamer over the Mediterranean" should be removed or replaced with something like, "during the development of the PV streamer".
20. [Fig. 2] The contour interval for SLP should be noted in the caption (it looks like 4 hPa). The fact that 500 hPa height is shown in yellow contours should be noted, as should the contour interval used for this field.

21. [Fig. 2] Suggest adding "322.5 K **isentropic** level" to avoid confusion with the equivalent potential temperature that serves as the departure threshold.
22. [L206] Suggest inverting to read, "The PV streamer broke up at the time of cyclogenesis, resulting in the …".
23. [L207] Suggest replacing "following" with "subsequent".
24. [L209] Suggest referring to Fig. 3a for locations to help with this discussion.
25. [L245] "Landfall" is usually written as a single word.
26. [L248] The phrase "about one-day period" is approximate and slightly strange.  Was this an 18-h period?  The exact length doesn't matter very much here, so the phrase could simply read "… prior to cyclogenesis, the initial cyclone intensification, and the formation …".
27. [Fig. 4] A greyscale bar should be included for the brightness temperatures.
28. [L267] To be "substantial" this spread would need to be larger than typical ensemble spread at these lead times.  If such a typical spread is known, it would be useful to add this line to Fig. 5 for reference.  If it is not (and is not readily computed), then this sentence could be restructured to focus on the decrease in spread in the medium range.
29.  [L287] Suggest removing "a" before "substantial".
30. [L288-290] Suggest simplifying to, "To establish the dynamical link between uncertainties in the position (thus thermal structure) of the cyclone and upstream uncertainties …".
31. [L290] Suggest "classifies" instead of "allows separating the".
32. [L292] Suggest "(clusters)" rather than the current subordination.
33. [L297] Suggest a full stop rather than a colon.
34. [L321] Suggest "… and amplification of these uncertainties along the …".
35. [L324] This section title would be more complete as "Uncertainty propagation from the North Atlantic jet streak to the Mediterranean PV streamer".
36. [L336-L338] I agree that there are no WCB trajectories ending in this region.  However, this does not mean that moist diabatic processes are not relevant to the uncertainty.  I've included a satellite retrieval (Fig. 1 below) that shows extensive high-topped cloud cover in the region.  In addition to potential latent heating effects, these clouds will affect the radiative heating profile.  If the clouds are handled differently by the members in the different composite groups, they could explain the differences in the solutions rather than "dry upper-tropospheric dynamics" (L338).  Imagine, for example, that members in which the local SPPT coefficients suppress the radiative heating tendencies do not amplify the ridge while those in which the coefficients amplify the heating signal create a much more robust ridge.  If this is truly the sensitive region for the Mediterranean streamer, this difference in heating could cause the eventual separation of the solutions that is observed.  I haven't demonstrated that any of this is true, but the analysis in the manuscript does not rule out this possibility despite the assertions in the text.
37. [L346 and L362-367] This is a very 2D way of describing the tropopause evolution.  What if you considered the "approach" of the high-PV contours as a steepening of the tropopause or development of a tropopause fold.  This is of course necessarily related to the jet streak, but might give the readers a useful way to conceptualize the process that's promoting wave amplification in some members.  For example, the approach of a northern stream PV

perturbation towards the jet seems dynamically similar to the events discussed by Winters and Martin (2017).

38. [L351] Please avoid parenthetical negation (Robock 2010).

39. [L357] This trough is referred to as S2 in Fig. 1d. I think that's good because it's definitely a PV streamer rather than a trough at this time; however, the reference needs to be corrected here. Also, there is no S2 in Fig. 1c. Is it possible that the panel references are inverted here?

40. [L368-370] As in Comment #28, it would be useful to have a spread climatology in Fig. 5 for reference so that it is clear that this spread reduction is larger than would be expected by the decreasing lead time.

41. [L376-L379] There is also a notable difference in streamer tilt, with W more positively tilted and E notably more meridional (note the westward shift of the high-PV region over Eastern Europe in the E cluster). This suggests that the streamer may be in a slightly different stage of its life cycle in the different groups.

42. [Fig. 11] If it doesn't take too much effort, you could consider masking out significance regions that are too small to be meaningful. This would help to clean up the otherwise-beautiful plots a bit without any loss of important information.

43. [L411] Remove comma before "because".

44. [L413] Remove dash after "upper-" (this is not a compound adjective).

45. [Fig. 12] I really like Fig. 12 and the associated discussion, but the precipitation panel should appear as its own figure (Fig. 13). There is no association of either axis between panels a and b with panel c, so placing them in a single figure is not beneficial.

46. [L427] This sentence could/should be moved to the figure caption.

47. [L436] Suggest replacing "following" with "subsequent".

48. [L439] Consider replacing "crucial role" (which doesn't really fit with the remainder of the sentence) with "necessity".

49. [L439] Replace "… heating and cross-isentropic …" with "heating, cross-isentropic".

50. [L444] Change "much lower" to "much reduced" to avoid ambiguity with precipitation fluxes in the vertical.

51. [L445] Change "indicating much lower latent heating" to "indicative of a reduction in column-integrated latent heating".

52. [L445-L446] Without more context or an analysis of why this precipitation bias exists, this does not seem like an "interesting side remark". I think that this sentence should be removed to maintain the focus of this discussion.

53. [L447] Suggest changing "pathway" to "evolution".

54. [L447] Unless WCD uses the APA style, I do not think that the word following a colon should be capitalized.

55. [L451] Consider replacing "… this storyline and they are mostly …" with "… this archetype, mostly …".

56. [L459] Suggest "… were used to assess how uncertainties originating in a short-wave perturbation on the North Atlantic wave guide influenced a downstream PV stream and, as a result, …".

57. [L461] Suggest inverting "appeared first" and changing "at" to "on".

58. [L461-463] This summary makes it sound as though the PV differences and the short-wave perturbation happened (by chance) to occur at the same time. Is there not a dynamical link between them? The subsequent sentence suggests that they were both prompted by the high-PV perturbation in the polar stream, but this all seems like a weaker connection than the earlier analysis implied.

59. [L465 and L491] As noted in Specific Comment #36, I think that the current analysis does not rule out the importance of upper-tropospheric moist processes (cloud formation) and their secondary effects.

60. [L465] The referent of his pronoun ("they") is unclear.

61. [L471] Suggest "… cyclogenesis that affects cyclone …".

62. [L478-480] I like this discussion and the context of systematic errors that you cast it in.

63. [L482] I think that your analysis (including trajectories) was robust enough to let you conclude using the definite article that "The reason for this …".

64. [L483-L484] Recommend putting a full stop after "region" and beginning the next sentence as "This prevented …".

65. [L484] Replace "… to be strong enough and reach …" with "… from being strong enough and reaching …".

66. [L488] Suggest adding "the" before "medium-range".

67. [L493-L494] I think that I understand what you're trying to say here, but the concept of "upstream influence" is strange enough that I think that a reformatting of this sentence would help to clarify your conclusion.

68. [L499] Suggest removing "for example".

69. [Fig. S4] Please use the full citation method requested at https://www.lightningmaps.org/about.

**References**

Robock, A. (2010), Parentheses are (are not) for references and clarification (saving Space), *Eos Trans. AGU, 91*(45), 419–419, doi:10.1029/2010EO450004.

Winters, A. C., and J. E. Martin, 2017: Diagnosis of a North American Polar–Subtropical Jet Superposition Employing Piecewise Potential Vorticity Inversion. *Mon. Wea. Rev.*, **145**, 1853–1873, https://doi.org/10.1175/MWR-D-16-0262.1.

**Figures**

[Figure]

**Figure 1: Composite satellite image for 24 September 2018 from NASA Worldview. Aqua/MODIS estimates of cloud top pressure are shown in warm colours for >500 hPa as shown on the colour bar.**

---

## Author Response (AR3)

Paper wcd-2019-1

**"How an uncertain short-wave perturbation on the North Atlantic wave guide affects the forecast of an intense Mediterranean cyclone (Medicane Zorbas)"**

Raphael Portmann, Juan Jesús González-Alemán, Michael Sprenger, and Heini Wernli

**Reply to the reviewers' comments:**

We thank the reviewers for again carefully reading and commenting on the manuscript. All comments have been carefully considered. In particular, the general comment by Reviewer 1 is addressed in detail and led to a corresponding paragraph in the conclusions of the article. The reference to Wikipedia is avoided by using two other references.

**Reviewer 1 (Ron McTaggart-Cowan)**

Notes for WCD-0001, "How an uncertain short-wave perturbation on the North Atlantic wave guide affects the forecast of an intense Mediterranean cyclone (Medicane Zorbas)" by R. Portmann et al.

**Background**

This is a massively improved manuscript. The authors have clearly taken the time that they needed to revise their approach to the study and present the material in a clear and logical way. The result is a document that is fun to read and easy to draw conclusions from.

*Many thanks for this very positive assessment. We are of course happy that our hard work during the last revisions led to a notable improvement of the manuscript.*

I have one general concern regarding the robustness of the storyline for different (primarily earlier) initialization times, and a few specific questions that may require some adjustments to the analysis or text to address (notably Specific Comments # 28, 36, 37 and 59). However, once the authors have completed a set of relatively minor revisions in response to these remaining issues, I look forward to seeing this article in print. I truly appreciate the efforts that the authors have made to bring this manuscript up to a very worthy standard.

Reviewer: Ron McTaggart-Cowan
Recommendation: Minor Revisions

**General Comments**

1. This detailed analysis is presented for a single initialization time. Do you know if these results are robust for other (primarily earlier) initializations? Does the previous set of ensemble forecasts show a similar range of solutions for the PV streamer that map onto difference cyclone developments? I understand that repeating this analysis for other initializations would be a huge undertaking and make the work much longer than it is. However, can a subjective evaluation of the robustness of the result be made relatively easily and the results noted in the conclusions?

*Thank you for this comment. Indeed, it is quite difficult to say if a similar uncertainty propagation and differences in the cyclone evolution occurs for other initializations. We can, however, look at the ensemble spread of geopotential height at 500 hPa at 0000 UTC 27 Sep 2018 for these initializations (see Fig. 1). For the forecast initialized at 0000 UTC 24 Sep 2018, i.e. the initialization considered in the paper, ensemble spread is zonally extended at the tip of the PV streamer, indicative of the zonal position uncertainties discussed in detail in the paper (Fig. 1c). A similar zonally extended spread pattern is visible for the forecast initialized one day earlier (Fig. 1b), However, the enhanced ensemble spread reaches further northeast towards eastern Europe and the Black Sea. This indicates that in this forecast, the overall direction of the PV streamer was mostly uncertain. The forecast initialized two days earlier (Fig. 1a), shows an even more different spread pattern with a meridionally extended region of enhanced ensemble spread poleward of the PV streamer in the operational analysis. This indicates that in this forecast, the reason for the enhanced ensemble spread is not linked to the PV streamer but rather to larger-scale uncertainties in the wave pattern and the onset of the Rossby wave breaking. Based on this analysis, we add a paragraph in the conclusions as requested by the reviewer:*

" Finally, we note that this study is based on one particular ensemble initialization time where the occurrence of Rossby wave breaking and cyclone formation was already certain (but not the exact zonal position of the resulting PV streamer). Later initializations are characterized by much lower positional uncertainties of the PV streamer and, hence, strongly reduced ensemble spread of 500 hPa geopotential height (as shown in Fig. 5). In contrast, earlier initializations have even larger ensemble spread with high values extending from the Mediterranean to eastern Europe (not shown), indicating uncertainty in the onset of Rossby wave breaking and the overall orientation of the PV streamer. This implies that the uncertainty patterns discussed in this study are not representative for earlier initializations of the ensemble. While in this study, the ensemble forecast was used to investigate uncertainties in the PV streamer position and the evolution of the Mediterranean cyclone, similar studies with forecasts initialized earlier could shed light on processes that determine uncertainties in the onset of the wave breaking. However, they would then be less suitable for investigating uncertainties in the formation of the medicane."

[Figure]

*Figure 1: Ensemble spread of geopotential height at 500 hPa over the Mediterranean for initializations at (a) 0000 UTC 22 Sep 2018, (b) 0000 UTC 23 Sep 2018, and (c) 0000 UTC 24 Sep 2018 valid at 0000 UTC 27 Sep 2018. The black line marks the 2 PVU contour at 325 K in the operational analysis*

Specific Comments

1. [L10] Suggest "... amplified on the stratospheric side ...".
2. [L24] Suggest the more direct "... investigates the impact of PV streamer position uncertainty on medicane development".
3. [L27] Would everyone agree that medicanes are the main meteorological threat in the Mediterranean region? There are other local wind hazards and there's the flooding on the Alpine south-side that can be pretty intense. I don't mind if you want to stick with the current wording, but perhaps consider adjusting this to something like, "... are therefore a leading meteorological threat ...".
4. [L34] Add "... a range of modelling case studies ..." to make it clear that the errors are in a numerical model, not the real atmosphere.
5. [L40] Replace "far equatorward reaching" with "high amplitude" or "meridionally extended".
6. [L48] Suggest citing Clark et al. (2017) and Keller et al. (2019) when mentioning ET here.

Thanks for these suggestions. All of them have been considered.

7. [L55 and L57] Wikipedia is an unstable reference, so ideally primary material should be cited here instead. If Wikipedia absolutely needs to be cited, then an access date should be provided.

Thanks for this comment. We avoided using Wikipedia and replaced it by two other references.

8. [L57] Suggest simplifying to "This cyclone belonged to a special class ...".
9. [L62] Suggest "significant damage" as more common.

Thanks, the manuscript has been changed accordingly.

10. [L62] The note that "sometimes they acquire the typical appearance of a hurricane" suggests that sometimes medicanes don't acquire hurricane-like characteristics. Given that medicanes are typically identified by morphology, this seems unlikely and therefore the statement should be strengthened.

The sentence has been adapted to "They acquire…

11. [L73] Remove comma after "during ET".
12. [L97-107] This paragraph is too general to be particularly useful, and contains a defense of the experimental design that would be better placed in a data/methods section. It also disrupts the flow between the "study objectives" paragraph and the "outline" paragraph at the end of the introduction. I think that this paragraph should be removed.

Thanks for pointing this out. We agree that this paragraph is interrupting the flow. Therefore, the first part has been removed and the second part added to the previous paragraph, where it does not interrupt the flow.

13. [L131-L133] The TRMM mission ended in 2015, replaced by the GPM. Are the 2018 data used here processed by the legacy TRMM algorithm or the newer IMERG algorithm? The text here may be completely correct, but please confirm.

The data processed by the original TRMM algorithm is available until 2019: https://trmm.gsfc.nasa.gov/3b42.html

14. [L138-L139] The idea of intersection points between a trajectory and a layer is a bit strange. I would ordinarily have thought of a trajectory line intersecting a 2D plane (e.g. the 325 K surface). Doesn't the use of the 322.5 K lower boundary for the layer simply imply that the points are defined as the locations at which the ascending WCB trajectories cross the 322.5 K surface? The Fig. 1 caption suggests that the simpler 325 K surface definition is used, at odds with the description here. The Fig. 2 caption suggests that the 322.5 K surface is used instead. I realize that the difference between these is relatively small, but consistency is desirable.

Please apologize for this inconsistency. The idea in Fig. 1 is to identify trajectories crossing an isentropic surface in order to see if WCB air parcels are present at that isentropic surface. However, the potential temperature value of the air parcel will mostly not be exactly 325 K, therefore a layer has to be specified in which air parcels are assigned to the 325 K level (i.e. if air parcels are close enough to that level their trajectories are identified as "trajectories crossing the 325K isentropic surface").

In contrast, in Fig. 2, all air parcels are shown that reach above the 322.5 K level in order to show which air parcels were able to reach high enough to affect the erosion of the PV cutoff (at the 325 K level shown in the plot or even at higher levels). So, the purpose here is slightly different. We changed the description in the data section to make this clearer.

15. [L140] Suggest "around the cyclogenesis position".
16. [L148] This use of the CPS does not seem to distinguish between convection- and seclusion-induced warm cores. Is there any risk that some of the members classified as medicanes by the CPS are the result of frontal seclusion, a process that is not associated with tropical transition? Might B be useful to distinguish between storms at the early stages of seclusion and those whose warm cores are diabatically generated?

This is indeed possible. In fact it is known that the warm core in some medicanes is actually produced by frontal seclusions (e.g. Fita and Flaounas 2018). We did not look at this in this study and possibly both scenarios occur in the ensemble prediction we used for this paper. The fact that most medicanes in this case are also associated to high precipitation rates and upper level PV erosion could be seen as indication that, here, warm cores are mostly driven by latent heating. However, it would probably require detailed trajectory analyses to properly differentiate between convection and seclusion-induced warm cores. Something that might be addressed in our planned follow up study.
If B is useful to distinguish the different mechanisms producing a warm core is difficult to judge. It would require a classification of the medicanes into these two types, which has, as stated above, not been done.

17. [L172-L182] This is an excellent description of your testing technique: very well done.

Thank you!

18. [L184] Suggest reversing "provides first".

Done

19. [Fig. 1] The PV streamer has already formed in (d), so the phrase "before the formation of the PV streamer over the Mediterranean" should be removed or replaced with something like, "during the development of the PV streamer".

It was changed to "before the formation of Medicane Zorbas"

20. [Fig. 2] The contour interval for SLP should be noted in the caption (it looks like 4 hPa). The fact that 500 hPa height is shown in yellow contours should be noted, as should the contour interval used for this field.
21. [Fig. 2] Suggest adding "322.5 K isentropic level" to avoid confusion with the equivalent potential temperature that serves as the departure threshold.

Thanks for carefully reading the caption. Changes have been made accordingly.

22. [L206] Suggest inverting to read, "The PV streamer broke up at the time of cyclogenesis, resulting in the ...".
23. [L207] Suggest replacing "following" with "subsequent".
24. [L209] Suggest referring to Fig. 3a for locations to help with this discussion.

25. [L245] "Landfall" is usually written as a single word.
26. [L248] The phrase "about one-day period" is approximate and slightly strange. Was this an 18-h period? The exact length doesn't matter very much here, so the phrase could simply read "... prior to cyclogenesis, the initial cyclone intensification, and the formation ...".

Thanks for the above suggestions. Changes have been made accordingly.

27. [Fig. 4] A greyscale bar should be included for the brightness temperatures.

Thanks for this comment. The satellite data is given in radiances and a conversion to brightness temperatures seems rather complicated and, in our opinion, does not provide substantial benefit to the reader. The satellite images are used to show the presence of clouds and their structure and no quantitative conclusions are drawn from it. We could, of course, add a greyscale bar for radiances, but we doubt that this is of benefit to the reader.

28. [L267] To be "substantial" this spread would need to be larger than typical ensemble spread at these lead times. If such a typical spread is known, it would be useful to add this line to Fig. 5 for reference. If it is not (and is not readily computed), then this sentence could be restructured to focus on the decrease in spread in the medium range.

Thanks for this comment. This is a very good idea. Unfortunately, we don't have access to enough data to calculate the climatological ensemble spread in this region. Therefore, the sentence has been modified accordingly to "The synoptic situation over the Mediterranean was associated to uncertainties in the operational ECMWF ensemble forecasts, which decreased particularly strongly for initialization later than three days prior to genesis of Zorbas."

29. [L287] Suggest removing "a" before "substantial".
30. [L288-290] Suggest simplifying to, "To establish the dynamical link between uncertainties in the position (thus thermal structure) of the cyclone and upstream uncertainties ...".
31. [L290] Suggest "classifies" instead of "allows separating the".
32. [L292] Suggest "(clusters)" rather than the current subordination.
33. [L297] Suggest a full stop rather than a colon.
34. [L321] Suggest "... and amplification of these uncertainties along the ...".
35. [L324] This section title would be more complete as "Uncertainty propagation from the North Atlantic jet streak to the Mediterranean PV streamer".

Thanks for these suggestions, they have been adopted.

36. [L336-L338] I agree that there are no WCB trajectories ending in this region. However, this does not mean that moist diabatic processes are not relevant to the uncertainty. I've included a satellite retrieval (Fig. 1 below) that shows extensive high-topped cloud cover in the region. In addition to potential latent heating effects, these clouds will affect the radiative heating profile. If the clouds are handled differently by the members in the different composite groups, they could explain the differences in the solutions rather than "dry upper-tropospheric dynamics" (L338). Imagine, for example, that members in which the local SPPT coefficients suppress the radiative heating tendencies do not amplify the ridge while those in which the coefficients amplify the heating signal create a much more robust ridge. If this is truly the sensitive region for the Mediterranean streamer, this difference in heating could cause the eventual separation of the solutions that is observed. I haven't demonstrated

that any of this is true, but the analysis in the manuscript does not rule out this possibility despite the assertions in the text.

Thank you for this comment and the attached image. We agree that, given the presence of clouds in the vicinity of the region, where the uncertainty amplifies, moist diabatic processes can not be excluded by our analysis. We apologize for not formulating this part more carefully. We changed the corresponding parts in the manuscript. We now only exclude a substantial contribution of WCBs, but not of moist diabatic processes in general.

37. [L346 and L362-367] This is a very 2D way of describing the tropopause evolution. What if you considered the "approach" of the high-PV contours as a steepening of the tropopause or development of a tropopause fold. This is of course necessarily related to the jet streak, but might give the readers a useful way to conceptualize the process that's promoting wave amplification in some members. For example, the approach of a northern stream PV perturbation towards the jet seems dynamically similar to the events discussed by Winters and Martin (2017).

Thanks for this perspective. We added a sentence on that "The strengthening of the jet streak in such a situation can also be understood as a steepening of the tropopause in this region (e.g. Winters and Martin, 2017)."

38. [L351] Please avoid parenthetical negation (Robock 2010).

Thanks, sentence has been changed.

39. [L357] This trough is referred to as S2 in Fig. 1d. I think that's good because it's definitely a PV streamer rather than a trough at this time; however, the reference needs to be corrected here. Also, there is no S2 in Fig. 1c. Is it possible that the panel references are inverted here?

Thanks for seeing this, indeed, panel references (d and c) were inverted.

40. [L368-370] As in Comment #28, it would be useful to have a spread climatology in Fig. 5 for reference so that it is clear that this spread reduction is larger than would be expected by the decreasing lead time.

We agree that it would be interesting to see how large Z500 ensemble spread (and the spread decrease) is compared to the climatological evolution in this region. However, this would require further analysis with additional data. On a global northern hemispheric average, we know that this evolution is very gradual (e.g. Rodwell et al. 2018). Most likely, this is also the case for the Mediterranean. Hence, it is very likely that the almost step-like evolution in this case cannot be fully explained by a climatological decrease of ensemble spread with lead time. We therefore decide to stick to the current formulation.

Rodwell, M. J., D. S. Richardson, D. B. Parsons, and H. Wernli, 2018: Flow-Dependent Reliability: A Path to More Skillful Ensemble Forecasts. *Bull. Amer. Meteor. Soc.*, 99, 1015–1026, https://doi.org/10.1175/BAMS-D-17-0027.1.

41. [L376-L379] There is also a notable difference in streamer tilt, with W more positively tilted and E notably more meridional (note the westward shift of the high-PV region over Eastern Europe in the E cluster). This suggests that the streamer may be in a slightly different stage of its life cycle in the different groups.

Thanks for this observation. We agree that the tilt is different between clusters. However, it is not straightforward to clearly separate differences in the streamers tilt and the position of the streamers tip. The latter is obviously strongly influenced by the former. And as the clustering focuses on the position of the streamer, we decided to stick to the current formulation in order to avoid confusion (as we are not discussing the streamers tilt anywhere else in the article).

42. [Fig. 11] If it doesn't take too much effort, you could consider masking out significance regions that are too small to be meaningful. This would help to clean up the otherwise beautiful plots a bit without any loss of important information.

Thanks for this suggestion. Very small significance regions have been masked out.

43. [L411] Remove comma before "because".
44. [L413] Remove dash after "upper-" (this is not a compound adjective).

Thanks, text has been changed accordingly.

45. [Fig. 12] I really like Fig. 12 and the associated discussion, but the precipitation panel should appear as its own figure (Fig. 13). There is no association of either axis between panels a and b with panel c, so placing them in a single figure is not beneficial.

Thanks for this comment. We agree that the axes of panel c are not associated to the ones of panels a and b. However, the legend is the same for all panels. Also, the discussion of the different panels is strongly connected. Therefore, we decided to keep them within the same Figure. To better avoid potential confusions with the axes of panels a and b, we moved the y-axis labels to the left of panel c.

46. [L427] This sentence could/should be moved to the figure caption.
47. [L436] Suggest replacing "following" with "subsequent".
48. [L439] Consider replacing "crucial role" (which doesn't really fit with the remainder of the sentence) with "necessity".
49. [L439] Replace "... heating and cross-isentropic ..." with "heating, cross-isentropic".
50. [L444] Change "much lower" to "much reduced" to avoid ambiguity with precipitation fluxes in the vertical.
51. [L445] Change "indicating much lower latent heating" to "indicative of a reduction in column-integrated latent heating".
52. [L445-L446] Without more context or an analysis of why this precipitation bias exists, this does not seem like an "interesting side remark". I think that this sentence should be removed to maintain the focus of this discussion.
53. [L447] Suggest changing "pathway" to "evolution".
54. [L447] Unless WCD uses the APA style, I do not think that the word following a colon should be capitalized.
55. [L451] Consider replacing "... this storyline and they are mostly ..." with "... this archetype, mostly ...".
56. [L459] Suggest "... were used to assess how uncertainties originating in a short-wave perturbation on the North Atlantic wave guide influenced a downstream PV stream and, as a result, ...".
57. [L461] Suggest inverting "appeared first" and changing "at" to "on".

Thanks for the above suggestions, they have all been adopted.

58.[L461-463] This summary makes it sound as though the PV differences and the short-wave perturbation happened (by chance) to occur at the same time. Is there not a dynamical link between them? The subsequent sentence suggests that they were both prompted by the high-PV perturbation in the polar stream, but this all seems like a weaker connection than the earlier analysis implied.

Thanks for pointing this out. We changed the sentence to "They were tightly connected to…" to make the connection clearer.

59. [L465 and L491] As noted in Specific Comment #36, I think that the current analysis does not rule out the importance of upper-tropospheric moist processes (cloud formation) and their secondary effects.

See reply to comment #36

60. [L465] The referent of his pronoun ("they") is unclear.

Thanks, the sentence has been made clearer.

61. [L471] Suggest "... cyclogenesis that affects cyclone ...".
62. [L478-480] I like this discussion and the context of systematic errors that you cast it in.

Thanks:-)

63. [L482] I think that your analysis (including trajectories) was robust enough to let you conclude using the definite article that "The reason for this ...".
64. [L483-L484] Recommend putting a full stop after "region" and beginning the next sentence as "This prevented ...".
65. [L484] Replace "... to be strong enough and reach ..." with "... from being strong enough and reaching ...".
66. [L488] Suggest adding "the" before "medium-range".

Thanks, suggestions have been adopted.

67. [L493-L494] I think that I understand what you're trying to say here, but the concept of "upstream influence" is strange enough that I think that a reformatting of this sentence would help to clarify your conclusion.

Thanks for pointing this out. We simplified the sentence to "...how frequently this mechanism *limits* the medium-range predictability of Mediterranean PV streamers and, as a result, …

68. [L499] Suggest removing "for example".
69. [Fig. S4] Please use the full citation method requested at https://www.lightningmaps.org/about.

Thanks, we adopted the citation in the manuscript and supplement accordingly.

**Figures**

[Figure]

Figure 1: Composite satellite image for 24 September 2018 from NASA Worldview. Aqua/MODIS estimates of cloud top pressure are shown in warm colours for >500 hPa as shown on the colour bar.

**Reviewer 2: Florian Pantillon**

"How an uncertain short-wave perturbation on the North Atlantic wave guide affects the forecast of an intense Mediterranean cyclone (Medicane Zorbas)" by Raphael Portmann, Juan Jesús González-Alemán, Michael Sprenger, and Heini Wernli

The paper has changed substantially since the last round of reviews. Its scope is now well introduced and its limitations explained, while discussions have gained in clarity and are supported by very illustrative figures. Altogether, the paper highlights the role of remote uncertainties along the upper-level wave guide in the medicane forecast and presents an interesting hypothesis of upper-level PV erosion by latent heating that allows tropical transition to take place. I thus recommend the paper for publication pending minor corrections listed below to further improve clarity.

Specific comments

l. 23–24 could you give a more general statement or perspective to conclude the abstract?

Thanks for this comment. We switched the last two sentences in the abstract and slightly changed them to "This study is the first that explicitly investigates the impact of PV streamer position uncertainty for medicane development. Overall, results extend current knowledge of the role of upstream uncertainties for the medium-range predictability and unsteady forecast behavior of Mediterranean cyclones including medicanes." In this way, it ends with a more general statement.

l. 27 "the main meteorological threat": exaggerated, as local high-impact weather is not necessarily related with cyclones (see HyMeX program)

Thanks, this statement has been changed to "leading meteorological threat", as suggested by reviewer 1.

l. 45 remove "for instance"
l. 52 typo: "causal link s"

Thanks, the manuscript was changed accordingly.

l. 54 "whose formation due to an upper-level PV streamer is the subject of this study": too much information for this sentence (and not supported by Wikipedia); better keep for the end of the paragraph?

Thanks, we agree that this statement does not really fit here. We removed it completely, because a similar statement is already present at the end of the paragraph.

l. 79 "similar to Maier-Gerber et al., 2019": explicit "for a tropical transition over the North Atlantic"?

Thanks, reference was changed accordingly.

l. 84–87 this sentence is disconnected from the previous one (no ET in Di Muzio et al. 2019)
It is true that ET is not covered in DiMuzio et al 2019. However, the connection of the sentences is via the topic of the different stages in the cyclone predictability/forecast jumps. We still think that this connection works.

l. 92–96 unclear (it becomes clearer in Section 4)

Unfortunately, without more specific comments about what is unclear, we don't see how we could make this paragraph clearer.

l. 228–230 do you mean that more than half of those parcels actually reach the upper levels? giving the average ascent in km or hPa rather than in K would be easier

Yes, that is what we mean. We provided the heating rate in K because what is relevant for the diabatic erosion is the isentropic level which the air parcels reach, rather than the pressure level or height. We therefore stick to the current formulation.

l. 230–231 "they contributed substantially": too strong statement

We deleted "substantially".

l. 231 "Despite limitations of trajectories in convective situations": detail (convection has not been mentioned yet)

Thanks for this comment. We removed this part of the sentence.

l. 232 "as a result": why?

Because replacing the upper-level cold core with a warm-core requires to get rid of the positive upper-level PV anomaly (which is necessarily associated to a mid/upper tropospheric cold core).

l. 233–235 the link with the previous sentence (PV erosion) is not straightforward

The previous sentence also contains that low-level air with high THE was important to form a warm core. This is also the case for the coupling index. Therefore, we think that this sequence of sentences is reasonable and stick to the current formulation.

l. 255 typo: "precentile"

Thanks for pointing this out, it was corrected.

l. 278 "only 21 ensemble members predicted a medicane" that's not bad!

We think this judgement strongly depends on the perspective. From a forecaster's perspective, 21 members is less than 50% of the cases, which could be interpreted as "bad". According to Di Muzio et al. 2019 (their Fig. 4), the evolution of medicane predictability with lead time is very variable. For example, for Ilona, more than 60% of the members forecasted a medicane three days ahead (i.e. a surface cyclone with an upper level warm core), for Trixie more than 60% forecasted no surface cyclone at all with 3 days lead time (and the remaining 40% did mostly not forecast a medicane). In order to cleanly judge if the forecast of a medicane is good or bad, we would need a stable reference, i.e. average forecast skill over many medicanes, which I think we don't have yet. The word "only" was used in this case simply because less than 50% of the members forecasted a medicane.

l. 299 refer to black contour in Fig. 7 for the analysis

Thanks, we adopted this suggestion.

l. 337 unclear in what the region is "clearly stratospheric": is it associated with PV>2pvu? It also appears to contradict the "upper-tropospheric" dynamics and forecast uncertainties mentioned below

Thanks, we clarified this by adding that it is in a region with PV > 2 PVU and changed "upper-tropospheric" to "tropopause-level".

l. 350 a westward shift of the wave pattern is apparent rather than a superimposed short-wave pattern

Thanks for this comment. Maybe this is a question of perspective. We would argue that it is definitely not a shift of the large-scale wave pattern, as the troughs over Newfoundland and Europe are pretty much the same in the two clusters at 0000 UTC 25 Sept 2018. In fact, the western flank of the ridge in cluster E is "steeper" (i.e. more meridionally directed) than in cluster W, and vice versa for the western flank. This results in the wave-like shape of the mean 2 PVU contour in cluster W, if cluster E is taken as reference (or also vice versa). Or in other words: It is not possible to bring the two mean 2 PVU contours to cover just by shifting them around, at least to our eyes. We acknowledge your perspective, which is obviously slightly different, but as it still seems to be the best description of the situation to us, we stick to the current formulation.

l. 353–355 emphasize that the negative (positive) difference becomes more and more (less and less) significant?

Thanks for pointing this out, a sentence about this aspect has been added.

l. 367 "large drop" seems exaggerated for a decrease that is steep but continuous

Thanks for pointing this out. We agree and changed it to "substantial reduction"

l. 379 "in these regions": which ones? Refer to white contours?

Thanks for pointing out this unclarity. We added a reference to the white contours in Fig. 11.

l. 394 Fig 11g

Thanks, this was changed accordingly.

l. 402 84h and 106h lead times are not mentioned before

They are the ones shown in Fig. 6, and are therefore mentioned before, but not explicitly. We changed the sentence structure and added a reference to Fig. 6 to make this clearer.

l. 407–408 refer to Fig. 6 again?

Thanks for this suggestion, we added such a reference.

l. 421 rephrase: the stratospheric PV anomaly is not equivalent to the cyclone thermal structure

Thanks for this comment, we changed this part to "...and thereby the differences in the vertical thermal structure of the cyclones…"

l. 427ff remind which color corresponds to which cluster to help the reader follow the discussion

Thanks, we adopted your suggestion.

l. 465 "they": the significant differences?

Thanks, this unclarity was resolved by specifically using "these differences".

l. 496 better "relevance of a PV streamer position for medicane formation" (without uncertain)?

Thanks for this suggestion, we adopted it.

l. 497 indeed, climatological studies have shown that some regions are more prone to medicane formation than others, but what is meant here exactly?

Thanks for pointing out this unclarity, we added a sentence to clarify: "In this case, only members with cyclogenesis in a relatively confined region at the Libyan coast developed a medicane."

l. 515ff many DOI links are broken due to unwanted characters

We check many links and it seems to us that most of them work fine. For a few, we found that curved brackets around the doi in the bibtex file were a problem. We removed curved brackets for all references and all links should work now.

l. 651 The Wikipedia reference is incomplete (I let the Editor judge whether it is appropriate): https://en.wikipedia.org/wiki/Mediterranean_tropical like_cyclone#Zorbas_(27_Sep_%E2%80%93_1_Oct_2018)

Thanks for this comment. The Wikipedia reference has been replaced by two other references.

Figure 1 Explain labeled features, or refer to the text?

Thanks for this advice, we added "Labels mark relevant flow features (for details see text)" to the caption.

Figures 8–9 Labeling relevant features as in Figure 1 would be helpful and would strengthen the link between the different Sections

Thanks for this suggestion, we labelled the relevant features in Figures 8 and 9 accordingly.

Figure S5 do clusters 1, 2, 3 refer to C, W, E? The figure should be mentioned somewhere in the text or in the caption of Fig. 11 (or removed otherwise)

Thanks for noting this mistake and the missing reference. The caption of Fig. 11 and the supplement Fig. S5 have been changed accordingly.

**Reviewer 3**

I have already expressed my positive evaluation of this manuscript in the previous versions, since the results improve our understanding of the mechanisms of development of Medicanes. In the newer version, I repeat my appreciation for this work, evaluating positively the Authors' efforts to further improve the manuscript (very clear the present version of Fig. 12!). I think the paper is really mature for publication, a few comments needed for better clarification are reported below.

Minor points:

- Line 40: please clarify what you mean with "far equatorward reaching PV streamers";

We changed this according to a suggestion of reviewer 1 to "high amplitude PV streamers".

- Line 56 and elsewhere: Libya instead of Lybia;

- Line 136: Lagrangian instead of Langrangian;

- Line 173 and elsewhere: (Wilks, 2011) instead of (Wilks, D. S., 2011);

- Line 218: acquired instead of aquired;

Thanks for pointing at these mistakes. The manuscript has been changed accordingly

- Line 222: this suggests that the cyclone belongs to the first category of Medicanes, following Miglietta and Rotunno (2019);

Thanks for making this link. We agree that Zorbas probably fits into the first category. However, we do not analyze the role of baroclinicity and the WISHE mechanism explicitly for Zorbas. Therefore, we think that such a statement should not be made in the article.

- Lines 225-228: Include a reference to better explain this point, e.g. Fig. 9.10 in Holton, IV edition;

Thanks for this suggestion. We added the following sentence to explain this point better: "The larger $\theta_e$ of the low level air parcels, the higher they can rise through moist adiabatic ascent (see e.g. Holton (2004) their Fig. 9.10)"

- Line 230: (average … of about 30K): do you mean that the maximum equivalent potential temperature difference between the parcel and the environment is about 30 K?

No, we mean that they have been heated diabatically by 30 K during their ascent. We changed the phrase in brackets to "(average diabatic heating of about 30 K)".

- Figure 1 caption: the values of wind speed (white contours) are not indicated;

Thanks for pointing this out. We added it.

- Figure 2 caption: the reason for the white areas in the lower panels is not clear;

Thanks, we added the explanation that it is missing values due to orography.

- Figure 7 caption: the definition of PVav should be included in the caption;

Thanks, the definition has been added to the caption

- Figure 11 caption: the meaning of the small circles and of the bars in the insets is not clear;

Thanks for this remark. The explanation of the circle has been added (they are outliers). The bars show standard box plots (i.e. upper and lower quartile), we added the term "standard" to the description to make this more clear.

- Figure 5S caption: the names of the clusters should be C, E, W not 1, 2, 3.

Thanks for noting this. We changed the names.

[revised manuscript text omitted]